# Rectified LpJEPA: Joint-Embedding Predictive Architectures with Sparse and Maximum-Entropy Representations

**Yilun Kuang** [1]   **Yash Dagade** [2]   **Tim G. J. Rudner** [3]   **Randall Balestriero** [4]   **Yann LeCun** [1]

GitHub    Blog

## Abstract

Joint-Embedding Predictive Architectures (JEPA) learn view-invariant representations and admit projection-based distribution matching for collapse prevention. Existing approaches regularize representations towards isotropic Gaussian distributions, but inherently favor dense representations and fail to capture the key property of sparsity observed in efficient representations. We introduce Rectified Distribution Matching Regularization (RDMReg), a sliced two-sample distribution-matching loss that aligns representations to a Rectified Generalized Gaussian (RGG) distribution. RGG enables explicit control over expected $\ell_0$ norm through rectification, while its continuous truncated component admits a maximum-entropy characterization under expected $\ell_p$ norm and support constraints. Equipping JEPAs with RDMReg yields Rectified LpJEPA, which strictly generalizes prior Gaussian-based JEPAs. Empirically, Rectified LpJEPA learns sparse, non-negative representations with favorable sparsity–performance trade-offs and competitive downstream performance on image classification benchmarks, showing that RDMReg can enforce sparsity while preserving task-relevant information.

## 1. Introduction

Self-supervised representation learning has emerged as a promising paradigm for advancing machine intelligence without explicit supervision (Radford et al., 2018; Chen et al., 2020). A prominent class of methods—Joint-Embedding Predictive Architectures (JEPAs)—learn rep-

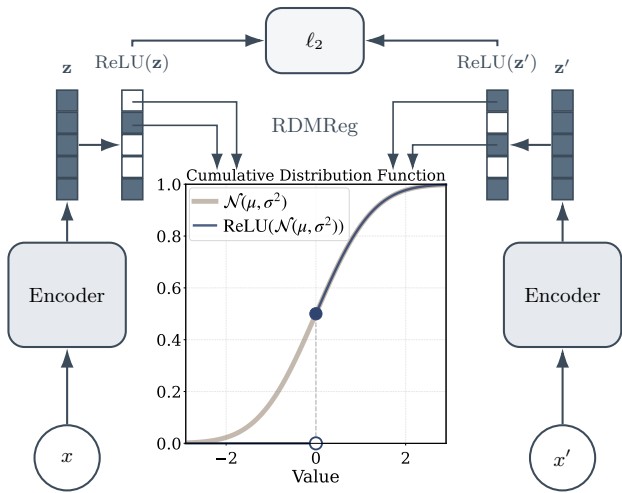

*Figure 1.* **Rectified LpJEPA.** Two views $(x, x')$ of the same underlying data are embedded and rectified to obtain $\mathrm{ReLU}(\mathbf{z})$ and $\mathrm{ReLU}(\mathbf{z}') \in \mathbb{R}^d$. Rectified LpJEPA minimizes the $\ell_2$ distance between rectified features while regularizing the $d$-dimensional rectified feature distribution towards a product of i.i.d. Rectified Gaussian distributions $\mathrm{ReLU}(\mathcal{N}(\mu, \sigma^2))$ using RDMReg. As a result, each coordinate of the learned representation aligns towards a Rectified Gaussian distribution (CDF shown above), a special case of the Rectified Generalized Gaussian family $\mathcal{RGN}_p(\mu, \sigma)$ when $p = 2$. In the absence of rectification on both the features and the target distribution, Rectified LpJEPA reduces to isotropic Gaussian regularization as in LeJEPA (Balestriero & LeCun, 2025).

resentations by enforcing consistency across multiple views of the same data in the latent space, while avoiding explicit reconstructions or density estimations in the observation space (LeCun, 2022; Assran et al., 2023).

By decoupling learning from observation-level constraints, JEPAs operate at a higher level of abstraction, enabling flexibility in encoding task-relevant information. However, invariance alone admits degenerate solutions, including complete or dimensional collapse, where representations concentrate in trivial or low-rank subspaces (Jing et al., 2022).

[1]New York University. [2]Duke University. [3]University of Toronto. [4]Brown University. Correspondence to: Yilun Kuang <yilun.kuang@nyu.edu>, Yann LeCun <yann.lecun@nyu.edu>.

*Proceedings of the 43rd International Conference on Machine Learning*, Seoul, South Korea. PMLR 306, 2026. Copyright 2026 by the author(s).

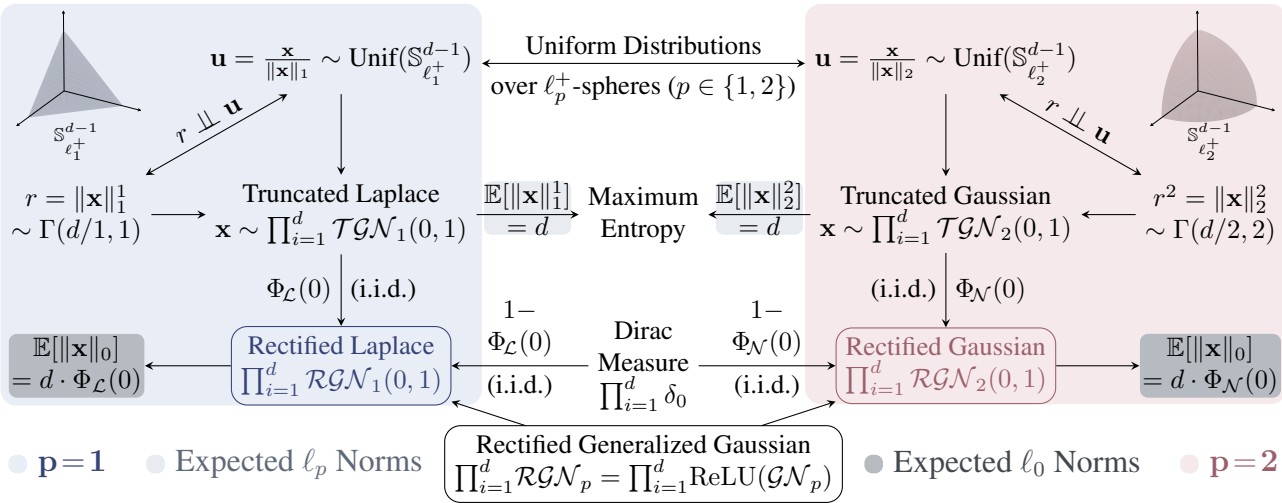

*Figure 2.* **Rectified Laplace ($p=1$) and Rectified Gaussian ($p=2$) as special cases of Rectified Generalized Gaussian distributions.** Assume $\mu = 0$ and $\sigma = 1$. For any $p > 0$, the Truncated Generalized Gaussian $\prod_{i=1}^d \mathcal{TGN}_p$ over the product support $(0, \infty)^d$ is the maximum differential entropy distribution under a fixed expected $\ell_p$-norm constraint. For $p \in \{1, 2\}$, $\prod_{i=1}^d \mathcal{TGN}_p$ further admits a radial–angular decomposition $\mathbf{x} = r \cdot \mathbf{u}$ with $r \perp\!\!\!\perp \mathbf{u}$, where $\mathbf{u}$ is uniformly distributed with respect to the surface measure on the unit $\ell_p$-sphere confined to the positive orthant and $r^p$ follows a Gamma distribution. Rectified Laplace and Rectified Gaussian arise via coordinate-wise mixing of the corresponding truncated distributions with a Dirac measure at zero, yielding a distribution with expected $\ell_0$-norm guarantees, where $\Phi_{\mathcal{L}}$ and $\Phi_{\mathcal{N}}$ denote the cumulative distribution functions of the standard Laplace and standard Gaussian distributions respectively.

Recent efforts culminate towards a distribution-matching approach for collapse prevention. In particular, LeJEPA (Balestriero & LeCun, 2025) introduces the SIGReg loss, which aligns one-dimensional projected feature marginals towards a univariate Gaussian across random projections, thereby regularizing the full representation towards isotropic Gaussian with asymptotic convergence guaranteed by the Cramér–Wold theorem. By decomposing high-dimensional distribution matching into parallel one-dimensional projection-based optimizations, SIGReg mitigates the curse of dimensionality and enables scalable representation learning. The resulting features are encouraged to be maximum-entropy under expected $\ell_2$ norm constraints, and projection-based matching can be viewed as extending second-order regularization methods such as VICReg (Bardes et al., 2022) beyond covariance matching by also penalizing higher-order distributional discrepancies.

However, restricting feature distributions to isotropic Gaussian severely limits the range of representational structures that can be expressed. In particular, isotropic Gaussian features alone do not capture a key property of effective representations: *sparsity*. Across neuroscience, signal processing, and deep learning, sparse and non-negative codes repeatedly emerge as efficient and interpretable representations (Olshausen & Field, 1996; Donoho, 2006; Lee & Seung, 1999; Glorot et al., 2011).

To this end, we propose Rectified Distribution Matching

Regularization (RDMReg), a two-sample sliced distribution-matching regularizer that aligns JEPA representations to the Rectified Generalized Gaussian (RGG) distribution, a novel family of probability distributions with controllable expected $\ell_p$ norms and induced $\ell_0$ regularizations from explicit rectifications. The continuous truncated component of RGG inherits a maximum-entropy characterization under expected $\ell_p$ norm and support constraints, while rectification adds explicit, analytically controlled sparsity through a point mass at zero.

The resulting method, Rectified LpJEPA, strictly generalizes LeJEPA, which arises as a special case corresponding to the dense regime of the Generalized Gaussian family. By introducing a principled inductive bias toward sparsity and non-negativity, Rectified LpJEPA jointly enforces invariance and enables controllable sparsity while retaining task-relevant information.

We summarize our contributions as follows:

1. **Rectified Generalized Gaussian Distributions**. We introduce the Rectified Generalized Gaussian (RGG) distribution, relate its truncated continuous component to maximum-entropy distributions under expected $\ell_p$ norm and support constraints, and show how rectification induces explicit $\ell_0$ sparsity control.

2. **Rectified LpJEPA with RDMReg**. We propose Rectified LpJEPA, a novel JEPA architecture equipped with Rectified Distribution Matching Regularization (RDM-

Reg), enabling controllable sparsity and non-negativity in learned representations.

3. **Empirical Validation**. We empirically demonstrate that Rectified LpJEPA achieves controllable sparsity, favorable sparsity–performance trade-offs, improved statistical independence, and competitive downstream accuracy across image classification benchmarks.

## 2. Background

In this section, we review key notions of sparsity. Additional background can be found in Appendix A.

**Sparsity**. Beyond its role in robust recovery and compressed sensing (Mallat, 1999; Donoho, 2006), sparsity has long been argued to be a fundamental organizing principle of efficient information processing in human and animal intelligence (Barlow et al., 1961). In sensory neuroscience, extensive empirical evidences suggest that neural systems encode dense and high-dimensional sensory inputs into non-negative, sparse activations under strict metabolic and signaling constraints (Olshausen & Field, 1996; Attwell & Laughlin, 2001).

In signal processing, sparse coding seeks to reconstruct signals using a minimal number of active components, typically enforced through $\ell_1$ regularization (Chen et al., 2001). Complementarily, non-negative matrix factorization enforces non-negativity by restricting representations to the positive orthant, inducing a conic geometry that yields parts-based and interpretable decompositions (Lee & Seung, 1999).

In deep learning, rectifying nonlinearities such as ReLU enforce non-negativity by zeroing negative responses, inducing support sparsity akin to $\ell_0$ constraints and underpinning the success of modern deep networks (Nair & Hinton, 2010; Glorot et al., 2011).

**Metrics of Sparsity.** To quantify sparsity, we consider a vector $\mathbf{x} \in \mathbb{R}^d$. The $\ell_0$ (pseudo-)norm $\|\mathbf{x}\|_0 := \sum_{i=1}^{d} \mathbb{1}_{\mathbb{R} \setminus \{0\}}(\mathbf{x}_i)$ counts the number of nonzero elements in $\mathbf{x}$, where $\mathbb{1}_S(x)$ is the indicator function that evaluates to 1 if $x \in S$ and 0 otherwise. Direct minimization of $\ell_0$ norm is however an NP-hard problem (Natarajan, 1995). A standard relaxation is the $\ell_1$ norm, $\|\mathbf{x}\|_1 := \sum_{i=1}^{d} |\mathbf{x}_i|$, which is the tightest convex envelope of $\ell_0$ on bounded domains and enables tractable optimization (Tibshirani, 1996).

More generally, $\ell_p$ quasi-norms $\|\mathbf{x}\|_p^p := \sum_{i=1}^{d} |\mathbf{x}_i|^p$ with $0 < p < 1$ provide a closer, nonconvex approximation to $\ell_0$: their singular behavior near zero strongly favors exact sparsity while exerting weaker penalties on large-magnitude components. Although nonconvexity complicates optimization, such penalties have been shown to yield sparser and less biased solutions than $\ell_1$ under suitable conditions (Chartrand, 2007; Chartrand & Yin, 2008).

## 3. Sparse and Maximum-Entropy Distributions

In the following section, we show that the proposed Rectified Generalized Gaussian distribution is a rectified mixture built from maximum-entropy truncated Generalized Gaussian components under $\ell_p$ constraints, yielding a target family that combines high-entropy continuous components with explicit $\ell_0$ sparsity control. We first introduce the Generalized Gaussian distribution (Section 3.1) and its truncated variant (Section 3.2), and show that they are the maximum-entropy distributions under an expected $\ell_p$ norm constraint (Section 3.3). We then show that incorporating rectification yields the Rectified Generalized Gaussian distribution (Section 3.4) with an entropy characterization using Rényi information dimension and an analytical guanratee of $\ell_0$ sparsity (Section 3.5).

### 3.1. Generalized Gaussian Distributions

In Definition 3.1, we present the standard form of the Generalized Gaussian Distribution (Subbotin, 1923; Goodman & Kotz, 1973; Nadarajah, 2005).

**Definition 3.1** (Generalized Gaussian Distribution ). The Generalized Gaussian distribution $\mathcal{GN}_p(\mu, \sigma)$ over the support $(-\infty, \infty)$ has the probability density function

$$f_{\mathcal{GN}_p(\mu,\sigma)}(x) = \frac{p^{1-1/p}}{2\sigma\Gamma(1/p)} \exp\left( - \frac{|x - \mu|^p}{p\sigma^p} \right) \quad (1)$$

where $\Gamma(s) := \int_0^\infty t^{s-1} e^{-t} dt$ is the gamma function.

We observe that $\mathcal{GN}_p(\mu, \sigma)$ reduces to the Laplace distribution when $p = 1$ and the Gaussian distribution for $p = 2$.

### 3.2. Truncated Generalized Gaussian Distributions

If we restrict the support, we obtain the Truncated Generalized Gaussian Distributions in Definition 3.2.

**Definition 3.2** (Truncated Generalized Gaussian Distribution). Let $S \subseteq \mathbb{R}$ be a subset of $\mathbb{R}$ with positive Lebesgue measure. The Truncated Generalized Gaussian distribution $\mathcal{TGN}_p(\mu, \sigma, S)$ is the restriction of the Generalized Gaussian distribution $\mathcal{GN}_p(\mu, \sigma)$ to the support $S$. The probability density function of $\mathcal{TGN}_p(\mu, \sigma, S)$ is given by

$$f_{\mathcal{TGN}_p(\mu,\sigma,S)}(x) = \frac{\mathbb{1}_S(x)}{Z_S(\mu,\sigma,p)} \exp\left( - \frac{|x - \mu|^p}{p\sigma^p} \right) \quad (2)$$

where $\mathbb{1}_S(x)$ is the indicator function that evaluates to 1 if $x \in S$ and 0 otherwise. The partition function is

$$Z_S(\mu, \sigma, p) = \int_S \exp\left( - \frac{|x - \mu|^p}{p\sigma^p} \right) d\mathbf{x} \quad (3)$$

When $S = \mathbb{R}$, $\mathcal{TGN}_p(\mu, \sigma)$ is equivalent to $\mathcal{GN}_p(\mu, \sigma)$.

### 3.3. Maximum Entropy under $\ell_p$ Constraints

We consider the multivariate generalization (Goodman & Kotz, 1973) as the joint distribution resulting from the product measure of independent and identically distributed (i.i.d.) Truncated Generalized Gaussian random variables, i.e. $\mathbf{x} \sim \prod_{i=1}^{d} \mathcal{TGN}_p(\mu, \sigma, S)$ where $\mathbf{x} = (x_1, \ldots, x_d)$ for each $x_i \sim \mathcal{TGN}_p(\mu, \sigma, S)$. For our purposes, we only need $S = (0, \infty)$ and thus the product support is $(0, \infty)^d$.

In Proposition 3.3, we show that the zero-mean Multivariate Truncated Generalized Gaussian Distribution is in fact the maximum differential entropy distribution under the expected $\ell_p$ norm constraints.

**Proposition 3.3** (Maximum Entropy Characterizations of Multivariate Truncated Generalized Gaussian Distributions)**.** *The maximum entropy distribution over $S \subseteq \mathbb{R}^d$, where $S$ is a subset of $\mathbb{R}^d$ with positive Lebesgue measure, under the constraints*

$$\int_S p(\mathbf{x})d\mathbf{x} = 1, \quad \mathbb{E}[\|\mathbf{x}\|_p^p] = \frac{d}{d\lambda_1} \log Z_S(\lambda_1) \quad (4)$$

*is the Multivariate Truncated Generalized Gaussian distribution $\prod_{i=1}^{d} \mathcal{TGN}_p(0, \sigma, S)$ with probability density function*

$$p(x) = \frac{1}{Z_S(\lambda_1)} \exp\left( - \frac{\|\mathbf{x}\|_p^p}{p\sigma^p} \right) \cdot \mathbb{1}_S(\mathbf{x}) \quad (5)$$

*where $\lambda_1 = -1/p\sigma^p$ and $Z_S(\lambda_1)$ is the partition function.*

*Proof.* See Appendix E.2. □

In fact, if $S = \mathbb{R}^d$, we show in Corollary E.2 that $\mathbb{E}[\|\mathbf{x}\|_p^p] = d\sigma^p$. An immediate consequence of Proposition 3.3 is the well-known fact that Truncated Laplace and Truncated Gaussian over the same support set $S$ are maximal entropy under the expected $\ell_1$ and $\ell_2$ norm constraints respectively. For any $0 < p < 1$, this proposition still holds true and thus we obtain a continuous spectrum of sparse distributions.

For the rest of the paper, we will always assume product support for multivariate generalizations, but we note that Proposition 3.3 applies more generally to joint support.

### 3.4. Rectified Generalized Gaussian Distributions

In Definition 3.4, we introduce the Rectified Generalized Gaussian (RGG) distribution.

**Definition 3.4** (Rectified Generalized Gaussian)**.** The Rectified Generalized Gaussian distribution $\mathcal{RGN}_p(\mu, \sigma)$ is a mixture between a discrete Dirac measure $\delta_0(x)$ (Definition B.4) and a Truncated Generalized Gaussian distribution

$\mathcal{TGN}_p(\mu, \sigma, (0, \infty))$ with probability density function

$$f_{\mathcal{RGN}_p(\mu, \sigma)}(x) = \Phi_{\mathcal{GN}_p(0,1)}\left( - \frac{\mu}{\sigma} \right) \cdot \mathbb{1}_{\{0\}}(x) \quad (6)$$

$$+ \frac{p^{1-1/p}}{2\sigma\Gamma(1/p)} \exp\left( - \frac{|x - \mu|^p}{p\sigma^p} \right) \cdot \mathbb{1}_{(0,\infty)}(x) \quad (7)$$

where $\Phi_{\mathcal{GN}_p(0,1)}$ is the cumulative distribution function for the standard Generalized Gaussian distribution $\mathcal{GN}_p(0, 1)$.

In Appendices B and C, we present additional technical details of the Rectified Generalized Gaussian distribution. We also visualizes the connections between Truncated Generalized Gaussian and Rectified Generalized Gaussian distributions in Figure 2. For $p = 2$, we recover the Rectified Gaussian distribution (Socci et al., 1997; Anderson et al., 1997). To the best of our knowledge, our extension and application of the Generalized Gaussian distribution to its rectified variant is novel for $p \neq 2$.

Nardon & Pianca (2009) proposed the simulation technique for Generalized Gaussian random variables. In Algorithm 1, we show how to sample from the Rectified Generalized Gaussian distribution $\mathcal{RGN}_p(\mu, \sigma)$. Essentially, we only need to first sample from the Generalized Gaussian distribution, and then rectify. In other words, $x \sim \text{ReLU}(\mathcal{GN}_p(\mu, \sigma))$ is equivalent to $x \sim \mathcal{RGN}_p(\mu, \sigma)$.

In Proposition 3.5, we show the expected $\ell_0$ norm of the Multivariate Rectified Generalized Gaussian distribution is determined by the parameters $\{\mu, \sigma, p\}$.

### 3.5. Sparsity and Entropy

**Proposition 3.5** (Sparsity)**.** *Let $\mathbf{x} \sim \prod_{i=1}^{d} \mathcal{RGN}_p(\mu, \sigma)$ in $d$ dimension. Then*

$$\mathbb{E}[\|\mathbf{x}\|_0] = d \cdot \Phi_{\mathcal{GN}_p(0,1)}\left( \frac{\mu}{\sigma} \right) \quad (8)$$

$$= \frac{d}{2}\left( 1 + \text{sgn}\left( \frac{\mu}{\sigma} \right) P\left( \frac{1}{p}, \frac{|\mu/\sigma|^p}{p} \right) \right) \quad (9)$$

*where $\text{sgn}(\cdot)$ is the sign function and $P(\cdot, \cdot)$ is the lower regularized gamma function.*

*Proof.* See Appendix C.4. □

Due to explicit rectifications, the RGG family is absolutely continuous with respect to the mixture between the Dirac and Lebesgue measure (Lemma B.6), rendering differential entropy ill-defined. Thus we resort to the concept of $d(\boldsymbol{\xi})$-dimensional entropy by (Rényi, 1959), which measures the Shannon entropy of quantized random vector under successive grid refinement. In Theorem 3.6, we provide a $d(\boldsymbol{\xi})$-dimensional entropy characterization of Rectified

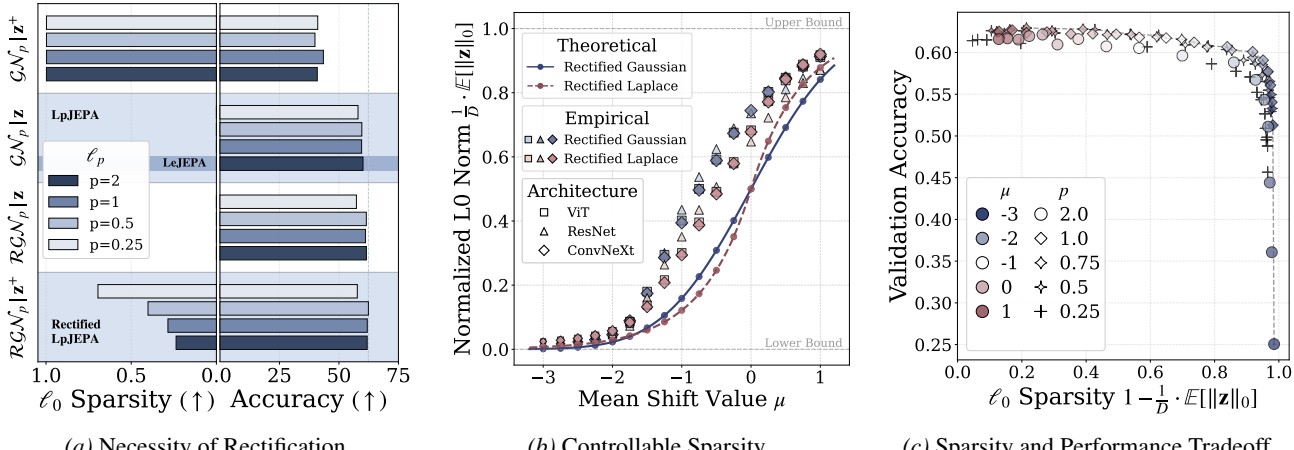

*Figure 3.* **Rectified LpJEPA achieves controllable sparsity and favorable sparsity-performance tradeoffs under proper parameterizations.** (a) We report CIFAR-100 validation accuracy and the $\ell_0$ sparsity metric $1 - (1/d) \cdot \mathbb{E}[\|\mathbf{x}\|_0]$ for four settings where we match non-rectified features $\mathbf{z}$ or rectified features $\mathbf{z}^+ := \mathrm{ReLU}(\mathbf{z})$ to either Rectified Generalized Gaussian $\mathcal{RGN}_p$ or conventional Generalized Gaussian $\mathcal{GN}_p$. Rectified LpJEPA ($\mathcal{RGN}_p \mid \mathbf{z}^+$) achieves the best sparsity-performance tradeoffs compared to other settings. (b) We compare the normalized $\ell_0$ norm of pretrained Rectified LpJEPA features against the theoretical predictions of Proposition 3.5 as $\mu$ varies. Empirical sparsity closely follows the predicted behavior across different values of $\mu$ and $p$. (c) We plot the Pareto frontier of sparsity versus accuracy across varying values of $\mu$ and $p$. Performance drops sharply only when more than $\sim 95\%$ of entries are zero.

Generalized Gaussian, where $d(\boldsymbol{\xi})$ is the Rényi information dimension. We defer additional details on the Rényi information dimension to Appendix F.

**Theorem 3.6** (Rényi Information Dimension Characterizations of Multivariate Rectified Generalized Gaussian Distributions). *Let $\boldsymbol{\xi} \sim \prod_{i=1}^{D} \mathcal{RGN}_p(\mu, \sigma)$ be a Rectified Generalized Gaussian random vector. The Rényi information dimension of $\boldsymbol{\xi}$ is $d(\boldsymbol{\xi}) = D \cdot \Phi_{\mathcal{GN}_p(0,1)}(\mu/\sigma)$, and the $d(\boldsymbol{\xi})$-dimensional entropy of $\boldsymbol{\xi}$ is given by*

$$\mathbb{H}_{d(\boldsymbol{\xi}_i)}(\boldsymbol{\xi}_i) = \Phi_{\mathcal{GN}_p(0,1)}\left(\frac{\mu}{\sigma}\right) \cdot \mathbb{H}_1(\mathcal{TGN}_p(\mu, \sigma)) \quad (10)$$

$$+ \; \mathbb{H}_0(\mathbb{1}_{(0,\infty)}(\boldsymbol{\xi}_i)) \quad (11)$$

$$\mathbb{H}_{d(\boldsymbol{\xi})}(\boldsymbol{\xi}) = \sum_{i=1}^{D} \mathbb{H}_{d(\boldsymbol{\xi}_i)}(\boldsymbol{\xi}_i) = D \cdot \mathbb{H}_{d(\boldsymbol{\xi}_i)}(\boldsymbol{\xi}_i) \quad (12)$$

*where $\mathbb{H}_0(\cdot)$ is the discrete Shannon entropy, $\mathbb{H}_1(\cdot)$ denotes the differential entropy, and $\mathbb{1}_{(0,\infty)}(\boldsymbol{\xi}_i)$ is a Bernoulli random variable that equals 1 with probability $\Phi_{\mathcal{GN}_p(0,1)}(\mu/\sigma)$ and 0 with probability $1 - \Phi_{\mathcal{GN}_p(0,1)}(\mu/\sigma)$.*

*Proof.* See Appendix F.2. $\square$

Thus we have shown that rectifications still preserve the maximal entropy property of the original distribution up to rescaling by the Rényi information dimension and constant offsets. In Lemma F.6, we further shows that the $d(\boldsymbol{\xi})$-dimensional entropy coincides with differential entropy under change of measure, enabling the interpretation of entropy under a Dirac and Lebesgue mixed measure.

## 4. Rectified LpJEPA

In the following section, we present a distributional regularization method based on the Cramér–Wold device (Section 4.1) for matching feature distributions towards Rectified Generalized Gaussian targets, resulting in Rectified LpJEPA with Rectified Distribution Matching Regularization (RDMReg) (Section 4.2). Contrary to isotropic Gaussian distributions, Rectified Generalized Gaussian distributions are not closed under linear combinations, leading to the necessity of two-sample sliced distribution matching (Section 4.3). We further demonstrate that RDMReg recovers a form of Non-Negative VCReg, which we defined in Section 4.4. Finally, we discuss various design choices of the target distribution with the parameter sets $(\mu, \sigma, p)$ which balance between sparsity and maximum-entropy (Section 4.5).

### 4.1. Cramér–Wold Based Distribution Matching

The Cramér–Wold device states that two random vectors $\mathbf{x}, \mathbf{y} \in \mathbb{R}^d$ are equal in distribution, i.e. $\mathbf{x} \stackrel{\mathrm{d}}{=} \mathbf{y}$, if and only if all their one-dimensional linear projections are equal in distribution (Cramér, 1936; Wold, 1938)

$$\mathbf{x} \stackrel{\mathrm{d}}{=} \mathbf{y} \iff \mathbf{c}^\top \mathbf{x} \stackrel{\mathrm{d}}{=} \mathbf{c}^\top \mathbf{y} \text{ for all } \mathbf{c} \in \mathbb{R}^d \quad (13)$$

This result enables us to decompose a high-dimensional distribution matching problem into parallelized one-dimension optimizations, which significantly reduces the sample complexity in each of the one-dimensional problems.

## 4.2. Rectified LpJEPA with RDMReg

Let $(\mathbf{x}, \mathbf{x}') \sim \mathbb{P}_{\mathbf{x}, \mathbf{x}'}$ denote a pair of random vectors jointly distributed according to a view-generating distribution $\mathbb{P}_{\mathbf{x}, \mathbf{x}'}$, where $\mathbf{x}$ and $\mathbf{x}'$ represent two stochastic views (e.g., random augmentations) of the same underlying input data. Let $f_{\boldsymbol{\theta}}$ be a neural network. We write $\mathbf{z} = \mathrm{ReLU}(f_{\boldsymbol{\theta}}(\mathbf{x}))$ and $\mathbf{z}' = \mathrm{ReLU}(f_{\boldsymbol{\theta}}(\mathbf{x}'))$, where $\mathbf{z}, \mathbf{z}' \in \mathbb{R}^D$ are the output feature random vectors. We further sample $\mathbf{y} \sim \prod_{i=1}^d \mathcal{RGN}_p(\mu, \sigma)$ and the random projection vectors $\mathbf{c}$ from the uniform distribution on the $\ell_2$ sphere, i.e. $\mathbf{c} \sim \mathrm{Unif}(\mathbb{S}_{\ell_2}^{d-1})$. We denote the induced distribution under projections as $\mathbb{P}_{\mathbf{c}^\top \mathbf{z}}$ and $\mathbb{P}_{\mathbf{c}^\top \mathbf{y}}$.

Our self-supervised learning objective consists of (i) an invariance term enforcing consistency across views, and (ii) a two-sample sliced distribution-matching loss which we called the Rectified Distribution Matching Regularization (RDMReg). The resulting loss takes the form

$$\min_{\boldsymbol{\theta}} \mathbb{E}_{\mathbf{z}, \mathbf{z}'}[\|\mathbf{z} - \mathbf{z}'\|_2^2] \tag{14}$$

$$+ \mathbb{E}_{\mathbf{c}}[\mathcal{L}(\mathbb{P}_{\mathbf{c}^\top \mathbf{z}} \| \mathbb{P}_{\mathbf{c}^\top \mathbf{y}})] + \mathbb{E}_{\mathbf{c}}[\mathcal{L}(\mathbb{P}_{\mathbf{c}^\top \mathbf{z}'} \| \mathbb{P}_{\mathbf{c}^\top \mathbf{y}})] \tag{15}$$

where $\mathcal{L}(P \| Q)$ is any loss function that minimizes the distance between two univariate distributions $P$ and $Q$. The expectation over projection vectors represents the population sliced objective; in practice we approximate it with a finite number of projections and empirical mini-batch samples.

## 4.3. The Necessity of Two-Sample Hypothesis Testing

Contrary to the isotropic Gaussian, which is closed under linear combinations, the Rectified Generalized Gaussian (RGG) family is not preserved under linear projections: the one-dimensional projected marginals generally fall outside the RGG family. In fact, closure under linear combinations characterizes the class of multivariate stable distributions (Nolan, 1993), which is disjoint from our RGG family. As illustrated in Figure 4, while any linear projection of a Gaussian remains Gaussian, projecting a Rectified Gaussian along different directions yields distinctly different marginals that no longer belong to the Rectified Gaussian family.

Consequently, the distribution matching loss $\mathcal{L}(\cdot \| \cdot)$ must rely on sample-based, nonparametric two-sample hypothesis tests on projected marginals (Lehmann & Romano, 1951; Gretton et al., 2012). Among many possible choices, we instantiate this objective using the sliced 2-Wasserstein distance (Bonneel et al., 2015; Kolouri et al., 2018) as it works well empirically. Let $\mathbf{Z}, \mathbf{Y} \in \mathbb{R}^{B \times D}$ be empirical neural network feature matrix and the samples from RGG where $B$ is batch size and $D$ is dimension. We denote a single random projection vector as $\mathbf{c}_i \in \mathbb{R}^D$ out of $N$ total projections.

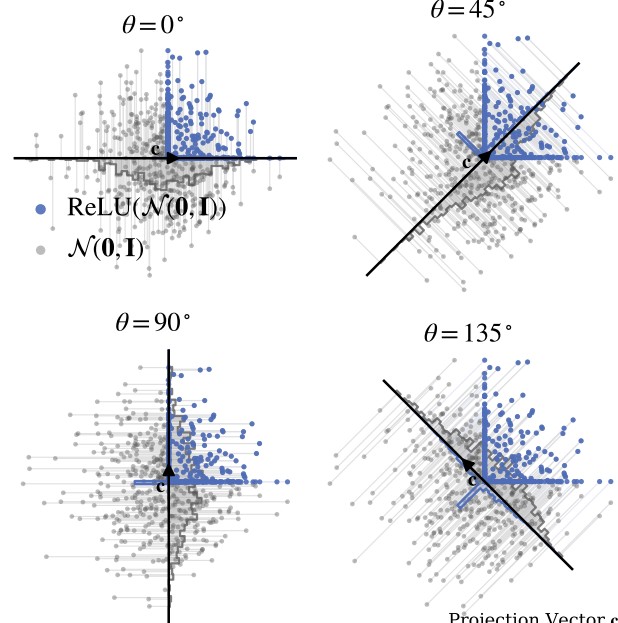

*Figure 4.* **Non-Closure under Projections.** Samples from 2-dimensional Gaussian $\mathcal{N}(\mathbf{0}, \mathbf{I})$ and Rectified Gaussian $\mathrm{ReLU}(\mathcal{N}(\mathbf{0}, \mathbf{I}))$ are drawn and projected along a certain direction $\mathbf{c}$. As opposed to Gaussian which is closed under linear combinations, the projected marginals of the Rectified Gaussian distribution no longer fall in the same family, motivating the necessity of using two-sample distribution-matching losses.

The RDMReg loss function is given by

$$\mathcal{L}(\mathbb{P}_{\mathbf{c}_i^\top \mathbf{z}} \| \mathbb{P}_{\mathbf{c}_i^\top \mathbf{y}}) := \frac{1}{B} \|(\mathbf{Z}\mathbf{c}_i)^{\uparrow} - (\mathbf{Y}\mathbf{c}_i)^{\uparrow}\|_2^2 \tag{16}$$

where $(\cdot)^{\uparrow}$ denotes sorting in ascending order. We additionally show in Figure 14c that a small, dimension-independent $N$ is sufficient to achieve strong empirical performance in our experiments, although finite-$N$ matching remains an approximation to the population Cramér–Wold criterion.

## 4.4. Connection to Non-Negative VCReg

In Appendix I, we show a conditional second-order connection between RDMReg and Non-Negative VCReg (Appendix H): if projected marginals match the RGG target along the eigenvectors of the feature covariance, then the centered feature covariance is isotropic. This is weaker than claiming that arbitrary finite random projections exactly recover VCReg, but it clarifies why eigenvector projections directly accelerate second-order dependency removal. We further show in Figure 15 that using eigenvectors of the empirical feature covariance matrices as projection vectors $\mathbf{c}_i$ leads to faster convergence toward optimal performance in our experiments.

### 4.5. Hyperparameters of the Target Distributions

Proposition 3.5 shows that the hyperparameter set $\{\mu, \sigma, p\}$ collectively determines the $\ell_0$ sparsity. $\sigma$ is a special parameter since we always want $\sigma > \epsilon$, where $\epsilon$ is some pre-specified threshold value, to prevent collapse.

We denote $\sigma_{\text{GN}} = \Gamma(1/p)^{1/2}/(p^{1/p} \cdot \Gamma(3/p)^{1/2})$ as the choice to ensure that the variance of the random variable before rectification is 1 since the closed form variance is readily available for the Generalized Gaussian distribution.

It's also possible to find $\sigma_{\text{RGN}}$ such that the variance after rectification is 1. In Proposition B.9, we derive the closed form expectation and variance of the Rectified Generalized Gaussian distribution. The choice of $\sigma_{\text{RGN}}$ can be determined by running a bisection search algorithm (see Algorithm 2) over the closed form variance formula. We defer additional comparisons between $\sigma_{\text{RGN}}$ and $\sigma_{\text{GN}}$ to Appendix D. Unless otherwise specified, we use $\sigma_{\text{GN}}$ as the default option.

## 5. Empirical Results

In the following sections, we introduce the basic settings and evaluations (Section 5.1). We establish our Rectified LpJEPA designs as the correct parameterizations to learn informative and sparse features compared to other possible alternatives (Sections 5.2 and 5.3). Rectified LpJEPA achieves controllable sparsity (Section 5.4) and favorable sparsity and performance tradeoffs (Section 5.5) with added benefits of learning more statistically independent (Section 5.6), high-entropy (Section 5.7) features, and performs competitively in pretraining and transfer evaluations (Section 5.8).

### 5.1. Experimental Settings

**Baselines**. We compare Rectified LpJEPA with dense baselines including SimCLR (denoted CL) (Chen et al., 2020) and VICReg (Bardes et al., 2022), as well as their sparse counterparts NCL (Wang et al., 2024) and Non-Negative VICReg (denoted NVICReg). We additionally compare against LpJEPA, which matches non-rectified features to Generalized Gaussian targets. Additional details for all baselines are provided in Appendix H.

**Sparsity Metrics**. We define the $\ell_1$ sparsity metric for a $D$-dimensional random vector $m_{\ell_1}(\mathbf{x}) = (1/D) \cdot \mathbb{E}[\|\mathbf{x}\|_1^2/\|\mathbf{x}\|_2^2]$, which attains its minimum value $1/D$ for extremely sparse vectors and its maximum value 1 for dense, uniformly distributed features. We additionally report the $\ell_0$ sparsity metric $m_{\ell_0}(\mathbf{x}) = (1/D) \cdot \mathbb{E}[\|\mathbf{x}\|_0]$, which measures the fraction of nonzero entries, with $m_{\ell_0} = 0$ indicating all-zero vectors and $m_{\ell_0} = 1$ indicating fully dense representations. In Figure 14b, we empirically observe strong correlations between $m_{\ell_1}$ and $m_{\ell_0}$ metrics. Sometimes we report $1 - m_{\ell_1}(\mathbf{x})$ or $1 - m_{\ell_0}(\mathbf{x})$ for visualization purposes.

**Backbones**. Following conventional practices in self-supervised learning (Balestriero et al., 2023), we adopt the encoder-projector design $\mathbf{z} = \text{ReLU}(f_{\boldsymbol{\theta}_2}(f_{\boldsymbol{\theta}_1}(\mathbf{x})))$ where $f_{\boldsymbol{\theta}_1}$ is a encoder like ResNet (He et al., 2016) or ViT (Dosovitskiy, 2020) and $f_{\boldsymbol{\theta}_2}$ is an additional multilayer perceptron. The Rectified LpJEPA loss is applied over $\mathbf{z}$ and linear probe evaluations are carried out on both $\mathbf{z}$ and $f_{\boldsymbol{\theta}_1}(\mathbf{x})$. We note that we add $\text{ReLU}(\cdot)$ at the end based on our design. The overall architecture is visualized in Figure 1.

### 5.2. Necessity of Rectifications

In Figure 3a, we report CIFAR-100 validation accuracy against the $\ell_0$ sparsity metric $1 - (1/D) \cdot \mathbb{E}[\|\mathbf{x}\|_0]$ under ablations that independently control rectification of the target distribution and the learned features. Corresponding results using $\ell_1$ sparsity are provided in Figure 12c. Without rectification, models achieve competitive accuracy but produce dense representations with no zero entries. When features are rectified, Rectified LpJEPA attains the best accuracy and sparsity tradeoff, whereas imposing an isotropic Gaussian distribution for rectified features leads to substantial performance drops.

### 5.3. Anti-Collapse via Continuous Mapping Theorems

By the continuous mapping theorem, convergence of $\mathbf{x} \in \mathbb{R}^d$ to a Generalized Gaussian implies that $\text{ReLU}(\mathbf{x})$ follows a Rectified Generalized Gaussian. In Figure 12b, we compare linear probe evaluations of Rectified LpJEPA versus LpJEPA features, where the linear probe is trained on pretrained LpJEPA after an additional rectification. We observe that performance drops sharply for the latter case, indicating that it's necessary to directly match to the Rectified Generalized Gaussian distribution.

### 5.4. Controllable Sparsity

Under the correct parameterizations of both the target distributions and the neural network features, we proceed to validate if we observe controllable sparsity in practice. Proposition 3.5 shows that the expected $\ell_0$ norm is collectively determined by the set of parameters $\{\mu, \sigma, p\}$. In Figure 3b, we show both the empirical $\ell_0$ norms measured over different pretrained backbones (ResNet (He et al., 2016), ViT (Dosovitskiy, 2020), ConvNext (Liu et al., 2022)) and the theoretical $\ell_0$ norm computed using Equation (9) as a function of varying $\mu$ and $p$ and the choice of $\sigma_{\text{GN}}$ mentioned in Section 4.5. We observe that across different mean shift values $\mu$ on the x-axis, the empirical $\ell_0$ closely tracks the theoretical predictions, and the theoretical ordering between $p$ in the expected $\ell_0$ norms is also preserved in the empirical results. We defer additional comparisons between $\sigma_{\text{GN}}$ and $\sigma_{\text{RGN}}$ and more choices of $p$ to Figure 10.

*Table 1.* **Linear Probe Results on ImageNet-100.** Acc1 (%) is higher-is-better (↑); sparsity is lower-is-better (↓). **Bold** denotes best and underline denotes second-best in each column (ties allowed).

| | | Encoder Acc1 ↑ | Projector Acc1 ↑ | L1 Sparsity ↓ | L0 Sparsity ↓ |
|---|---|---|---|---|---|
| **Rectified LpJEPA** | $\mathcal{RGN}_{1.0}(0, \sigma_{\text{GN}})$ | 84.72 | 80.40 | 0.2726 | 0.6940 |
| | $\mathcal{RGN}_{2.0}(0, \sigma_{\text{GN}})$ | **85.08** | 80.00 | 0.3412 | 0.7298 |
| | $\mathcal{RGN}_{1.0}(0.25, \sigma_{\text{GN}})$ | 84.98 | **80.76** | 0.3745 | 0.7437 |
| | $\mathcal{RGN}_{2.0}(1.0, \sigma_{\text{GN}})$ | **85.08** | 80.54 | 0.6278 | 0.8668 |
| | $\mathcal{RGN}_{2.0}(-2.5, \sigma_{\text{GN}})$ | 82.02 | 67.82 | 0.0137 | 0.0224 |
| | $\mathcal{RGN}_{1.0}(-3.0, \sigma_{\text{GN}})$ | 82.72 | 71.88 | 0.0058 | 0.0098 |
| **Sparse Baselines** | NVICReg-ReLU | 84.48 | 77.74 | 0.5207 | 0.7117 |
| | NCL-ReLU | 82.58 | 76.88 | 0.0037 | 0.0085 |
| | NVICReg-RepReLU | 84.20 | 78.18 | 0.4965 | 0.7549 |
| | NCL-RepReLU | 82.76 | 76.70 | **0.0024** | **0.0048** |
| **Dense Baselines** | VICReg | 84.18 | 78.88 | 0.7954 | 1.0000 |
| | SimCLR | 83.44 | 77.90 | 0.6338 | 1.0000 |
| | LeJEPA | 84.80 | 79.52 | 0.6365 | 1.0000 |

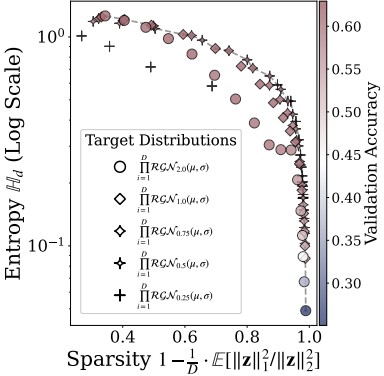

*(a) $d(\xi)$-dimensional Entropy*

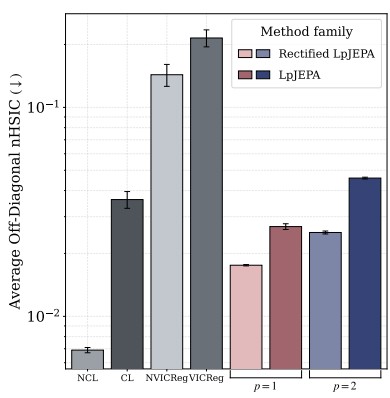

*(b) Hilbert-Schmidt independence Criterion (HSIC).*

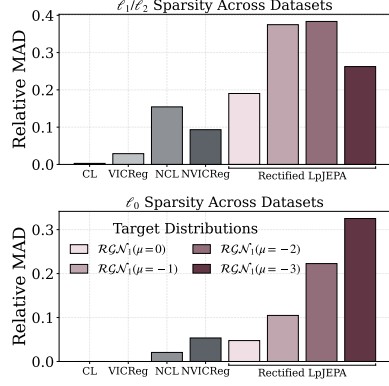

*(c) Dataset-Adaptive Sparsity*

*Figure 5.* **Rectified LpJEPA empirically achieves higher-entropy, more independent features with dataset-adaptive sparsity.** (a) The averaged univariate $d(\xi)$-dimensional entropy of the Rectified LpJEPA features are computed against the $\ell_1$ sparsity metric $1 - (1/D) \cdot \mathbb{E}[\|\mathbf{z}\|_1^2 / \|\mathbf{z}\|_2^2]$ across varying $\mu$ and $p$. Overall, we observe the expected behavior of sparsity-entropy tradeoff (b) We evaluate the normalized Hilbert-Schmidt independence Criterion (nHSIC) for LpJEPA, Rectified LpJEPA, and other baselines. Rectified LpJEPA achieves smaller nHSIC values compared to VICReg or NVICReg that only penalizes second-order statistics. (c) The relative mean absolute deviations (MAD) away from the median of the $\ell_1$ and $\ell_0$ sparsity metrics are computed over different methods. Rectified LpJEPA exhibits the highest variations of sparsity for different downstream dataset. Additional visualizations can be found in Figure 13.

## 5.5. Sparsity and Performance Tradeoff

With controllable sparsity at hand, we are interested in to what extent we can sparsify our features without performance drops. In Figure 3c, we plot the Pareto frontier of validation accuracy against the $\ell_0$ sparsity metrics $1 - (1/D) \cdot \mathbb{E}[\|\mathbf{x}\|_0]$ across varying $\mu$ and $p$ with the choice of $\sigma_{\text{GN}}$. We observe smooth and slow decay of performance as number of zeros in the feature representations increase, and the cliff-like drop in performance only occurs when roughly 95% of the entries are zero, indicating significant exploitable sparsity in our learned image representations. Additional visualizations are deferred to the Figure 14a.

## 5.6. Pair-wise Independence via HSIC

Beyond sparsity, we evaluate whether the learned representations form approximately independent, factorial encodings of the input data. A principled measure of dependence is the *total correlation*, defined as the KL divergence between the joint distribution and the product of its marginals. However, estimating total correlation is intractable in high-dimensional space (McAllester & Stratos, 2020). We therefore resort to the Hilbert–Schmidt Independence Criterion (HSIC) (Gretton et al., 2005) as a practical surrogate for detecting statistical dependence beyond second-order correlations captured by the covariance matrix.

In Figure 5b, we report the normalized HSIC values (see Appendix G for details) of Rectified LpJEPA and several dense and sparse baselines. Compared to methods such as VICReg and NVICReg, which explicitly regularize second-order statistics but do not constrain higher-order dependencies, Rectified LpJEPA consistently achieves lower nHSIC values, indicating representations that are closer to being statistically independent. Contrastive methods such as CL and NCL also attain low nHSIC scores; however, contrastive objectives are known to suffer from high sample complexity in high-dimensional representation spaces (Chen et al., 2020). Overall, these results suggest that RDMReg objectives encourage not only sparsity but also reduced higher-order dependence.

### 5.7. Rényi Information Dimension and Entropy

We would like to quantify whether the learned representations exhibit high entropy. However, due to rectification, the resulting feature distributions are not absolutely continuous with respect to the Lebesgue measure, rendering standard differential entropy ill-defined and obscuring whether the usual decomposition of total correlation into marginal and joint entropies remains valid. In Appendix F.5, we show that this decomposition continues to hold when entropy is defined in terms of the $d(\xi)$-dimensional entropy.

In Figure 5a, we report the sum of marginal $d(\xi)$-dimensional entropies as an upper bound on the joint entropy across a range of dense and sparse representations. The results reveal a clear Pareto frontier between entropy and sparsity. Moreover, since Rectified LpJEPA consistently attains lower nHSIC values than VICReg-style baselines, indicating reduced statistical dependence, the marginal entropy estimates for Rectified LpJEPA are expected to provide a tighter and more faithful approximation of the joint entropy.

### 5.8. Pretraining and Transfer Evaluations

In Table 1, we report linear probe results for Rectified LpJEPA pretrained on ImageNet100, compared against a range of dense and sparse baselines. Rectified LpJEPA consistently achieves a favorable trade-off between downstream accuracy and representation sparsity.

We further evaluate transfer performance under both few-shot and full-shot settings (see Tables 6 to 11). Across all configurations, Rectified LpJEPA achieves competitive accuracy, demonstrating strong transferability. In Figure 5c, we additionally observe that pretrained Rectified LpJEPA representations exhibit distinct sparsity patterns across multiple out-of-distribution datasets, suggesting that sparsity statistics can serve as a useful proxy for distinguishing in-distribution training data from OOD inputs. Additional ImageNet-1K, batch-size, runtime, and asymptotic-efficiency results can be seen in Appendices J.4 to J.7. We

also present additional nearest-neighbors retrieval and visual attribution maps in Appendix K.

## 6. Conclusion

We introduced Rectified LpJEPA, a JEPA model equipped with Rectified Distribution Matching Regularization (RDMReg) that induces sparse representations through distribution matching to the Rectified Generalized Gaussian distributions. By showing that sparsity can be achieved via target distribution design while preserving task-relevant information, our work opens new avenues for fundamental research on JEPA regularizers.

## Acknowledgements

We thank Deep Chakraborty and Nadav Timor for helpful discussions. This work was supported in part by AFOSR under grant FA95502310139, NSF Award 1922658, and Kevin Buehler's gift. This work was also supported through the NYU IT High Performance Computing resources, services, and staff expertise.

## Impact Statement

This paper presents work whose goal is to advance the field of Machine Learning. There are many potential societal consequences of our work, none which we feel must be specifically highlighted here.

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

# Appendix

## A. Additional Backgrounds

**Self-Supervised Learning**. Common self-supervised learning can be categorized into 1) contrastive methods (Chen et al., 2020; He et al., 2020), 2) non-contrastive methods (Zbontar et al., 2021; Bardes et al., 2022; Ermolov et al., 2021), 3) self-distillation methods (Grill et al., 2020; Caron et al., 2021; Chen & He, 2020) based on Balestriero et al. (2023). Along the line of statistical redundancy reductions, MCR$^2$ (Yu et al., 2020) regularizes the log determinant of the scaled empirical covariance matrix shifted by the identity matrix while MMCR (Yerxa et al., 2023) penalizes the nuclear norm of the centroid feature matrix. E2MC (Chakraborty et al., 2025) minimizes the sum of marginal entropies of the feature distribution on top of minimizing the VCReg loss (Bardes et al., 2022). Radial-VCReg (Kuang et al., 2025) and LeJEPA (Balestriero & LeCun, 2025) go beyond second-order dependencies by learning isotropic Gaussian features. Our Rectified LpJEPA also reduce higher-order dependencies by design, while enforcing sparsity over learned representations.

Prior work like Non-Negative Contrastive learning (NCL) (Wang et al., 2024) also aims to learn sparse features by optimizing contrastive losses over rectified features. Contrastive Sparse Representation (CSR) (Wen et al., 2025) develops a post-training sparsity adaptation method by learning a sparse auto-encoder (SAE) (Gao et al., 2024) over pretrained dense features using NCL loss, reconstruction loss, and a couple of SAE-specific auxiliary losses.

**Cramér–Wold Based Distribution Matching Losses**. Exemplars include sliced Wasserstein distances and their generative extensions (Bonneel et al., 2015; Kolouri et al., 2018), sliced kernel discrepancies (Nadjahi et al., 2020), projection-averaged multivariate tests (Kim et al., 2019), and more recently LeJEPA with SIGReg loss (Balestriero & LeCun, 2025), which also show that it suffices to sample $\mathbf{c} \in \mathbb{S}_{\ell_2}^{d-1} := \{\mathbf{c} \in \mathbb{R}^d \mid \|\mathbf{c}\|_2 = 1\}$.

## B. Properties of Univariate Generalized Gaussian, Truncated Generalized Gaussian, and Rectified Generalized Gaussian Distributions

In the following section, we present additional details on the Generalized Gaussian (Appendix B.1), Truncated Generalized Gaussian (Appendix B.2), and the Rectified Generalized Gaussian distributions (Appendix B.3). We also present the expectation and variance (Appendix B.4) and the sampling method (Appendix B.5) for the Rectified Generalized Gaussian distribution.

### B.1. Univariate Case - Generalized Gaussian

The Generalized Gaussian distribution $\mathcal{GN}_p(\mu, \sigma)$ (Subbotin, 1923; Goodman & Kotz, 1973; Nadarajah, 2005) has the probability density function given in Definition 3.1 with expectation and variance as

$$\mathbb{E}[x] = \mu \tag{B.1}$$

$$\mathrm{Var}[x] = \sigma^2 p^{2/p} \frac{\Gamma(3/p)}{\Gamma(1/p)} \tag{B.2}$$

The cumulative distribution function of $\mathcal{GN}_p(\mu, \sigma)$ is given by

$$\Phi_{\mathcal{GN}_p(\mu,\sigma)}(x) = \frac{1}{2} + \mathrm{sgn}(x - \mu) \frac{1}{2\Gamma(1/p)} \gamma\left(\frac{1}{p}, \frac{|x - \mu|^p}{p\sigma^p}\right) \tag{B.3}$$

where $\gamma(\cdot, \cdot)$ is the lower incomplete gamma function. We note that the probability density function of $\mathcal{GN}_p(\mu, \sigma)$ has other parameterizations (Remark B.1) and there are well-known special cases when $p = 1$ or $p = 2$ (Remark B.2).

*Remark* B.1. The probability density function of the Generalized Gaussian distribution can also be written as

$$f_{\mathcal{GN}_p(\mu,\sigma)}(x) = \frac{p}{2\alpha\Gamma(1/p)} \exp\left(-\frac{|x - \mu|^p}{\alpha^p}\right) \tag{B.4}$$

where $\alpha := p^{1/p}\sigma$. We choose the particular presentation in Definition 3.1 for its connection to the family of $L_p$-norm spherical distributions (Gupta & Song, 1997).

*Remark* B.2. When $p = 1$, the Generalized Gaussian distribution reduces to the Laplace distribution $\mathcal{L}(\mu, \sigma)$ with probability density function

$$f_{\mathcal{GN}_1(\mu,\sigma)}(x) = f_{\mathcal{L}(\mu,\sigma)}(x) = \frac{1}{2\sigma} \exp\left(-\frac{|x - \mu|}{\sigma}\right) \tag{B.5}$$

If $p = 2$, we recover the Gaussian distribution $\mathcal{N}(\mu, \sigma^2)$

$$f_{\mathcal{GN}_2(\mu,\sigma)}(x) = f_{\mathcal{N}(\mu,\sigma^2)}(x) = \frac{1}{\sigma\sqrt{2\pi}} \exp\left(-\frac{|x - \mu|^2}{2\sigma^2}\right) \tag{B.6}$$

For measure-theoretical characterizations of the Rectified Generalized Gaussian distribution in Appendix B.3, we denote the probability measure for $X \sim \mathcal{GN}_p(\mu, \sigma)$ as $\mathbb{P}_{\mathcal{GN}_p(\mu,\sigma)}$.

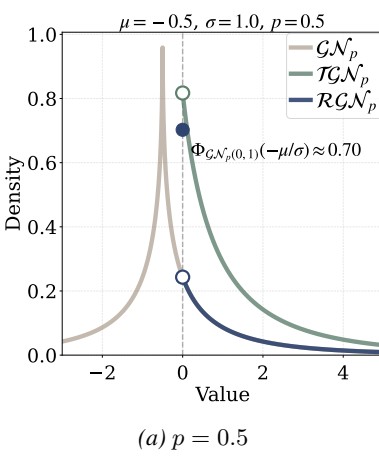 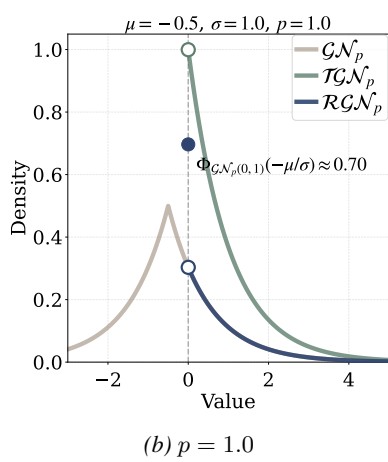 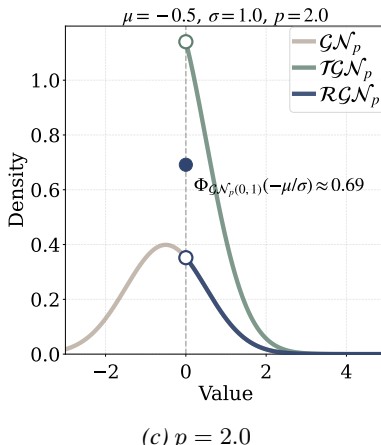

| *(a) $p = 0.5$* | *(b) $p = 1.0$* | *(c) $p = 2.0$* |

*Figure 6.* **The Probability Density Function of Generalized Gaussian $\mathcal{GN}_p$, Truncated Generalized Gaussian $\mathcal{TGN}_p$, and Rectified Generalized Gaussian $\mathcal{RGN}_p$ across varying $p$ with fixed $\mu = -0.5$ and $\sigma = 1$.** $\Phi_{\mathcal{GN}_p(0,1)}$ is the CDF of the Generalized Gaussian $\mathcal{GN}_p(0,1)$. (a) The case when $p = 0.5$. (b) When $p = 1$, we obtain Laplace, Truncated Laplace, and Rectified Laplace. (c) For $p = 2$, we have Gaussian, Truncated Gaussian, and Rectified Gaussian.

## B.2. Univariate Case - Truncated Generalized Gaussian

The Truncated Generalized Gaussian distribution is defined in Definition 3.2 in terms of the probability density function. In Definition B.3, we present the definition of the Truncated Generalized Gaussian probability measure.

**Definition B.3** (Truncated Generalized Gaussian Probability Measure). Let $X \sim \mathbb{P}_{\mathcal{GN}_p(\mu,\sigma)}$ be a Generalized Gaussian random variable with $\ell_p$ parameter $p > 0$, location $\mu \in \mathbb{R}$, and scale $\sigma > 0$. The Truncated Generalized Gaussian probability measure $\mathbb{P}_{\mathcal{TGN}_p(\mu,\sigma)}$ on the measurable space $(\mathbb{R}, \mathcal{B}(\mathbb{R}))$ is defined as the conditional distribution of $X$ given $X > 0$, i.e.,

$$\mathbb{P}_{\mathcal{TGN}_p(\mu,\sigma)}(A) := \mathbb{P}(X \in A \mid X > 0) = \frac{\mathbb{P}_{\mathcal{GN}_p(\mu,\sigma)}(A \cap (0,\infty))}{\mathbb{P}_{\mathcal{GN}_p(\mu,\sigma)}((0,\infty))} = \frac{\mathbb{P}_{\mathcal{GN}_p(\mu,\sigma)}(A \cap (0,\infty))}{1 - \Phi_{\mathcal{GN}_p(0,1)}(-\mu/\sigma)}$$

for any $A \in \mathcal{B}(\mathbb{R})$, where $\Phi_{\mathcal{GN}_p(0,1)}$ denotes the cumulative distribution function of the standardized Generalized Gaussian distribution.

## B.3. Univariate Case - Rectified Generalized Gaussian

In Definition 3.4, we provide a probability density function (PDF) characterization of the Rectified Generalized Gaussian distribution. However, we note that the PDF presented in Definition 3.4 is not the Radon–Nikodym derivative of the Rectified Generalized Gaussian probability measure with respect to the standard Lebesgue measure over $\mathbb{R}$, which we denote as $\lambda$. In Definition B.5, we provide a measure-theoretical treatment of the Rectified Generalized Gaussian distribution. We start by introducing the Dirac measure in Definition B.4.

**Definition B.4** (Dirac Measure). The Dirac measure $\delta_x$ over a measurable space $(X, \Sigma)$ for a given $x \in X$ is defined as

$$\delta_x(A) = \mathbb{1}_A(x) = \begin{cases} 0, x \notin A \\ 1, x \in A \end{cases} \tag{B.7}$$

for any measurable set $A \subseteq X$.

In Definition B.5, we formally introduce the Rectified Generalized Gaussian probability measure and its probability density function.

**Definition B.5** (Measure-Theoretical Definition of the Rectified Generalized Gaussian). Fix parameters $p > 0$, $\mu \in \mathbb{R}$, and $\sigma > 0$. We denote $(\mathbb{R}, \mathcal{B}(\mathbb{R}))$ as the real line equipped with Borel $\sigma$-algebra. Let $\lambda$ be the Lebesgue measure on $\mathcal{B}(\mathbb{R})$ and let $\delta_0$ be the Dirac measure at 0 presented in Definition B.4. The probability measure $\mathbb{P}_X$ of the Rectified Generalized Gaussian random variable $X$ is given by the mixture

$$\mathbb{P}_X = \Phi_{\mathcal{GN}_p(0,1)}\left(-\frac{\mu}{\sigma}\right) \cdot \delta_0 + \left(1 - \Phi_{\mathcal{GN}_p(0,1)}\left(-\frac{\mu}{\sigma}\right)\right) \cdot \mathbb{P}_{\mathcal{TGN}_p(\mu,\sigma)} \tag{B.8}$$

where $\mathbb{P}_{\mathcal{TGN}_p(\mu,\sigma)}$ is the Truncated Generalized Gaussian probability measure in Definition B.3 and $\Phi_{\mathcal{GN}_p(0,1)}$ is the CDF of the standard Generalized Gaussian $\mathcal{GN}_p(0,1)$. Define the mixed measure $\nu := \lambda + \delta_0$. By Lemma B.7, the Radon-Nikodym derivative of $\mathbb{P}_X$ with respect to $\nu$ exists and is given by

$$\frac{d\mathbb{P}_X}{d\nu}(x) = f_{\mathcal{RGN}_p(\mu,\sigma)}(x) = \Phi_{\mathcal{GN}_p(0,1)}\left(-\frac{\mu}{\sigma}\right) \cdot \mathbb{1}_{\{0\}}(x) + \frac{p^{1-1/p}}{2\sigma\Gamma(1/p)} \exp\left(-\frac{|x-\mu|^p}{p\sigma^p}\right) \cdot \mathbb{1}_{(0,\infty)}(x) \tag{B.9}$$

**Lemma B.6** (Absolute Continuity). *The Rectified Generalized Gaussian probability measure $\mathbb{P}_X$ in Definition B.5 is absolutely continuous with respect to the mixed measure $\nu := \delta_0 + \lambda$, i.e. $\mathbb{P}_X \ll \nu$.*

*Proof.* According to Folland (1999), if $\mathbb{P}_X$ is a signed measure and $\nu$ is a positive measure on the same measurable space $(\mathbb{R}, \mathcal{B}(\mathbb{R}))$, then $\mathbb{P}_X \ll \nu$ if $\nu(A) = 0$ for every $A \in \mathcal{B}(\mathbb{R})$ implies $\mathbb{P}_X(A) = 0$.

Let's consider the case of $\nu(A) = 0$. By definition, $\nu(A) = \delta_0(A) + \lambda(A) = 0$. Since both $\delta_0$ and $\lambda$ are non-negative measures, $\delta_0(A) = \lambda(A) = 0$. We observe that $\delta_0(A) = 0$ implies $0 \notin A$ by the definition of the Dirac measure. Thus

$$\mathbb{P}_X(A) = \Phi_{\mathcal{GN}_p(0,1)}\left(-\frac{\mu}{\sigma}\right) \cdot \delta_0(A) + \left(1 - \Phi_{\mathcal{GN}_p(0,1)}\left(-\frac{\mu}{\sigma}\right)\right) \cdot \mathbb{P}_{\mathcal{TGN}_p(\mu,\sigma)}(A) \tag{B.10}$$

$$= \left(1 - \Phi_{\mathcal{GN}_p(0,1)}\left(-\frac{\mu}{\sigma}\right)\right) \cdot \mathbb{P}_{\mathcal{TGN}_p(\mu,\sigma)}(A) \tag{B.11}$$

where the first term vanishes because $0 \notin A$. It's trivial that $\mathbb{P}_{\mathcal{TGN}_p(\mu,\sigma)}$ is absolutely continuous with respect to the Lebesgue measure. Since $\nu(A) = 0 \implies \lambda(A) = 0$, we have $\lambda(A) = 0 \implies \mathbb{P}_{\mathcal{TGN}_p(\mu,\sigma)}(A) = 0$. Thus

$$\mathbb{P}_X(A) = \left(1 - \Phi_{\mathcal{GN}_p(0,1)}\left(-\frac{\mu}{\sigma}\right)\right) \cdot \mathbb{P}_{\mathcal{TGN}_p(\mu,\sigma)}(A) = 0 \tag{B.12}$$

and we have proven the absolutely continuity result $\mathbb{P}_X \ll \nu$. $\square$

**Lemma B.7** (Radon–Nikodym Derivative). *The Radon–Nikodym derivative of the Rectified Generalized Gaussian probability measure $\mathbb{P}_X$ with respect to the mixed measure $\nu := \delta_0 + \lambda$ exists and is given by*

$$\frac{d\mathbb{P}_X}{d\nu}(x) = f_{\mathcal{RGN}_p(\mu,\sigma)}(x) = \Phi_{\mathcal{GN}_p(0,1)}\left(-\frac{\mu}{\sigma}\right) \cdot \mathbb{1}_{\{0\}}(x) + \frac{p^{1-1/p}}{2\sigma\Gamma(1/p)} \exp\left(-\frac{|x-\mu|^p}{p\sigma^p}\right) \cdot \mathbb{1}_{(0,\infty)}(x) \tag{B.13}$$

*Proof.* By Lemma B.6, $\mathbb{P}_X \ll \nu$ so the Radon–Nikodym derivative $d\mathbb{P}_X/d\nu$ exists and it suffices to show that for any $A \subseteq \mathcal{B}(\mathbb{R})$ we have

$$\mathbb{P}_X(A) = \int_A \frac{d\mathbb{P}_X}{d\nu} d\nu \tag{B.14}$$

We start by expanding the integral with respect to a sum of measures

$$\int_A \frac{d\mathbb{P}_X}{d\nu} d\nu = \int_A \frac{d\mathbb{P}_X}{d\nu} d\delta_0 + \int_A \frac{d\mathbb{P}_X}{d\nu} d\lambda \tag{B.15}$$

By the property of the Dirac measure, we have

$$\int_A \frac{d\mathbb{P}_X}{d\nu} d\delta_0 = \frac{d\mathbb{P}_X}{d\nu}(0)\delta_0(A) = f_{\mathcal{RGN}_p(\mu,\sigma)}(0)\delta_0(A) \tag{B.16}$$

We observe that $\mathbb{1}_{\{0\}}(x) = 1$ and $\mathbb{1}_{(0,\infty)}(0) = 0$. So we have

$$f_{\mathcal{RGN}_p(\mu,\sigma)}(0)\delta_0(A) = \Phi_{\mathcal{GN}_p(0,1)}\left(-\frac{\mu}{\sigma}\right) \cdot \delta_0(A) \tag{B.17}$$

Now the second term can be expanded as

$$\int_A \frac{d\mathbb{P}_X}{d\nu} d\lambda = \int_A \Phi_{\mathcal{GN}_p(0,1)}\left(-\frac{\mu}{\sigma}\right) \cdot \mathbb{1}_{\{0\}}(x)d\lambda(x) + \int_A \frac{p^{1-1/p}}{2\sigma\Gamma(1/p)} \exp\left(-\frac{|x-\mu|^p}{p\sigma^p}\right) \cdot \mathbb{1}_{(0,\infty)}(x)d\lambda(x) \tag{B.18}$$

$$= \Phi_{\mathcal{GN}_p(0,1)}\left(-\frac{\mu}{\sigma}\right) \cdot \int_A \mathbb{1}_{\{0\}}(x)d\lambda(x) + \int_A \frac{p^{1-1/p}}{2\sigma\Gamma(1/p)} \exp\left(-\frac{|x-\mu|^p}{p\sigma^p}\right) \cdot \mathbb{1}_{(0,\infty)}(x)d\lambda(x) \tag{B.19}$$

where the term

$$\int_A \mathbb{1}_{\{0\}}(x)d\lambda(x) = \lambda(A \cap \{0\}) = 0 \tag{B.20}$$

simply vanishes. Thus we are left with

$$\int_A \frac{d\mathbb{P}_X}{d\nu} d\lambda = \int_A \frac{p^{1-1/p}}{2\sigma\Gamma(1/p)} \exp\left(-\frac{|x-\mu|^p}{p\sigma^p}\right) \cdot \mathbb{1}_{(0,\infty)}(x) d\lambda(x) \tag{B.21}$$

$$= \int_{A\cap(0,\infty)} \frac{p^{1-1/p}}{2\sigma\Gamma(1/p)} \exp\left(-\frac{|x-\mu|^p}{p\sigma^p}\right) d\lambda(x) \tag{B.22}$$

$$= \int_{A\cap(0,\infty)} \frac{d\mathbb{P}_{\mathcal{GN}_p(\mu,\sigma)}}{d\lambda}(x) d\lambda(x) \tag{B.23}$$

$$= \mathbb{P}_{\mathcal{GN}_p(\mu,\sigma)}(A\cap(0,\infty)) \tag{B.24}$$

By Definition B.3, the Truncated Generalized Gaussian probability measure is given by

$$\mathbb{P}_{\mathcal{TGN}_p(\mu,\sigma)}(A) = \frac{\mathbb{P}_{\mathcal{GN}_p(\mu,\sigma)}(A\cap(0,\infty))}{\mathbb{P}_{\mathcal{GN}_p(\mu,\sigma)}((0,\infty))} \tag{B.25}$$

$$= \frac{\mathbb{P}_{\mathcal{GN}_p(\mu,\sigma)}(A\cap(0,\infty))}{1 - \Phi_{\mathcal{GN}_p(0,1)}\left(-\frac{\mu}{\sigma}\right)} \tag{B.26}$$

$$\tag{B.27}$$

Thus we have the identity

$$\mathbb{P}_{\mathcal{GN}_p(\mu,\sigma)}(A\cap(0,\infty)) = 1 - \Phi_{\mathcal{GN}_p(0,1)}\left(-\frac{\mu}{\sigma}\right) \cdot \mathbb{P}_{\mathcal{TGN}_p(\mu,\sigma)}(A) \tag{B.28}$$

Putting everything together, we arrive at

$$\int_A \frac{d\mathbb{P}_X}{d\nu} d\nu = \Phi_{\mathcal{GN}_p(0,1)}\left(-\frac{\mu}{\sigma}\right) \cdot \delta_0(A) + \left(1 - \Phi_{\mathcal{GN}_p(0,1)}\left(-\frac{\mu}{\sigma}\right)\right) \cdot \mathbb{P}_{\mathcal{TGN}_p(\mu,\sigma)}(A) \tag{B.29}$$

$$= \mathbb{P}_X(A) \tag{B.30}$$

Thus we have proven the form of the Radon–Nikodym Derivative. □

It's trivial to observe that the Rectified Generalized Gaussian probability measure is a valid probability measure since

$$\mathbb{P}_{\mathcal{RGN}_p(\mu,\sigma)}(\mathbb{R}) = \Phi_{\mathcal{GN}_p(0,1)}\left(-\frac{\mu}{\sigma}\right) \cdot \delta_0(\mathbb{R}) + \left(1 - \Phi_{\mathcal{GN}_p(0,1)}\left(-\frac{\mu}{\sigma}\right)\right) \cdot \mathbb{P}_{\mathcal{TGN}_p(\mu,\sigma)}(\mathbb{R}) \tag{B.31}$$

$$= \Phi_{\mathcal{GN}_p(0,1)}\left(-\frac{\mu}{\sigma}\right) + \left(1 - \Phi_{\mathcal{GN}_p(0,1)}\left(-\frac{\mu}{\sigma}\right)\right) = 1 \tag{B.32}$$

In Definition 3.4, we show that the Rectified Generalized Gaussian distribution can be presented as

$$f_{\mathcal{RGN}_p(\mu,\sigma)}(x) = \Phi_{\mathcal{GN}_p(0,1)}\left(-\frac{\mu}{\sigma}\right) \cdot \mathbb{1}_{\{0\}}(x) + \frac{p^{1-1/p}}{2\sigma\Gamma(1/p)} \exp\left(-\frac{|x-\mu|^p}{p\sigma^p}\right) \cdot \mathbb{1}_{(0,\infty)}(x) \tag{B.33}$$

At first glance, the second term is the probability density function of the Generalized Gaussian distribution instead of its truncated version. In Corollary B.8, we provide an alternative presentation of the Rectified Generalized Gaussian distribution with explicit components of the probability density function of the Truncated Generalized Gaussian distribution.

**Corollary B.8** (Equivalent Definition of Rectified Generalized Gaussian). *The probability density function of the Rectified Generalized Gaussian distribution $\mathcal{RGN}_p(\mu,\sigma)$ can also be written as*

$$f_{\mathcal{RGN}_p(\mu,\sigma)}(x) = \Phi_{\mathcal{GN}_p(0,1)}\left(-\frac{\mu}{\sigma}\right) \cdot \mathbb{1}_{\{0\}}(x) \tag{B.34}$$

$$+ \left(1 - \Phi_{\mathcal{GN}_p(0,1)}\left(-\frac{\mu}{\sigma}\right)\right) \frac{1}{Z_{(0,\infty)}(\mu,\sigma,p)} \exp\left(-\frac{|x-\mu|^p}{p\sigma^p}\right) \cdot \mathbb{1}_{(0,\infty)}(x) \tag{B.35}$$

*where $\Phi_{\mathcal{GN}_p(0,1)}$ is the cumulative distribution function for the standard Generalized Gaussian distribution $\mathcal{GN}_p(0,1)$.*

*Proof.* We can simplify the expression as

$$1 - \Phi_{\mathcal{GN}_p(0,1)}\left(-\frac{\mu}{\sigma}\right) = 1 - \Phi_{\mathcal{GN}_p(\mu,\sigma)}(0) = \Phi_{\mathcal{GN}_p(\mu,\sigma)}(0) = \int_{-\infty}^{0} \frac{p^{1-1/p}}{2\sigma\Gamma(1/p)} \exp\left(-\frac{|x-\mu|^p}{p\sigma^p}\right) dx \tag{B.36}$$

So we have

$$\left(1 - \Phi_{\mathcal{GN}_p(0,1)}\left(-\frac{\mu}{\sigma}\right)\right) \frac{1}{Z_{(0,\infty)}(\mu,\sigma,p)} = \frac{\int_{-\infty}^{0} \frac{p^{1-1/p}}{2\sigma\Gamma(1/p)} \exp\left(-\frac{|x-\mu|^p}{p\sigma^p}\right) dx}{\int_{0}^{\infty} \exp\left(-\frac{|x-\mu|^p}{p\sigma^p}\right) dx} = \frac{p^{1-1/p}}{2\sigma\Gamma(1/p)} \tag{B.37}$$

where the extra terms cancel out due to symmetry around 0. Thus we have recovered the forms in Definition 3.4. □

In Figure 6, we visualize the probability density of the Generalized Gaussian, Truncated Generalized Gaussian, and the Rectified Generalized Gaussian distributions across varying $p$.

## B.4. Expectation and Variance of the Rectified Generalized Gaussian Distribution

**Proposition B.9.** *Let $X \sim \mathcal{RGN}_p(\mu,\sigma)$ and $\operatorname{sgn}(\mu) \in \{-1,0,+1\}$ be the sign function. Let $\gamma(s,t)$ be the lower incomplete gamma function, $\Gamma(s,t)$ be the upper incomplete gamma function, $\Gamma(s)$ be the gamma function, and $P(s,t) = \gamma(s,t)/\Gamma(s)$ be the lower regularized gamma function. Then*

$$\mathbb{E}[X] = \frac{1}{2}\left[\mu\left(1 + \operatorname{sgn}(\mu)P\left(\frac{1}{p}, \frac{|\mu|^p}{p\sigma^p}\right)\right) + p^{1/p}\sigma\frac{\Gamma(2/p, |\mu|^p/(p\sigma^p))}{\Gamma(1/p)}\right] \tag{B.38}$$

$$\mathbb{E}[X^2] = \frac{1}{2}\left[\mu^2\left(1 + \operatorname{sgn}(\mu)P\left(\frac{1}{p}, \frac{|\mu|^p}{p\sigma^p}\right)\right) + 2\mu p^{1/p}\sigma\frac{\Gamma(2/p, |\mu|^p/(p\sigma^p))}{\Gamma(1/p)}\right. \tag{B.39}$$

$$\left. + p^{2/p}\sigma^2\frac{\Gamma(3/p)}{\Gamma(1/p)}\left(1 + \operatorname{sgn}(\mu)P\left(\frac{3}{p}, \frac{|\mu|^p}{p\sigma^p}\right)\right)\right] \tag{B.40}$$

$$\operatorname{Var}(X) = \mathbb{E}[X^2] - \left(\mathbb{E}[X]\right)^2. \tag{B.41}$$

*Proof.* Let $Z \sim \mathcal{GN}_p(\mu,\sigma)$ with density

$$f_Z(z) = \frac{p^{1-\frac{1}{p}}}{2\sigma\Gamma(1/p)} \exp\left(-\frac{|z-\mu|^p}{p\sigma^p}\right). \tag{B.42}$$

If $X = \operatorname{ReLU}(Z)$, then we know $X \sim \mathcal{RGN}_p(\mu,\sigma)$. Thus for any $k \in \{1,2\}$, we have

$$\mathbb{E}[X^k] = \mathbb{E}[Z^k \mathbb{1}_{(0,\infty)}(Z)] = \int_{0}^{\infty} z^k f_Z(z)\, dz. \tag{B.43}$$

To simplify notations, let's denote $C := p^{1-(1/p)}/(2\sigma\Gamma(1/p))$, $a := 1/(p\sigma^p)$, and $t_0 := a|\mu|^p = |\mu|^p/(p\sigma^p)$. Then

$$\mathbb{E}[X^k] = C \int_{0}^{\infty} z^k \exp\left(-a|z-\mu|^p\right) dz. \tag{B.44}$$

Define the change of variables $t = z - \mu$. Thus we have $z = t + \mu$ and $z \geq 0 \iff t \geq -\mu$. Rewrite the integral as

$$\mathbb{E}[X^k] = C \int_{-\mu}^{\infty} (t+\mu)^k \exp\left(-a|t|^p\right) dt. \tag{B.45}$$

Let's define the three auxiliary integrals

$$I_0 := \int_{-\mu}^{\infty} e^{-a|t|^p}\, dt \tag{B.46}$$

$$I_1 := \int_{-\mu}^{\infty} t\, e^{-a|t|^p}\, dt \tag{B.47}$$

$$I_2 := \int_{-\mu}^{\infty} t^2\, e^{-a|t|^p}\, dt. \tag{B.48}$$

Then we can rewrite (B.45) for $k = 1, 2$ as

$$\mathbb{E}[X] = C(\mu I_0 + I_1), \tag{B.49}$$

$$\mathbb{E}[X^2] = C(\mu^2 I_0 + 2\mu I_1 + I_2). \tag{B.50}$$

Now we just need to compute $I_0$, $I_1$, and $I_2$. By Lemma B.11, Lemma B.12, and Lemma B.13, we have

$$I_0 = \frac{1}{p} a^{-1/p} \Gamma\left(\frac{1}{p}\right) \left(1 + \operatorname{sgn}(\mu) P\left(\frac{1}{p}, t_0\right)\right) \tag{B.51}$$

$$I_1 = \frac{1}{p} a^{-2/p} \Gamma\left(\frac{2}{p}, t_0\right) \tag{B.52}$$

$$I_2 = \frac{1}{p} a^{-3/p} \Gamma\left(\frac{3}{p}\right) \left(1 + \operatorname{sgn}(\mu) P\left(\frac{3}{p}, t_0\right)\right). \tag{B.53}$$

So we can substitute and get the expression for $\mathbb{E}[X]$ as

$$\mathbb{E}[X] = C(\mu I_0 + I_1) \tag{B.54}$$

$$= C\mu \cdot \frac{1}{p} a^{-1/p} \Gamma\left(\frac{1}{p}\right) \left(1 + \operatorname{sgn}(\mu) P(1/p, t_0)\right) + C \cdot \frac{1}{p} a^{-2/p} \Gamma\left(\frac{2}{p}, t_0\right) \tag{B.55}$$

$$= \frac{1}{2}\mu\left(1 + \operatorname{sgn}(\mu) P\left(\frac{1}{p}, t_0\right)\right) + \frac{1}{2} p^{1/p} \sigma \frac{\Gamma(2/p, t_0)}{\Gamma(1/p)} \tag{B.56}$$

$$= \frac{1}{2}\left[\mu\left(1 + \operatorname{sgn}(\mu) P\left(\frac{1}{p}, \frac{|\mu|^p}{p\sigma^p}\right)\right) + p^{1/p} \sigma \frac{\Gamma(2/p, |\mu|^p/(p\sigma^p))}{\Gamma(1/p)}\right] \tag{B.57}$$

Similarly, the second moment is given by

$$\mathbb{E}[X^2] = C(\mu^2 I_0 + 2\mu I_1 + I_2) \tag{B.58}$$

$$= C\mu^2 I_0 + 2C\mu I_1 + C I_2 \tag{B.59}$$

$$= C\mu^2 \cdot \frac{1}{p} a^{-1/p} \Gamma\left(\frac{1}{p}\right) \left(1 + \operatorname{sgn}(\mu) P(1/p, t_0)\right) + 2C\mu \cdot \frac{1}{p} a^{-2/p} \Gamma\left(\frac{2}{p}, t_0\right) \tag{B.60}$$

$$+ C \cdot \frac{1}{p} a^{-3/p} \Gamma\left(\frac{3}{p}\right) \left(1 + \operatorname{sgn}(\mu) P(3/p, t_0)\right) \tag{B.61}$$

$$= \frac{1}{2}\mu^2\left(1 + \operatorname{sgn}(\mu) P\left(\frac{1}{p}, t_0\right)\right) + \frac{1}{2}\left(2\mu\, p^{1/p} \sigma \frac{\Gamma(2/p, t_0)}{\Gamma(1/p)}\right) \tag{B.62}$$

$$+ \frac{1}{2} p^{2/p} \sigma^2 \frac{\Gamma(3/p)}{\Gamma(1/p)} \left(1 + \operatorname{sgn}(\mu) P\left(\frac{3}{p}, t_0\right)\right) \tag{B.63}$$

$$= \frac{1}{2}\left[\mu^2\left(1 + \operatorname{sgn}(\mu)\, P\left(\frac{1}{p}, \frac{|\mu|^p}{p\sigma^p}\right)\right) + 2\mu p^{1/p} \sigma \frac{\Gamma(2/p, |\mu|^p/(p\sigma^p))}{\Gamma(1/p)}\right. \tag{B.64}$$

$$\left. + p^{2/p} \sigma^2 \frac{\Gamma(3/p)}{\Gamma(1/p)} \left(1 + \operatorname{sgn}(\mu)\, P\left(\frac{3}{p}, \frac{|\mu|^p}{p\sigma^p}\right)\right)\right], \tag{B.65}$$

$$\tag{B.66}$$

Thus we have proven the expression. □

**Definition B.10** (Gamma Functions). If $u \geq 0$ and $b > -1$, then

$$\int_0^u t^b e^{-at^p}\, dt = \frac{1}{p}\, a^{-(b+1)/p}\, \gamma\left(\frac{b+1}{p}, au^p\right), \tag{B.67}$$

$$\int_u^\infty t^b e^{-at^p}\, dt = \frac{1}{p}\, a^{-(b+1)/p}\, \Gamma\left(\frac{b+1}{p}, au^p\right), \tag{B.68}$$

where $\gamma(\cdot, \cdot)$ and $\Gamma(\cdot, \cdot)$ are the lower and upper incomplete gamma functions. By definition, we also have

$$P(s, t) := \frac{\gamma(s, t)}{\Gamma(s)}, \qquad \Gamma(s, t) = \Gamma(s) - \gamma(s, t) = \Gamma(s)\big(1 - P(s, t)\big). \tag{B.69}$$

**Lemma B.11** ($I_0$ Integral). *The $I_0$ integral in Proposition B.9 is given by*

$$I_0 = \frac{1}{p} a^{-1/p} \Gamma\left(\frac{1}{p}\right) \left(1 + \operatorname{sgn}(\mu)\, P\left(\frac{1}{p}, t_0\right)\right). \tag{B.70}$$

*Proof.* If $\mu \geq 0$, then $-\mu \leq 0$. So we can split the integral at 0 and get:

$$I_0 = \int_{-\mu}^{0} e^{-a|t|^p} \, dt + \int_{0}^{\infty} e^{-at^p} \, dt = \int_{0}^{\mu} e^{-as^p} \, ds + \int_{0}^{\infty} e^{-at^p} \, dt. \tag{B.71}$$

Applying (B.67) with $b = 0$ to the first term and (B.68) with $u = 0$ to the second term gives us

$$I_0 = \frac{1}{p} a^{-1/p} \gamma\left(\frac{1}{p}, t_0\right) + \frac{1}{p} a^{-1/p} \Gamma\left(\frac{1}{p}\right) = \frac{1}{p} a^{-1/p} \Gamma\left(\frac{1}{p}\right)\left(1 + P\left(\frac{1}{p}, t_0\right)\right). \tag{B.72}$$

Now if $\mu < 0$, then $-\mu = |\mu| > 0$ and we have

$$I_0 = \int_{|\mu|}^{\infty} e^{-at^p} \, dt = \frac{1}{p} a^{-1/p} \Gamma\left(\frac{1}{p}, t_0\right) = \frac{1}{p} a^{-1/p} \Gamma\left(\frac{1}{p}\right)\left(1 - P\left(\frac{1}{p}, t_0\right)\right). \tag{B.73}$$

Combining both cases, we arrive at

$$I_0 = \frac{1}{p} a^{-1/p} \Gamma\left(\frac{1}{p}\right)\left(1 + \mathrm{sgn}(\mu) \, P\left(\frac{1}{p}, t_0\right)\right). \tag{B.74}$$

$\square$

**Lemma B.12** ($I_1$ Integral)**.** *The $I_1$ integral in Proposition B.9 is given by*

$$I_1 = \frac{1}{p} a^{-2/p} \Gamma\left(\frac{2}{p}, t_0\right). \tag{B.75}$$

*Proof.* If $\mu \geq 0$, then $-\mu \leq 0$. So we can split the integral at 0 and get:

$$I_1 = \int_{-\mu}^{0} t e^{-a|t|^p} \, dt + \int_{0}^{\infty} t e^{-at^p} \, dt. \tag{B.76}$$

On $[-\mu, 0]$, we can substitute $s = -t$ to get

$$\int_{-\mu}^{0} t e^{-a|t|^p} \, dt = -\int_{0}^{\mu} s e^{-as^p} \, ds. \tag{B.77}$$

So we have

$$I_1 = -\int_{0}^{\mu} s e^{-as^p} \, ds + \int_{0}^{\infty} t e^{-at^p} \, dt = \int_{\mu}^{\infty} t e^{-at^p} \, dt. \tag{B.78}$$

Applying (B.68) with $b = 1$, we have

$$I_1 = \frac{1}{p} a^{-2/p} \Gamma\left(\frac{2}{p}, t_0\right). \tag{B.79}$$

Now if $\mu < 0$, then $-\mu = |\mu| > 0$ and we have

$$I_1 = \int_{|\mu|}^{\infty} t e^{-at^p} \, dt = \frac{1}{p} a^{-2/p} \Gamma\left(\frac{2}{p}, t_0\right). \tag{B.80}$$

Combining both cases, we arrive at

$$I_1 = \frac{1}{p} a^{-2/p} \Gamma\left(\frac{2}{p}, t_0\right). \tag{B.81}$$

$\square$

**Lemma B.13** ($I_2$ Integral)**.** *The $I_2$ integral in Proposition B.9 is given by*

$$I_2 = \frac{1}{p} a^{-3/p} \Gamma\left(\frac{3}{p}\right)\left(1 + \mathrm{sgn}(\mu) \, P\left(\frac{3}{p}, t_0\right)\right). \tag{B.82}$$

---

**Algorithm 1** Simulation of the Rectified Generalized Gaussian Random Variables $\mathcal{RGN}_p(\mu, \sigma)$

---

**Input:** $\ell_p$ parameter $p > 0$, location $\mu \in \mathbb{R}$, scale $\sigma > 0$
**Output:** sample $Y \sim \mathcal{RGN}_p(\mu, \sigma)$
Sample $S \sim \mathrm{Unif}\{-1, +1\}$
Sample $G \sim \mathrm{Gamma}\left(\mathrm{shape} = \frac{1}{p}, \mathrm{rate} = 1\right)$
Set $X \leftarrow \mu + \sigma S \cdot (pG)^{1/p}$
Set $Y \leftarrow \max(0, X)$
**return** $Y$

---

**Algorithm 2** Bisection Search for the Scale Parameter $\sigma$ for Rectified Generalized Gaussian with Unit Variance

---

**Input:** $\ell_p$ parameter $p > 0$, location $\mu \in \mathbb{R}$, tolerance $\varepsilon > 0$
**Output:** scale $\sigma^\star > 0$ such that $\mathrm{Var}(\mathcal{RGN}_p(\mu, \sigma^\star)) \approx 1$ {$\mathrm{Var}(\mathcal{RGN}_p(\mu, \sigma^\star))$ is defined in Proposition B.9.}
Define $V(\sigma) := \mathrm{Var}(\mathcal{RGN}_p(\mu, \sigma))$
Define $f(\sigma) := V(\sigma) - 1$
Choose initial bounds $\sigma_L > 0$ and $\sigma_U > \sigma_L$ such that $f(\sigma_L) < 0, f(\sigma_U) > 0$
**repeat**
    $\sigma_M \leftarrow (\sigma_L + \sigma_U)/2$
    **if** $f(\sigma_M) > 0$ **then**
        $\sigma_U \leftarrow \sigma_M$
    **else**
        $\sigma_L \leftarrow \sigma_M$
    **end if**
**until** $|\sigma_U - \sigma_L| \leq \varepsilon$
$\sigma^\star \leftarrow (\sigma_L + \sigma_U)/2$
**return** $\sigma^\star$

---

*Proof.* If $\mu \geq 0$, then $-\mu \leq 0$. So we can split the integral at 0 and get:

$$I_2 = \int_{-\mu}^{0} t^2 e^{-a|t|^p} \, dt + \int_{0}^{\infty} t^2 e^{-at^p} \, dt = \int_{0}^{\mu} s^2 e^{-as^p} \, ds + \int_{0}^{\infty} t^2 e^{-at^p} \, dt. \tag{B.83}$$

Apply (B.67) with $b = 2$ to the first term and the full gamma integral to the second term, we have:

$$I_2 = \frac{1}{p} a^{-3/p} \gamma\left(\frac{3}{p}, t_0\right) + \frac{1}{p} a^{-3/p} \Gamma\left(\frac{3}{p}\right) = \frac{1}{p} a^{-3/p} \Gamma\left(\frac{3}{p}\right)\left(1 + P\left(\frac{3}{p}, t_0\right)\right). \tag{B.84}$$

Now if $\mu < 0$, then $-\mu = |\mu| > 0$ and we have

$$I_2 = \int_{|\mu|}^{\infty} t^2 e^{-at^p} \, dt = \frac{1}{p} a^{-3/p} \Gamma\left(\frac{3}{p}, t_0\right) = \frac{1}{p} a^{-3/p} \Gamma\left(\frac{3}{p}\right)\left(1 - P\left(\frac{3}{p}, t_0\right)\right). \tag{B.85}$$

Combining both cases, we arrive at

$$I_2 = \frac{1}{p} a^{-3/p} \Gamma\left(\frac{3}{p}\right)\left(1 + \mathrm{sgn}(\mu)\, P\left(\frac{3}{p}, t_0\right)\right). \tag{B.86}$$

$\square$

## B.5. Simulation Techniques for Rectified Generalized Gaussian

In Algorithm 1, we show how to sample from the Rectified Generalized Gaussian distribution.

## C. Properties of Multivariate Generalized Gaussian, Truncated Generalized Gaussian, and Rectified Generalized Gaussian Distributions

In the following section, we present additional definitions and properties of the Multivariate Generalized Gaussian (Appendix C.1), Truncated Generalized Gaussian (Appendix C.2), and Rectified Generalized Gaussian distributions (Appendix C.3). We further derive the expected $\ell_0$ norm for a Multivariate Rectified Generalized Gaussian distribution in Appendix C.4.

## C.1. Multivariate Case - Multivariate Generalized Gaussian

We consider the multivariate generalization (Goodman & Kotz, 1973) as the joint distribution resulting from the product measure of independent and identically distributed (i.i.d.) Generalized Gaussian random variables, i.e. $\mathbf{x} \sim \prod_{i=1}^{d} \mathcal{GN}_p(\mu, \sigma)$ where $\mathbf{x} = (x_1, \ldots, x_d)$ for each $x_i \sim \mathcal{GN}_p(\mu, \sigma)$. The probability density function is given by

$$f_{\prod_{i=1}^{d} \mathcal{GN}_p(\mu,\sigma)}(\mathbf{x}) = \frac{p^{d-d/p}}{(2\sigma)^d \Gamma(1/p)^d} \exp\left( - \frac{\|\mathbf{x} - \boldsymbol{\mu}\|_p^p}{p\sigma^p} \right) \tag{C.87}$$

Assume that $\mu = 0$. Barthe et al. (2005) show that $r^p := \|\mathbf{x}\|_p^p \sim \Gamma(d/p, p\sigma^p)$ up to different notations. Also, $\mathbf{u} := \mathbf{x}/\|\mathbf{x}\|_p$ follows the cone measure on the $\ell_p$ sphere $\mathbb{S}_{\ell_p}^{d-1} := \{\mathbf{x} \in \mathbb{R}^d | \|\mathbf{x}\|_p = 1\}$. It's shown that $\mathbf{x} = r \cdot \mathbf{u}$ and $r \perp \mathbf{u}$ (Barthe et al., 2005). In fact, the cone measure is identical to the $(d-1)$-dimensional Hausdorff measure $\mathcal{H}^{d-1}$ (also called surface measure) when $p \in \{1, 2, \infty\}$ (Alonso-Gutierrez et al., 2018). So if $A \subseteq \mathbb{S}_{\ell_p}^{d-1}$, then $p(\mathbf{u} \in A) = \mathcal{H}^{d-1}(A)/\mathcal{H}^{d-1}(\mathbb{S}_{\ell_p}^{d-1})$.

Thus for zero-mean Laplace ($p = 1$) and zero-mean Gaussian ($p = 2$), the distributions of $\mathbf{u}$ are the uniform distributions on the simplex $\Delta^{d-1}$ (or $\mathbb{S}_{\ell_1}^{d-1}$) and the standard Euclidean unit sphere $\mathbb{S}_{\ell_2}^{d-1}$ respectively.

More generally, the Multivariate Generalized Gaussian distribution (Goodman & Kotz, 1973) is a special case of the family of $p$-symmetric distributions (Fang et al., 1990) or $L_p$-norm spherical distributions (Gupta & Song, 1997). The $L_p$-norm spherical distributions has density functions of the form $g(\|\mathbf{x}\|_p^p)$ for $g : [0, \infty) \to [0, \infty)$. If $\mathbf{x}$ follows the $L_p$-norm spherical distribution, then $\|\mathbf{x}\|_p$ and $\mathbf{x}/\|\mathbf{x}\|_p$ are also independent with each other.

There exist many other $L_p$-norm spherical distributions induced by the choice of the density generator function $g(\cdot)$ like $p$-generalized Weibull distribution, $L_p$-norm Pearson Type II distribution, $L_p$-norm Pearson Type VII distribution, $L_p$-norm multivariate t-distribution, $L_p$-norm multivariate Cauchy distributions etc (Gupta & Song, 1997). We particularly choose the Generalized Gaussian distribution with the density generator function $g(\cdot) = \exp(\cdot)$ for the inevitable consequences of the exponential function as the maximum entropy solutions. We show this in Lemma E.1.

## C.2. Multivariate Case - Multivariate Truncated Generalized Gaussian

Let $\mathbf{x} = (x_1, \ldots, x_d) \sim \prod_{i=1}^{d} \mathcal{TGN}_p(\mu, \sigma, S)$ be a Multivariate Truncated Generalized Gaussian random vector where each $x_i \sim \mathcal{TGN}_p(\mu, \sigma, S)$. For our purposes, we only need $S = (0, \infty)$ and thus the product support is $(0, \infty)^d$.

We observe that the angular distribution $\mathbf{x}/\|\mathbf{x}\|_p$ is still uniform over the support after truncation to the positive orthant $[0, \infty)^d$ for $p \in \{1, 2\}$. This is because truncation only rescales the density, which is already constant over the support. Due to the independence between $\|\mathbf{x}\|_p$ and $\mathbf{x}/\|\mathbf{x}\|_p$, the radial distribution is unchanged. Thus if $\mathbf{x} \sim \prod_{i=1}^{d} \mathcal{TGN}_{2.0}(0, \sigma, [0, \infty))$, then $\|\mathbf{x}\|_2^2 \sim \Gamma(d/2, 2\sigma^2)$ and $\mathbf{x}/\|\mathbf{x}\|_2 \sim \text{Unif}(\mathbb{S}_{\ell_2^+}^{d-1})$ where $\mathbb{S}_{\ell_p^+}^{d-1} := \{\mathbf{x} \in \mathbb{R}^d \cap [0, \infty)^d | \|\mathbf{x}\|_p = 1\}$ is the $\ell_p$ sphere confined to the positive orthant and $\text{Unif}(\cdot)$ is uniform distribution over the $\ell_p$ sphere confined to the positive orthant.

When $p = 1.0$, the multivariate truncated Laplace distribution $\prod_{i=1}^{d} \mathcal{TGN}_{1.0}(0, \sigma, [0, \infty))$ reduces to the product of i.i.d exponential distribution. Thus $\|\mathbf{x}\|_1 \sim \Gamma(d/1, \sigma)$ and $\mathbf{x}/\|\mathbf{x}\|_1$ is the Dirichlet distribution with all concentration parameters being 1 on the simplex $\Delta^{d-1}$, which we also denote as $\mathbb{S}_{\ell_1^+}^{d-1}$ (Devroye, 2006).

## C.3. Multivariate Case - Multivariate Rectified Generalized Gaussian

We denote $\mathbf{x} = (x_1, \ldots, x_d) \sim \prod_{i=1}^{d} \mathcal{RGN}_p(\mu, \sigma)$ as a Multivariate Rectified Generalized Gaussian random vector where each $x_i \sim \mathcal{RGN}_p(\mu, \sigma)$. Contrary to the family of Truncated Generalized Gaussian distribution with smooth isotropic $\ell_p$ geometry, rectification collapses most of the samples in the interior of the positive orthant into an exponentially large family of lower-dimensional faces, inducing polyhedral conic geometry. In fact, the probability of the random vector being in the interior of the positive orthant $[0, \infty)^d$ is $(1 - \Phi_{\mathcal{GN}_p(0,1)}(-\mu/\sigma))^d$, which decays to 0 exponentially fast as $d \to \infty$. Thus in high dimensions, most of the rectified samples concentrates on the boundary of the positive orthant cone.

## C.4. Proof of Proposition 3.5 (Theoretical Expected $\ell_0$ Sparsity)

*Proof.* Let $\mathbf{z} \sim \prod_{i=1}^{d} \mathcal{GN}_p(\mu, \sigma)$ be a Generalized Gaussian random vector in $d$ dimensions and $\mathbf{x} = \text{ReLU}(\mathbf{z})$, or equivalently, $\mathbf{x} \sim \prod_{i=1}^{d} \mathcal{RGN}_p(\mu, \sigma)$. By construction, we have independence between dimensions. Thus

$$\|\mathbf{x}\|_0 = \sum_{i=1}^{d} \mathbb{1}_{\mathbf{x}_i > 0} = \sum_{i=1}^{d} \mathbb{1}_{\mathbf{z}_i > 0} \tag{C.88}$$

So we have the expectation given by

$$\mathbb{E}[\|\mathbf{x}\|_0] = \sum_{i=1}^{d} \mathbb{E}[\mathbb{1}_{\mathbf{z}_i > 0}] = \sum_{i=1}^{d} \mathbb{P}(\mathbf{z}_i > 0) = \sum_{i=1}^{d} \Phi_{\mathcal{GN}_p(0,1)}\left(\frac{\mu}{\sigma}\right) = d \cdot \Phi_{\mathcal{GN}_p(0,1)}\left(\frac{\mu}{\sigma}\right) \tag{C.89}$$

where the CDF defined in Definition 3.1 evaluates to

$$\Phi_{\mathcal{GN}_p(0,1)}\left(\frac{\mu}{\sigma}\right) = \frac{1}{2}\left(1 + \text{sgn}\left(\frac{\mu}{\sigma}\right)P\left(\frac{1}{p}, \frac{|\mu/\sigma|^p}{p}\right)\right) \tag{C.90}$$

Thus

$$\mathbb{E}[\|\mathbf{x}\|_0] = \frac{d}{2}\left(1 + \text{sgn}\left(\frac{\mu}{\sigma}\right)P\left(\frac{1}{p}, \frac{|\mu/\sigma|^p}{p}\right)\right) \tag{C.91}$$

$\square$

# D. Choice of $\sigma$ for Rectified Generalized Gaussian

In the following section, we investigate how we should pick the scale parameter $\sigma$ for the Rectified Generalized Gaussian distribution $\mathcal{RGN}_p(\mu, \sigma)$. We show that different choices of $\sigma$ leads to different per-dimension variance (Appendix D.1), sparsity as measured by $\ell_0$ metrics (Appendix D.2), and also the sparsity-performance tradeoffs (Appendix D.3). We also provide our final recommendation of $\sigma$ at the end of this section.

## D.1. How does $\sigma$ affect the variance?

Equation (B.2) and Equation (B.41) are the closed form expression for the variance of the Generalized Gaussian $\mathcal{GN}_p(\mu, \sigma)$ and Rectified Genealized Gaussian distributions $\mathcal{RGN}_p(\mu, \sigma)$ respectively.

To prevent feature collapses along each feature dimension, we always want non-zero variance and hence the target distribution should have non-zero variance as well. We consider two strategies of picking $\sigma$.

First, we can set $\sigma = \sigma_{\text{GN}} = \Gamma(1/p)^{1/2}/(p^{1/p} \cdot \Gamma(3/p)^{1/2})$. In this case, the variance of the Generalized Gaussian distribution is fixed to be 1, i.e. $\text{Var}(\mathcal{GN}_p(\mu, \sigma_{\text{GN}})) = 1$ for any $\mu$ and $p$. However, the variance of the Rectified Generalized Gaussian distribution under the choice of $\sigma_{\text{GN}}$ is no longer fixed. In Figure 7, we plotted the variance of the Generalized Gaussian and the Rectified Generalized Gaussian distributions under the choice of $\sigma_{\text{GN}}$ with varying $\mu$ and $p$. We observe that the variance for the Generalized Gaussian distribution is indeed 1, but the variance for the Rectified Generalized Gaussian distribution decreases as we increase $p$ and decrease $\mu$. In the worst case, the variance of the Rectified Gaussian distribution $\mathcal{RGN}_2(-3, \sigma_{\text{GN}})$ is around 0.0002.

Second, we can also pick $\sigma = \sigma_{\text{RGN}}$ such that the variance of the Rectified Generalized Gaussian distribution is 1, i.e. $\text{Var}(\mathcal{RGN}_p(\mu, \sigma_{\text{GN}})) = 1$ for any $\mu$ and $p$. Since the closed form expression in Equation (B.41) is complicated, we resort to using a bisection search algorithm (see Algorithm 2) to estimate $\sigma_{\text{RGN}}$. In Figure 12a, we observe that it only takes around 30 iterations, invariant to choices of $\mu$ and $p$, to estimate $\sigma_{\text{RGN}}$ with bisection error below $10^{-10}$. We also only need to estimate $\sigma_{\text{RGN}}$ once for any $\mu$ and $p$.

In Figure 8, we reported the variance of the Generalized Gaussian and the Rectified Generalized Gaussian distributions if we choose $\sigma = \sigma_{\text{RGN}}$. Both theoretically and empirically, the variance of the Rectified Generalized Gaussian distribution is 1. Under the choice of $\sigma_{\text{RGN}}$, we observe that the variance of the Generalized Gaussian distribution increases as we increase $p$ and decreases $\mu$. In the extreme case, the variance of the Gaussian distribution $\mathcal{GN}_2(-3, \sigma_{\text{RGN}})$ is around 11.56.

In Figure 9, we also visualize values of $\sigma_{\text{GN}}$ and $\sigma_{\text{RGN}}$ under varying $\mu$ and $p$. We observe that none of the values of $\sigma$'s are extreme, and thus our sampling method (Algorithm 1) for Rectified Generalized Gaussian also won't be subjected to extreme value multiplications.

## D.2. How does $\sigma$ affect the sparsity?

Intuitively, it seems desirable to pick $\sigma_{\text{RGN}}$ over $\sigma_{\text{GN}}$ because the choice of $\sigma_{\text{RGN}}$ encourages the per-dimensional variance of the features to be 1, which is desirable as we know from the results in VICReg (Bardes et al., 2022). However, we observe that there is no simple free lunch here. Rectifications in general reduce variance by squashing negative values to be zeros, and enforcing the variance after rectifications being 1 will reduce sparsity.

In Figure 10, we report the theoretical $\ell_0$ norms evaluated based on Proposition 3.5 and the empirical $\ell_0$ norms computed over pretrained Rectified LpJEPA features. The choice of $\sigma_{\text{RGN}}$ leads to reduced sparsity measured by increased normalized $\ell_0$ norms $(1/D) \cdot \mathbb{E}[\|\mathbf{x}\|_0]$ both theoretically and empirically. Interestingly, we note that for the choice of $\sigma_{\text{RGN}}$, the primary way to increase sparsity is to reduce $p$. If we choose $\sigma = \sigma_{\text{GN}}$, then sparsity is more easily induced by decreasing $\mu$, whereas varying $p$ only induces small gaps in the amount of sparsity both theoretically and empirically.

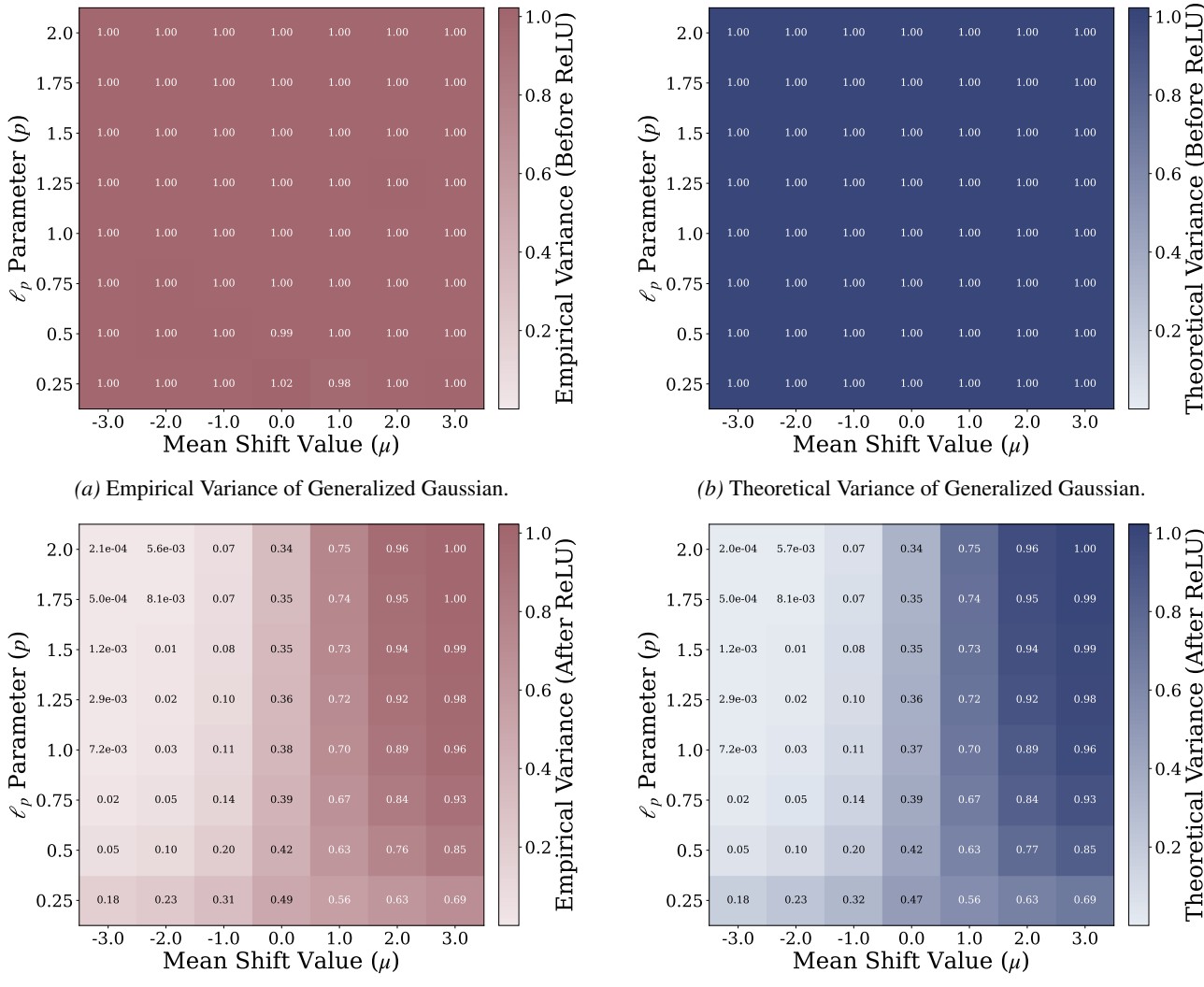

*(a)* Empirical Variance of Generalized Gaussian.

*(b)* Theoretical Variance of Generalized Gaussian.

*(c)* Empirical Variance of Rectified Generalized Gaussian.

*(d)* Theoretical Variance of Rectified Generalized Gaussian.

*Figure 7.* **Variance of Generalized Gaussian Distribution and Rectified Generalized Gaussian distributions under the choice of $\sigma = \sigma_{\mathrm{GN}}$. Top row:** Variance of $x \sim \mathcal{GN}_p(\mu, \sigma_{\mathrm{GN}})$. (a) The empirical variance of $\mathcal{GN}_p(\mu, \sigma_{\mathrm{GN}})$. (b) The theoretical variance of $\mathcal{GN}_p(\mu, \sigma_{\mathrm{GN}})$ by evaluating Equation (B.2). **Bottom row:** Variance of $x \sim \mathcal{RGN}_p(\mu, \sigma_{\mathrm{GN}})$. (c) The empirical variance of $\mathcal{RGN}_p(\mu, \sigma_{\mathrm{GN}})$. (d) The theoretical variance of $\mathcal{RGN}_p(\mu, \sigma_{\mathrm{GN}})$ by evaluating Equation (B.41). The empirical variance in (a) and (c) are computed by i.i.d sampling 100000 samples from 32 dimensions from either $\mathcal{GN}_p(\mu, \sigma_{\mathrm{GN}})$ or $\mathcal{RGN}_p(\mu, \sigma_{\mathrm{GN}})$. The per-dimension variance is estimated and we report the average variance across dimensions as a function of the mean shift value $\mu$ and the parameter $p$.

### D.3. How does $\sigma$ affect performance?

We have already observed in Figure 10 that for the same value of $\mu$ (more specifically, $\mu < 0$ as we're interested in sparse representations) and $p$, choosing $\sigma_{\mathrm{GN}}$ always lead to sparser representations. However, we're rather more interested in whether the pareto frontier of sparsity-performance tradeoff induced by the choices of $\{\mu, \sigma_{\mathrm{GN}}, p\}$ can be significantly different from that of $\{\mu, \sigma_{\mathrm{GN}}, p\}$. In other words, we would like to know if the choices of $\sigma_{\mathrm{GN}}$ or $\sigma_{\mathrm{RGN}}$ can lead to systematically better sparsity-performance tradeoffs as we vary $\mu$ and $p$.

In Figure 11, we show that in fact there is again no free lunch here. We report CIFAR-100 validation accuracy of pretrained Rectified LpJEPA projector representations against different mean shift values $\mu$ (Figure 11a), $\ell_1$ sparsity metrics $(1/D) \cdot \mathbb{E}[\|\mathbf{z}\|_1^2 / \|\mathbf{z}\|_2^2]$ (Figure 11b), and $\ell_0$ metrics $(1/D) \cdot \mathbb{E}[\|\mathbf{z}\|_0]$ (Figure 11c) across varying $p$. In general, $\{\mu, \sigma_{\mathrm{GN}}, p\}$ has different sparsity patterns compared to $\{\mu, \sigma_{\mathrm{RGN}}, p\}$, but the overall sparsity-performance tradeoffs are largely overlapped. Under $\ell_0$ metric, we actually observe that Rectified Laplace $\mathcal{RGN}_1(\mu, \sigma_{\mathrm{GN}})$ stands out as the setting that attains the best sparsity and accuracy tradeoff. Thus even if the choice of $\sigma_{\mathrm{GN}}$ can lead to small variance as we show in Figure 7, we still choose $\sigma_{\mathrm{GN}}$ as the default scale parameter for our target Rectified Generalized Gaussian distribution.

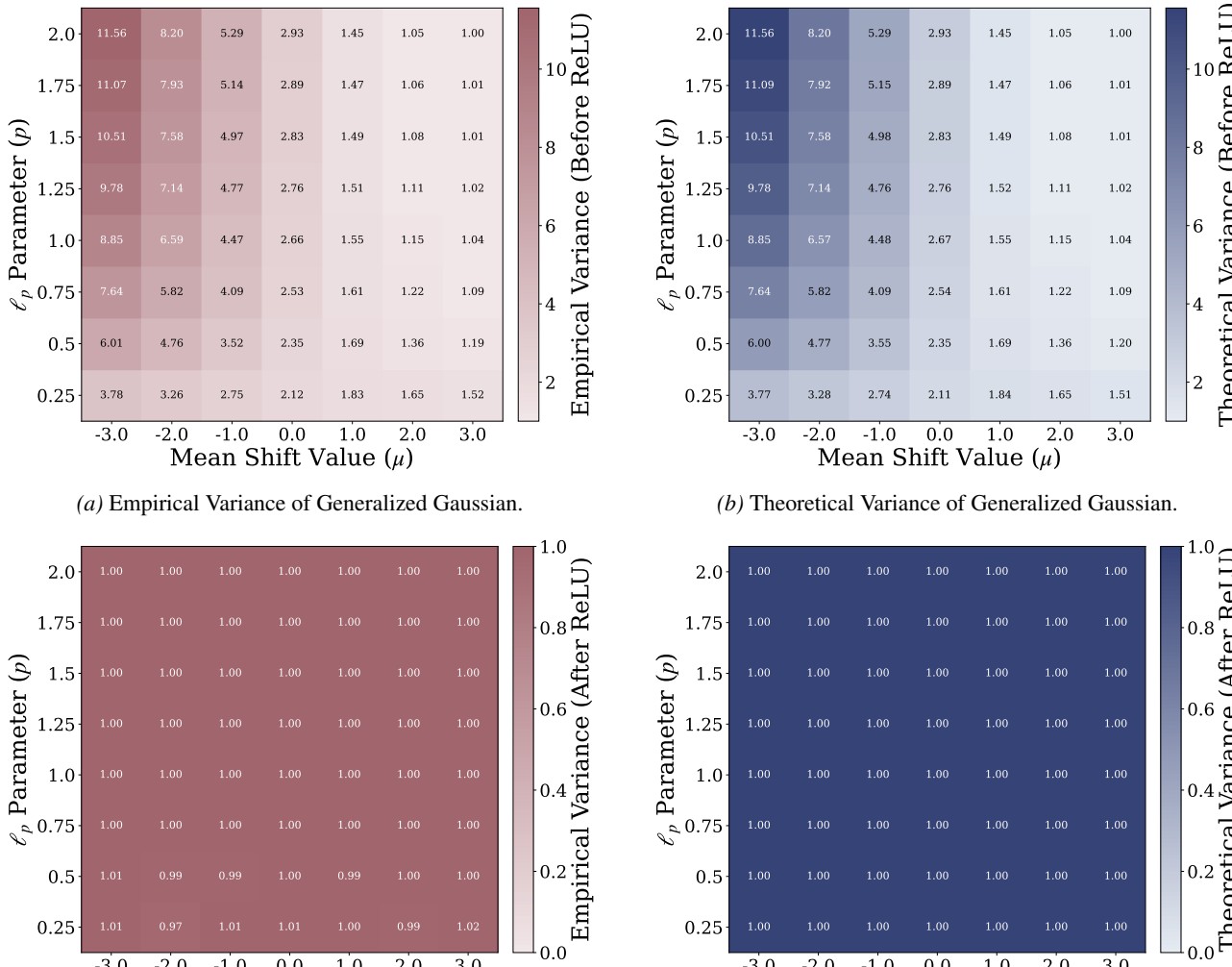

*(a)* Empirical Variance of Generalized Gaussian.

*(b)* Theoretical Variance of Generalized Gaussian.

*(c)* Empirical Variance of Rectified Generalized Gaussian.

*(d)* Theoretical Variance of Rectified Generalized Gaussian.

*Figure 8.* **Variance of Generalized Gaussian Distribution and Rectified Generalized Gaussian distributions under the choice of** $\sigma = \sigma_{\mathrm{RGN}}$**. Top row:** Variance of $x \sim \mathcal{GN}_p(\mu, \sigma_{\mathrm{RGN}})$. (a) The empirical variance of $\mathcal{GN}_p(\mu, \sigma_{\mathrm{RGN}})$. (b) The theoretical variance of $\mathcal{GN}_p(\mu, \sigma_{\mathrm{RGN}})$ by evaluating Equation (B.2). **Bottom row:** Variance of $x \sim \mathcal{RGN}_p(\mu, \sigma_{\mathrm{RGN}})$. (c) The empirical variance of $\mathcal{RGN}_p(\mu, \sigma_{\mathrm{RGN}})$. (d) The theoretical variance of $\mathcal{RGN}_p(\mu, \sigma_{\mathrm{RGN}})$ by evaluating Equation (B.41). The empirical variance in (a) and (c) are computed by i.i.d sampling 100000 samples from 32 dimensions from either $\mathcal{GN}_p(\mu, \sigma_{\mathrm{RGN}})$ or $\mathcal{RGN}_p(\mu, \sigma_{\mathrm{RGN}})$. The per-dimension variance is estimated and we report the average variance across dimensions as a function of the mean shift value $\mu$ and the parameter $p$.

## E. Maximum Differential Entropy Distributions

In the following section, we present a well-known statement for the form of the maximum-entropy probability distributions (Appendix E.1) and use the result to prove that the Multivariate Truncated Generalized Gaussian distribution is the maximum-entropy distribution under the expected $\ell_p$ norm constraints given a fixed support (Appendix E.2). We further show that the constraint is $\mathbb{E}[\|\mathbf{z}\|_p^p] = d\sigma^p$ without truncation (Appendix E.3). In Appendix E.4 and Appendix E.5, we present the well-known corollary of product Laplace and isotropic Gaussian being the maximum-entropy distributions under expected $\ell_1$ and $\ell_2$ norm constraints respectively.

### E.1. Derivation of Maximum Entropy Continuous Multivariate Probability Distributions under Support Constraints

Cover & Thomas (2006) provided a characterization of maximum entropy continuous univariate probability distributions. In Lemma E.1, we provide a multivariate extension of the maximum entropy probability distribution under the support set with positive Lebesgue measure.

**Lemma E.1** (Maximum Entropy Continuous Multivaraite Probability Distributions). *Let $S \subseteq \mathbb{R}^d$ be a measurable set with positive*

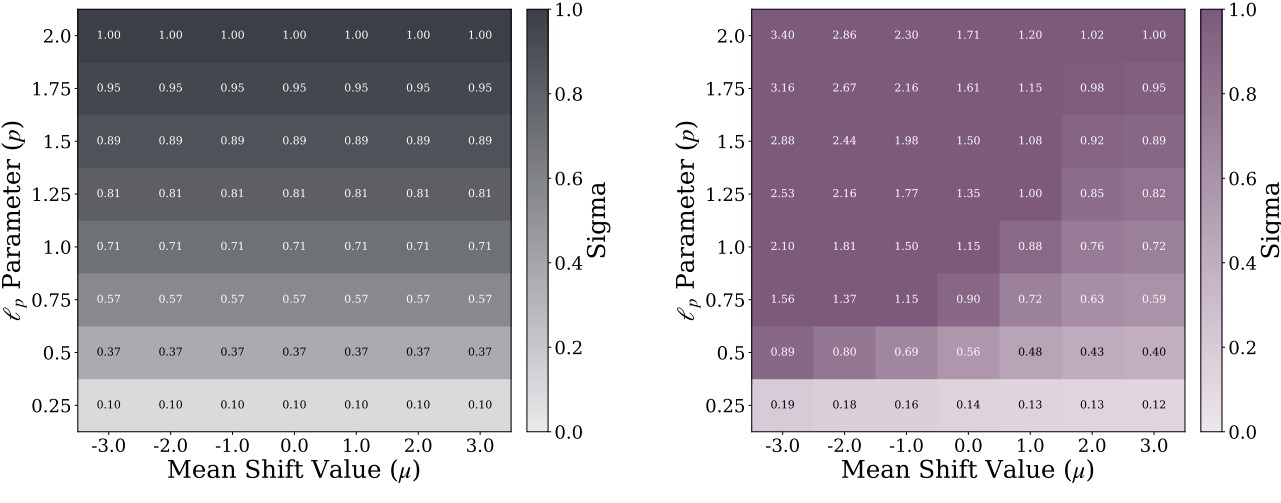

*(a)* Values of $\sigma_{\mathrm{GN}}$ across varying $\mu$ and $p$.

*(b)* Values of $\sigma_{\mathrm{RGN}}$ across varying $\mu$ and $p$.

*Figure 9.* **Values of $\sigma_{\mathrm{GN}}$ and $\sigma_{\mathrm{RGN}}$ under Different Choices of $\mu$ and $p$.** (a) The values of $\sigma_{\mathrm{GN}}$ are invariant to the mean shift value $\mu$. (b) $\sigma_{\mathrm{RGN}}$ changes as a function of both $\mu$ and $p$.

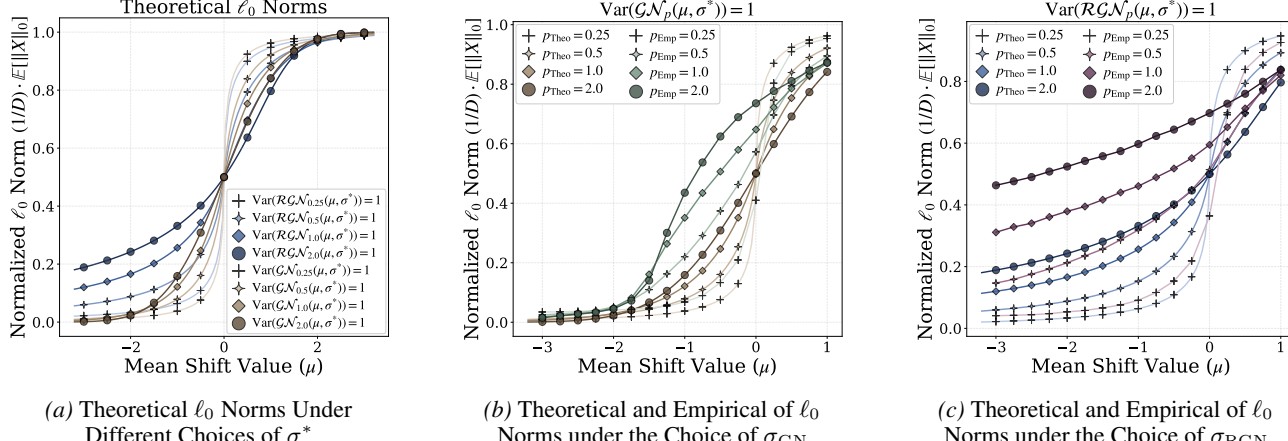

*(a)* Theoretical $\ell_0$ Norms Under Different Choices of $\sigma^*$

*(b)* Theoretical and Empirical of $\ell_0$ Norms under the Choice of $\sigma_{\mathrm{GN}}$

*(c)* Theoretical and Empirical of $\ell_0$ Norms under the Choice of $\sigma_{\mathrm{RGN}}$

*Figure 10.* **The theoretical and empirical normalized $\ell_0$ norms under Different Choices of $\sigma^*$.** (a) We report the theoretical $\ell_0$ norms based on Proposition 3.5 for $\sigma^* \in \{\sigma_{\mathrm{GN}}, \sigma_{\mathrm{RGN}}\}$ for varying $\mu$ and $p$. (b) The empirical $\ell_0$ norms of pretrained Rectified LpJEPA features are measured against the theoretical $\ell_0$ norms of the target Rectified Generalized Gaussian distribution $\mathcal{RGN}_p(\mu, \sigma_{\mathrm{GN}})$ for varying $\mu$ and $p$. (c) We plot the empirical $\ell_0$ norms of pretrained Rectified LpJEPA features against the theoretical $\ell_0$ norms of the target Rectified Generalized Gaussian distribution $\mathcal{RGN}_p(\mu, \sigma_{\mathrm{RGN}})$ for varying $\mu$ and $p$.

*Lebesgue measure. We define $r_1, \cdots, r_m : S \to \mathbb{R}$ as measurable functions and let $\alpha_1, \cdots, \alpha_m \in \mathbb{R}$. Consider the optimization problem*

$$\max_p \quad -\int_S p(\mathbf{x}) \ln p(\mathbf{x}) d\mathbf{x} \tag{E.92}$$

$$s.t. \quad \int_S p(\mathbf{x}) d\mathbf{x} = 1, \tag{E.93}$$

$$\int_S r_i(\mathbf{x}) p(\mathbf{x}) d\mathbf{x} = \alpha_i, \quad i = 1, \cdots, m, \tag{E.94}$$

$$p(\mathbf{x}) \geq 0 \quad a.e. \text{ on } S. \tag{E.95}$$

*We denote the set of functions that satisfies the given constraints as*

$$\mathcal{P} := \left\{ p :\to [0, \infty) \,\Big|\, \int_S p(\mathbf{x}) \, d\mathbf{x} = 1, \int_S r_i(\mathbf{x}) p(\mathbf{x}) \, d\mathbf{x} = \alpha_i \, \forall i \right\} \tag{E.96}$$

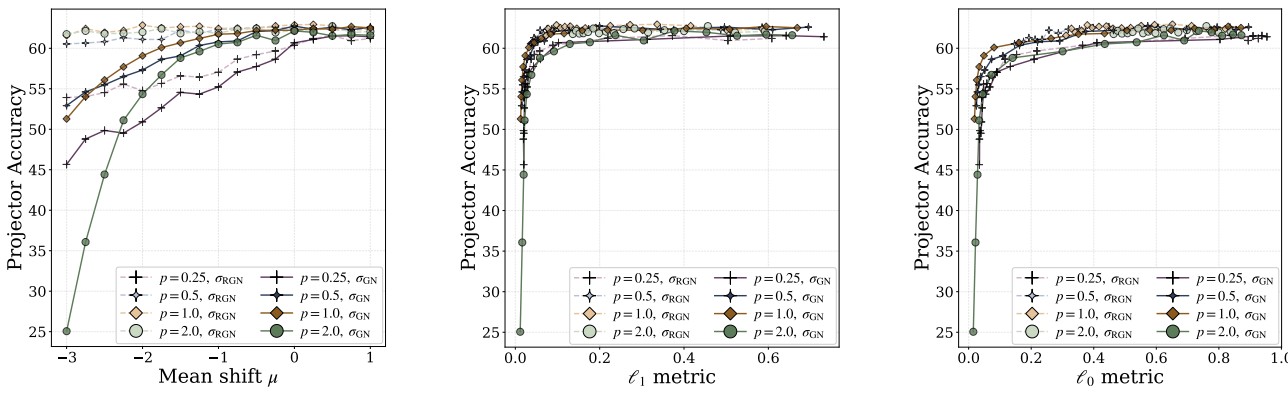

*(a)* Accuracy versus Mean Shift Value $\mu$

*(b)* Accuracy versus $\ell_1$ Sparsity Metric

*(c)* Accuracy versus $\ell_0$ Sparsity Metric

*Figure 11.* **The Sparsity-Performance Tradeoffs under Different Chocies of** $\sigma^* \in \{\sigma_{\mathrm{GN}}, \sigma_{\mathrm{RGN}}\}$. (a) We report CIFAR-100 validation accuracy for pretrained Rectified LpJEPA projector representations under varying $\{\mu, \sigma, p\}$. Under the same mean shift value $\mu$, choosing $\sigma_{\mathrm{RGN}}$ leads to better performance compared to $\sigma_{\mathrm{GN}}$ if $\mu$ is more negative. (b) Projector accuracy is plotted against the $\ell_1$ sparsity metrics measured over the pretrained Rectified LpJEPA projector representations. The gaps between $\sigma_{\mathrm{GN}}$ and $\sigma_{\mathrm{RGN}}$ are negligible. (c) Switching from $\ell_1$ to $\ell_0$ sparsity metrics, we observe the same behavior. In fact, $\sigma_{\mathrm{GN}}$ attains minor advantages in the sparsity-performance tradeoffs, especially when $p = 1$ or $p = 0.5$.

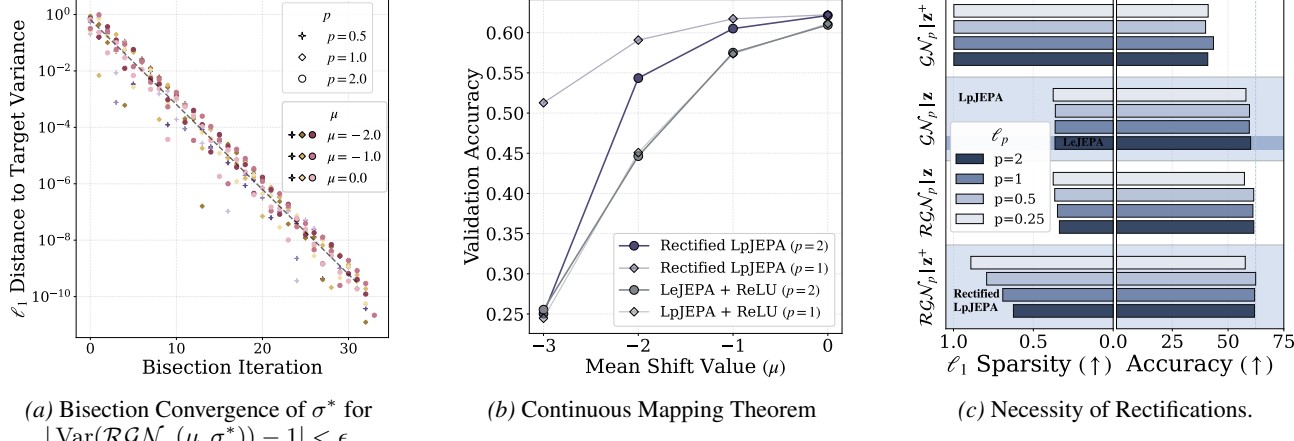

*(a)* Bisection Convergence of $\sigma^*$ for $|\mathrm{Var}(\mathcal{RGN}_p(\mu, \sigma^*)) - 1| < \epsilon$.

*(b)* Continuous Mapping Theorem

*(c)* Necessity of Rectifications.

*Figure 12.* **Additional Results on the Choices of** $\sigma$**, the Location of** $\mathrm{ReLU}(\cdot)$**, and the Ablations of** $\mathrm{ReLU}(\cdot)$ **for Rectified LpJEPA**. (a) We report the bisection convergence error as a function of optimization iterations for finding the optimal $\sigma_{\mathrm{RGN}}$ (see Appendix D). (b) We compared Rectified LpJEPA versus a version of distribution matching towards the Rectified Generalizd Gaussian distribution via the continuous mapping theorem (see Section 5.3). Rectified LpJEPA is the better design. (c) We show that Rectified LpJEPA attains the best sparsity-performance tradeoffs across various ablations of $\mathrm{ReLU}(\cdot)$ under the $\ell_1$ sparsity metrics $(1/D) \cdot \mathbb{E}[\|\mathbf{z}\|_1^2/\|\mathbf{z}\|_2^2]$. See Section 5.2 for details.

*Assume the set $\mathcal{P}$ is nonempty and that there exists $\boldsymbol{\lambda} = (\lambda_1, \ldots, \lambda_m) \in \mathbb{R}^m$ such that*

$$Z_S(\boldsymbol{\lambda}) = \int_S \exp\left(\sum_{i=1}^m \lambda_i r_i(\mathbf{x})\right) d\mathbf{x} \; < \; \infty. \tag{E.97}$$

*Then any maximizer $p^\star$ of the optimization problem has the form*

$$p^\star(\mathbf{x}) = \frac{1}{Z_S(\boldsymbol{\lambda})} \exp\left(\sum_{i=1}^m \lambda_i r_i(\mathbf{x})\right) \cdot \mathbb{1}_S(\mathbf{x}), \tag{E.98}$$

*where $\{\lambda_i\}_{i=1}^m$ are chosen to satisfy the constraints $\{\alpha_i\}_{i=1}^m$.*

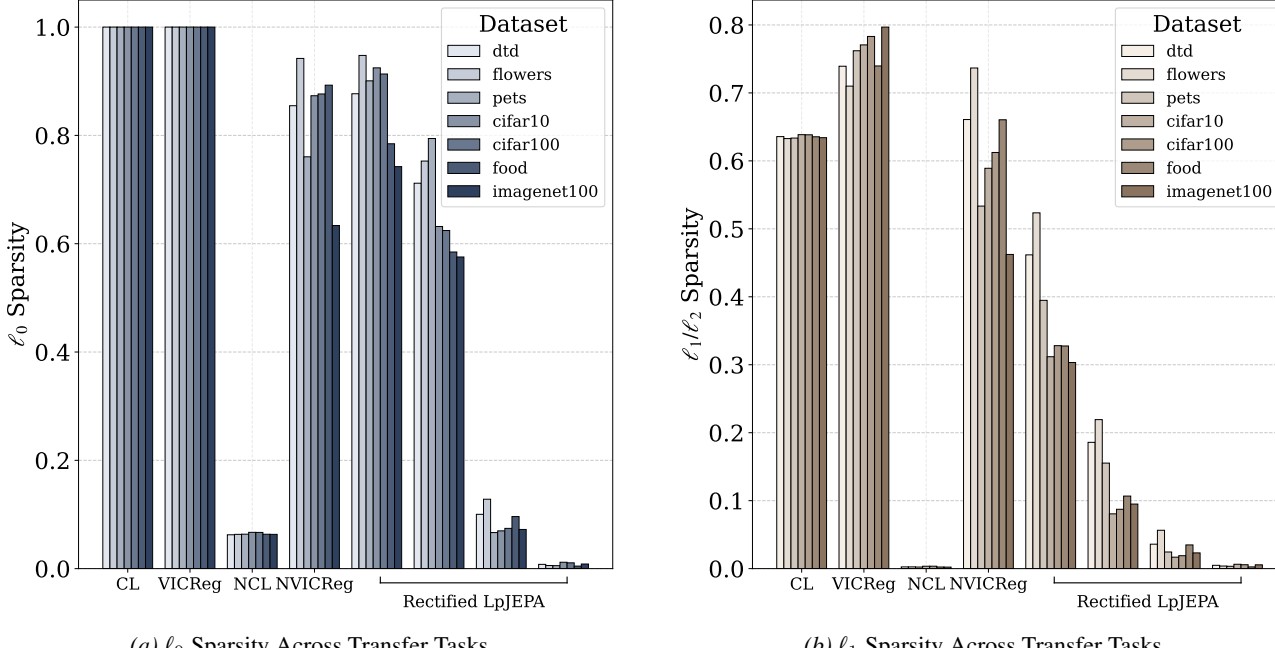

*(a) $\ell_0$ Sparsity Across Transfer Tasks*  *(b) $\ell_1$ Sparsity Across Transfer Tasks*

*Figure 13.* **Pretrained dense and sparse representations exhibits varying level of sparsity across different downstream tasks**. We compare the $\ell_0$ and $\ell_1$ sparsity metrics for Rectified LpJEPA versus other baselines (see Appendix H) pretrained over ImageNet-100 across a variety of downstream tasks. (a) Rectified LpJEPA has varying $\ell_0$ sparsity $(1/D) \cdot \mathbb{E}[\|\mathbf{z}\|_0]$ over different datasets as we vary the mean shift value $\mu \in \{0, -1, -2, -3\}$. CL and VICReg always have all entries being non-zero due to the lack of explicit rectifications. (b) Under $\ell_1$ sparsity metric $(1/D) \cdot \mathbb{E}[\|\mathbf{z}\|_1^2/\|\mathbf{z}\|_2^2]$, we observe varying sparsity for Rectified LpJEPA over different datasets for mean shift values $\ell_0$ sparsity $(1/D) \cdot \mathbb{E}[\|\mathbf{z}\|_0]$. We observe that NCL in fact achieves the lowest $\ell_1$ metric, but as we show in Figure 5c the most amount of variations is attained by Rectified LpJEPA.

*Proof.* We can form the Lagrangian functional of the constrained optimization problem as

$$\mathcal{J}[p] = -\int_S p(\mathbf{x}) \ln p(\mathbf{x}) d\mathbf{x} + \lambda_0 \left( \int_S p(\mathbf{x}) d\mathbf{x} - 1 \right) + \sum_{i=1}^{m} \lambda_i \left( \int_S r_i(\mathbf{x}) p(\mathbf{x}) d\mathbf{x} - \alpha_i \right) \tag{E.99}$$

where $\lambda_0, \lambda_1, \ldots, \lambda_m \in \mathbb{R}$ are Lagrange multipliers. Let $p$ be a maximizer that is strictly positive almost everywhere (a.e.) over $S$. We denote $\delta p$ as an arbitrary integrable perturbation supported on $S$ such that $p + \epsilon \delta p \geq 0$ for sufficiently small $|\epsilon|$. Thus the Gateaux derivative of $\mathcal{J}$ in the direction of $\delta p$ is given by

$$\frac{d}{d\epsilon} \mathcal{J}[p + \epsilon \delta p]\Big|_{\epsilon=0} = -\int_S \frac{d}{d\epsilon}\left[ (p(\mathbf{x}) + \epsilon \delta p(\mathbf{x})) \ln(p(\mathbf{x}) + \epsilon \delta p(\mathbf{x})) \right] d\mathbf{x}\Big|_{\epsilon=0} + \lambda_0 \left( \int_S \frac{d}{d\epsilon}\left[ p(\mathbf{x}) + \epsilon \delta p(\mathbf{x}) \right] d\mathbf{x}\Big|_{\epsilon=0} \right) \tag{E.100}$$

$$+ \sum_{i=1}^{m} \lambda_i \left( \int_S \frac{d}{d\epsilon}\left[ r_i(\mathbf{x})(p(\mathbf{x}) + \epsilon \delta p(\mathbf{x})) ) d\mathbf{x} \right]\Big|_{\epsilon=0} \right) \tag{E.101}$$

$$= -\int_S \left[ \delta p(\mathbf{x}) \ln(p(\mathbf{x}) + \epsilon \delta p(\mathbf{x})) + \delta p(\mathbf{x}) \right] d\mathbf{x}\Big|_{\epsilon=0} + \lambda_0 \left( \int_S 1 \cdot \delta p(\mathbf{x}) d\mathbf{x} \right) \tag{E.102}$$

$$+ \sum_{i=1}^{m} \lambda_i \left( \int_S r_i(\mathbf{x}) \delta p(\mathbf{x}) d\mathbf{x} \right) \tag{E.103}$$

$$= -\int_S \left[ \ln p(\mathbf{x}) + 1 \right] \delta p(\mathbf{x}) d\mathbf{x} + \left( \int_S \lambda_0 \cdot \delta p(\mathbf{x}) d\mathbf{x} \right) + \left( \int_S \sum_{i=1}^{m} \lambda_i r_i(\mathbf{x}) \delta p(\mathbf{x}) d\mathbf{x} \right) \tag{E.104}$$

$$\tag{E.105}$$

Thus the functional derivative is

$$\frac{\delta \mathcal{J}}{\delta p} = -\ln p(\mathbf{x}) - 1 + \lambda_0 + \sum_{i=1}^{m} \lambda_i r_i(\mathbf{x}) \tag{E.106}$$

Since this expression must vanish for all admissible perturbations $\delta p$, we get $\frac{\delta \mathcal{J}}{\delta p} = 0$ almost everywhere on $S$. Solving for $p$ yields

$$p(\mathbf{x}) = \exp\left(\lambda_0 - 1 + \sum_{i=1}^{m} \lambda_i r_i(\mathbf{x})\right) \tag{E.107}$$

Absorbing the constant terms into $Z_S(\boldsymbol{\lambda})$, we end up with

$$p(\mathbf{x}) = \frac{1}{Z_S(\boldsymbol{\lambda})} \exp\left(\sum_{i=1}^{m} \lambda_i r_i(\mathbf{x})\right) \cdot \mathbb{1}_S(\mathbf{x}) \tag{E.108}$$

$\square$

### E.2. Proof of Proposition 3.3 (Maximum Entropy Distribution Under the $\ell_p$ Norm and Support Constraint)

*Proof.* By Lemma E.1, the target distribution has the form of

$$p(\mathbf{x}) = \frac{1}{Z_S(\lambda_1)} \exp(\lambda_1 \|\mathbf{x}\|_p^p) \cdot \mathbb{1}_S(\mathbf{x}) \tag{E.109}$$

$$= \frac{1}{Z_S(\lambda_1)} \exp\left(-\frac{\|\mathbf{x}\|_p^p}{p\sigma^p}\right) \cdot \mathbb{1}_S(\mathbf{x}) \tag{E.110}$$

where we choose $\lambda_1 = -\frac{1}{p\sigma^p}$ which satisfies the constraint $\lambda_1 < 0$ for integration. Thus we have recovered the zero-mean Generalized Gaussian distribution with scale parameter $\sigma$. Now notice that

$$\frac{d}{d\lambda_1} \log Z_S(\lambda_1) = \frac{1}{Z_S(\lambda_1)} \frac{d}{d\lambda_1} Z_S(\lambda_1) \tag{E.111}$$

$$= \frac{1}{Z_S(\lambda_1)} \int_S \frac{d}{d\lambda_1} \exp(\lambda_1 \|\mathbf{x}\|_p^p) d\mathbf{x} \tag{E.112}$$

$$= \frac{1}{Z_S(\lambda_1)} \int_S \|\mathbf{x}\|_p^p \exp(\lambda_1 \|\mathbf{x}\|_p^p) d\mathbf{x} \tag{E.113}$$

$$= \int_S \|\mathbf{x}\|_p^p \frac{1}{Z_S(\lambda_1)} \exp\left(-\frac{\|\mathbf{x}\|_p^p}{p\sigma^p}\right) \cdot \mathbb{1}_S(\mathbf{x}) d\mathbf{x} \tag{E.114}$$

$$= \mathbb{E}[\|\mathbf{x}\|_p^p] \tag{E.115}$$

Thus we also obtain the constraint as $\mathbb{E}[\|\mathbf{x}\|_p^p] = \frac{d}{d\lambda_1} \log Z_S(\lambda_1)$. $\square$

### E.3. Maximum Entropy Distribution Under the $\ell_p$ Norm with Full Support

**Corollary E.2.** *If $S = \mathbb{R}^d$ in Proposition 3.3, then the constraint*

$$\mathbb{E}[\|\mathbf{x}\|_p^p] = \frac{d}{d\lambda_1} \log Z_S(\lambda_1) = d\sigma^p \tag{E.116}$$

*and we recover the Generalized Gaussian distribution with zero mean and scale parameter $\sigma$.*

$$p(x) = \frac{p^{d-d/p}}{(2\sigma)^d \Gamma(1/p)^d} \exp\left(-\frac{\|\mathbf{x}\|_p^p}{p\sigma^p}\right) \tag{E.117}$$

*Proof.* By Proposition 3.3, the target distribution has the form of

$$p(x) = \frac{1}{Z_S(\lambda_1)} \exp\left(-\frac{\|\mathbf{x}\|_p^p}{p\sigma^p}\right) \cdot \mathbb{1}_S(\mathbf{x}) \tag{E.118}$$

If $S = \mathbb{R}^d$, then the normalization constant becomes

$$\frac{1}{Z_S(\lambda_1)} = \frac{1}{Z_{\mathbb{R}^d}(\lambda_1)} = \frac{p^{d-d/p}}{(2\sigma)^d \Gamma(1/p)^d} \tag{E.119}$$

According to Dytso et al. (2018), we know that $\mathbb{E}[|\mathbf{x}_i|^p] = \sigma^p$. Thus

$$\mathbb{E}[\|\mathbf{x}\|_p^p] = \frac{d}{d\lambda_1} \log Z_S(\lambda_1) = d\sigma^p \tag{E.120}$$

$\square$

## E.4. Maximum Entropy Distribution Under the $\ell_1$ Norm Constraint

In Corollary E.3, we show the well-known result that the maximum-entropy continuous multivariate distribution under the $\ell_1$ norm constraint is the product Laplace distribution.

**Corollary E.3.** *The maximum entropy distribution over $\mathbb{R}^d$ under the constraints*

$$\int_{\mathbb{R}^d} p(\mathbf{x}) d\mathbf{x} = 1, \quad \mathbb{E}[\|\mathbf{x}\|_1] = db \tag{E.121}$$

*is the product of independent univariate symmetric Laplace distributions with zero mean and scale parameter $b$*

$$p(\mathbf{x}) = \left(\frac{1}{2b}\right)^d \exp\left(-\frac{\|\mathbf{x}\|_1}{b}\right) \tag{E.122}$$

*Proof.* By Lemma E.1, the target distribution has the form of

$$p(\mathbf{x}) \propto \exp(\lambda_1 \|\mathbf{x}\|_1) \tag{E.123}$$

with the constraint $\lambda_1 < 0$ for integration. After normalization, we obtain

$$p(\mathbf{x}) = \left(-\frac{\lambda_1}{2}\right)^d \exp(\lambda_1 \|\mathbf{x}\|_1) \tag{E.124}$$

By a change of variable $b = -1/\lambda_1$, we arrive at

$$p(\mathbf{x}) = \left(\frac{1}{2b}\right)^d \exp\left(-\frac{\|\mathbf{x}\|_1}{b}\right) \tag{E.125}$$

Thus we have recovered the zero-mean product Laplace distribution with scale parameter $b$. $\quad\square$

We note that the product Laplace distribution is different from the multivariate elliptical Laplace distribution presented in Kotz et al. (2012). For our purposes of identifying the maximum-entropy distribution under the expected $\ell_1$ norm constraints, we should use the product Laplace distribution as the multivariate generalization of the univariate symmetric Laplace distribution.

## E.5. Maximum Entropy Distribution under the $\ell_2$ Norm Constraint

In Corollary E.4, we present the well-known result that the maximum-entropy continuous multivariate distribution under the $\ell_2$ norm constraint is the isotropic Gaussian distribution.

**Corollary E.4.** *The maximum entropy distribution over $\mathbb{R}^d$ under the constraints*

$$\int_{\mathbb{R}^d} p(\mathbf{x}) d\mathbf{x} = 1, \quad \mathbb{E}[\mathbf{x}] = \boldsymbol{\mu}, \quad \mathbb{E}[(\mathbf{x} - \boldsymbol{\mu})(\mathbf{x} - \boldsymbol{\mu})^\top] = \boldsymbol{\Sigma} \tag{E.126}$$

*is the multivariate Gaussian distribution with mean $\boldsymbol{\mu}$ and covariance $\boldsymbol{\Sigma}$*

$$p(\mathbf{x}) \propto \exp\left(-\frac{1}{2}(\mathbf{x} - \boldsymbol{\mu})^\top \boldsymbol{\Sigma}^{-1}(\mathbf{x} - \boldsymbol{\mu})\right) \tag{E.127}$$

*When $\boldsymbol{\mu} = 0$ and $\boldsymbol{\Sigma} = \mathbf{I}$, the density function takes the form of*

$$p(\mathbf{x}) \propto \exp\left(-\frac{1}{2}\|\mathbf{x}\|_2^2\right) \tag{E.128}$$

*Proof.* Notice that the vector-valued mean constraint and matrix-valued covariance constraint can be factorized as a collection of scalar-valued constraints

$$\mathbb{E}[\mathbf{x}_i] = \boldsymbol{\mu}_i, \quad \mathbb{E}[\mathbf{x}_i \mathbf{x}_j] = \boldsymbol{\Sigma}_{ij} + \boldsymbol{\mu}_i \boldsymbol{\mu}_j, \quad \forall i, j \in \{1, \cdots, d\} \tag{E.129}$$

By Lemma E.1, the maximum entropy distribution has the form

$$p(\mathbf{x}) \propto \exp\left(\sum_{i=1}^d \boldsymbol{\lambda}_i \mathbf{x}_i + \sum_{i=1}^d \sum_{j=1}^d \boldsymbol{\Lambda}_{ij} \mathbf{x}_i \mathbf{x}_j\right) \tag{E.130}$$

$$= \exp\left(\boldsymbol{\lambda}^\top \mathbf{x} + \mathbf{x}^\top \boldsymbol{\Lambda} \mathbf{x}\right) \tag{E.131}$$

$$\propto \exp\left(-\frac{1}{2}(\mathbf{x} - \boldsymbol{\mu})^\top \boldsymbol{\Sigma}^{-1}(\mathbf{x} - \boldsymbol{\mu})\right) \tag{E.132}$$

for $\boldsymbol{\lambda} = \boldsymbol{\Sigma}^{-1}\boldsymbol{\mu}$ and $\boldsymbol{\Lambda} = -\frac{1}{2}\boldsymbol{\Sigma}^{-1}$, which is the multivariate Gaussian distribution up to normalizations. When $\boldsymbol{\mu} = 0$ and $\boldsymbol{\Sigma} = \mathbf{I}$, the density function trivially evaluates to

$$p(\mathbf{x}) \propto \exp\left(-\frac{1}{2}\|\mathbf{x}\|_2^2\right) \tag{E.133}$$

which is the maximum-entropy distribution under the expected $\ell_2$ norm constraints based on Lemma E.1. □

# F. Rényi Information Dimension and Entropy

In the following section, we provide a self-contained review of the Rényi information dimension and the $d(\xi)$-dimensional entropy. The contents are based on the original paper by Rényi (1959). After introducing the basic concepts in Appendix F.1, we go on to derive and prove the corresponding quantities for our Rectified Generalized Distribution in Appendix F.2. In Appendix F.3, we provide an empirical estimator for the $d(\xi)$-dimensional entropy. We further show, perhaps somewhat trivially, that the $d(\xi)$-dimensional entropy is in fact equivalent to another notion of entropy where we replace the Lebesgue measure $\lambda$ in standard differential entropy with the mixed measure $\nu := \lambda + \delta_0$ (Appendix F.4) for $\delta_0$ being the Dirac measure in Definition B.4. In Appendix F.5, we discuss how the total correlation can be decomposed using different notions of entropy.

## F.1. Definition of the $d(\xi)$-dimensional Entropy

Conventionally, the differential entropy for a random variable $X$ with distribution function $\mathbb{P}_X$ is defined as

$$\mathbb{H}(X) = -\int \frac{d\mathbb{P}_X}{d\lambda}\log\left(\frac{d\mathbb{P}_X}{d\lambda}\right)d\lambda \tag{F.134}$$

where $\lambda$ is the Lebesgue measure. For a Rectified Generalized Gaussian random variable $X \sim \mathcal{RGN}_p(\mu, \sigma)$, the Radon-Nikodym derivative of $\mathbb{P}_X$ with respect to $\lambda$ does not exist as shown in Lemma B.6. As a result, differential entropy is ill-defined for the Rectified Generalized Gaussian distribution.

In the following section, we consider an alternative formulation known as the $d(\xi)$-dimensional entropy, where $d(\xi)$ is the Rényi information dimension of the random variable $\xi$ (Rényi, 1959). In Definition F.1, we review the basic definition of information dimension.

**Definition F.1** (Information Dimension (Rényi, 1959)). Consider a real-valued random variable $\xi \in \mathbb{R}$ and the discretization $\xi_n = (1/n) \cdot [n\xi]$, where $[x]$ preserves only the integral part of $x$. For example, $[3.42] = 3$. Under suitable conditions, the information dimension $d(\xi)$ exists and is given by

$$d(\xi) = \lim_{n\to\infty}\frac{\mathbb{H}_0(\xi_n)}{\log n} \tag{F.135}$$

where

$$\mathbb{H}_0(\eta) := \sum_{k=1}^{\infty}q_k\log\frac{1}{q_k} \tag{F.136}$$

is the Shannon entropy for a discrete random variable $\eta$ with probabilities $q_k$ for $k = 1, 2, \ldots$.

Intuitively, $\xi_n$ represents the quantization of the real-valued random variable $\xi$ at the grid resolution of $1/n$. Thus the information dimension $d(\xi)$ measures how fast the Shannon entropy grows as a result of finer and finer grind discretizations. In Definition F.2, we present the definition of the $d(\xi)$-dimensional entropy first introduced in Rényi (1959).

**Definition F.2** ($d(\xi)$-dimensional Entropy (Rényi, 1959)). If the information dimension $d(\xi)$ exists, the $d(\xi)$-dimensional entropy is defined as

$$\mathbb{H}_{d(\xi)} = \lim_{n\to\infty}\left(\mathbb{H}_0(\xi_n) - d(\xi)\log n\right) \tag{F.137}$$

Effectively, the $d(\xi)$-dimensional entropy measures the amount of uncertainty distributed along the $d(\xi)$ continuous degrees of freedom. For discrete random variable $x$, the information dimension $d(x) = 0$ since it's invariant to finer discretization (Rényi, 1959). Thus the discrete Shannon entropy is the 0-dimensional entropy $\mathbb{H}_0$. The continuous random variable $x'$ has information dimension $d(x') = 1$ and so the differential entropy is simply the 1-dimensional entropy $\mathbb{H}_1$ (Rényi, 1959).

In Definition F.3, we review the special case of $\mathbb{H}_{d(\xi)}$ when the random variable $\xi$ follows has a mixture probability measure.

**Definition F.3** ($d(\xi)$-dimensional Entropy for Mixed Measures (Rényi, 1959)). Let $\xi$ be a random variable with probability measure

$$\mathbb{P}_\xi = (1 - d) \cdot \mathbb{P}_0 + d \cdot \mathbb{P}_1 \tag{F.138}$$

where $\mathbb{P}_0$ is discrete and $\mathbb{P}_1$ is absolutely continuous with respect to the Lebesgue measure. Then the information dimension $d(\xi) = d$, and the $d(\xi)$-dimensional entropy is given by

$$\mathbb{H}_{d(\xi)}(\xi) = (1-d) \cdot \sum_{k=1}^{\infty} p_k \log \frac{1}{p_k} - d \cdot \int_{\mathbb{R}} \frac{d\mathbb{P}_1}{d\lambda}(x) \log\left(\frac{d\mathbb{P}_1}{d\lambda}(x)\right) d\lambda(x) + d \log \frac{1}{d} + (1-d) \log \frac{1}{1-d} \tag{F.139}$$

where $\lambda$ is the Lebesgue measure.

## F.2. Proof of Theorem 3.6 (The $d(\xi)$-dimensional Entropy of the Rectified Generalized Gaussian Distribution)

In the following section, we prove the $d(\boldsymbol{\xi})$-dimensional entropy characterization of the Multivariate Rectified Generalized Gaussian distribution presented in Theorem 3.6.

*Proof.* Since $\boldsymbol{\xi} \sim \prod_{i=1}^{D} \mathcal{RGN}_p(\mu, \sigma)$ where each $\boldsymbol{\xi}_i \sim \mathcal{RGN}_p(\mu, \sigma)$ are i.i.d, it's trivial that

$$d(\boldsymbol{\xi}) = D \cdot d(\boldsymbol{\xi}_i) \tag{F.140}$$

$$\mathbb{H}_{d(\boldsymbol{\xi})}(\boldsymbol{\xi}) = \sum_{i=1}^{D} \mathbb{H}_{d(\boldsymbol{\xi}_i)}(\boldsymbol{\xi}_i) = D \cdot \mathbb{H}_{d(\boldsymbol{\xi}_i)}(\boldsymbol{\xi}_i) \tag{F.141}$$

for all $i$ by independence. In Appendix F.5, we also present an alternative interpretation of the $d(\boldsymbol{\xi})$-dimensional entropy that enables the decomposition of the joint entropy $\mathbb{H}_{d(\boldsymbol{\xi})}(\boldsymbol{\xi})$ into the sums of the marginals $\mathbb{H}_{d(\boldsymbol{\xi}_i)}(\boldsymbol{\xi}_i)$ under the independence assumption. Thus it suffices to prove in the univariate case. By Definition B.5 and Definition F.3, we know that the information dimension is given by

$$d(\boldsymbol{\xi}_i) = 1 - \Phi_{\mathcal{GN}_p(0,1)}\left(-\frac{\mu}{\sigma}\right) = \Phi_{\mathcal{GN}_p(0,1)}\left(\frac{\mu}{\sigma}\right) \tag{F.142}$$

We observe that $\mathbb{P}_0$ in Definition F.3 correspond to the Dirac measure $\delta_0$ in Definition B.5. Thus

$$(1 - d(\boldsymbol{\xi}_i)) \cdot \sum_{k=1}^{\infty} p_k \log \frac{1}{p_k} = (1 - d(\boldsymbol{\xi}_i)) \cdot (1 \log 1) = 0 \tag{F.143}$$

Now we can define a Bernoulli gating random variable

$$\mathbb{1}_{(0,\infty)}(\boldsymbol{\xi}_i) = \begin{cases} 1, \text{if } \boldsymbol{\xi}_i \in (0,\infty) \implies \text{with probability } d(\boldsymbol{\xi}_i) \\ 0, \text{if } \boldsymbol{\xi}_i \notin (0,\infty) \implies \text{with probability } 1 - d(\boldsymbol{\xi}_i) \end{cases} \tag{F.144}$$

It's well known that the Shannon entropy for a Bernoulli random variable is

$$\mathbb{H}_0(\mathbb{1}_{(0,\infty)}(\boldsymbol{\xi}_i)) = d(\boldsymbol{\xi}_i) \log \frac{1}{d(\boldsymbol{\xi}_i)} + (1 - d(\boldsymbol{\xi}_i)) \log \frac{1}{1 - d(\boldsymbol{\xi}_i)} \tag{F.145}$$

Thus by Definition F.3, the $d(\boldsymbol{\xi}_i)$-dimensional entropy is

$$\mathbb{H}_{d(\boldsymbol{\xi}_i)}(\boldsymbol{\xi}_i) = 0 - d(\boldsymbol{\xi}_i) \cdot \int_{\mathbb{R}} \frac{d\mathbb{P}_{\mathcal{TGN}_p(\mu,\sigma)}}{d\lambda}(x) \log\left(\frac{d\mathbb{P}_{\mathcal{TGN}_p(\mu,\sigma)}}{d\lambda}(x)\right) d\lambda(x) + \mathbb{H}_0(\mathbb{1}_{(0,\infty)}(\boldsymbol{\xi}_i)) \tag{F.146}$$

$$= \Phi_{\mathcal{GN}_p(0,1)}\left(\frac{\mu}{\sigma}\right) \cdot \mathbb{H}_1(\mathcal{TGN}_p(\mu,\sigma)) + \mathbb{H}_0(\mathbb{1}_{(0,\infty)}(\boldsymbol{\xi}_i)) \tag{F.147}$$

So we have proven the expression in Theorem 3.6. $\square$

## F.3. Empirical Estimators of the $d(\xi)$-dimensional entropy

**Lemma F.4** (Probability Measure Under Rectification). *Let $X \sim \mathbb{P}_X$ be a real-valued random variable where $\mathbb{P}_X$ is absolutely continuous with respect to the Lebesgue measure $\lambda$, i.e. $\mathbb{P}_X \ll \lambda$. Then the probability measure of $Z := \max(0, X)$ over $([0,\infty), \mathcal{B}([0,\infty)))$ is*

$$\mathbb{P}_Z = (1-d) \cdot \delta_0 + d \cdot \mathbb{P}_{X|(0,\infty)} \tag{F.148}$$

*where $\delta_0$ is the Dirac measure and $1 - d := \mathbb{P}(Z = 0) = \mathbb{P}(X \leq 0)$ and*

$$\mathbb{P}_{X|(0,\infty)}(A) := \frac{\mathbb{P}_X(A \cap (0,\infty))}{\mathbb{P}_X((0,\infty))} = \frac{\mathbb{P}_X(A)}{d} \tag{F.149}$$

*for any Borel $A \subset (0,\infty)$.*

*Proof.* Let $\varphi : \mathbb{R} \to [0, \infty)$ be the rectification map $\varphi(x) := \max(0, x)$. Then $\mathbb{P}_Z$ is the pushforward of $\mathbb{P}_X$ by $\varphi$, i.e. for any Borel set $B \in \mathcal{B}([0, \infty))$,

$$\mathbb{P}_Z(B) = \mathbb{P}_X(\varphi^{-1}(B)). \tag{F.150}$$

We can write $\varphi^{-1}(B)$ as

$$\varphi^{-1}(B) = \left(\varphi^{-1}(B) \cap (-\infty, 0]\right) \cup \left(\varphi^{-1}(B) \cap (0, \infty)\right), \tag{F.151}$$

For $x \in (-\infty, 0]$, $\varphi(x) = 0$. So we have

$$\varphi^{-1}(B) \cap (-\infty, 0] = \begin{cases} (-\infty, 0], & 0 \in B, \\ \emptyset, & 0 \notin B. \end{cases} \tag{F.152}$$

Now for $(0, \infty)$, $\varphi(x) = x$. So we have

$$\varphi^{-1}(B) \cap (0, \infty) = B \cap (0, \infty). \tag{F.153}$$

Combining these together, we arrive at

$$\mathbb{P}_Z(B) = \mathbb{P}_X(\varphi^{-1}(B)) = \mathbb{P}_X(B \cap (0, \infty)) + \mathbb{P}_X((-\infty, 0]) \cdot \delta_0(B) \tag{F.154}$$

where $\delta_0(B)$ is the Dirac measure in Definition B.4 that evaluates to 1 if $0 \in B$ and 0 otherwise.

Let $d := \mathbb{P}(X > 0) = \mathbb{P}_X((0, \infty))$. So trivially, $1 - d = \mathbb{P}(X \le 0) = \mathbb{P}_X((-\infty, 0])$. By the definition of the conditional measure, we have that for any $A \in \mathcal{B}(\mathbb{R})$,

$$\mathbb{P}_{X|(0,\infty)}(A) := \frac{\mathbb{P}_X(A \cap (0, \infty))}{\mathbb{P}_X((0, \infty))} = \frac{\mathbb{P}_X(A \cap (0, \infty))}{d}, \qquad A \in \mathcal{B}(\mathbb{R}). \tag{F.155}$$

Then for every $B \in \mathcal{B}([0, \infty))$,

$$\mathbb{P}_Z(B) = d\, \mathbb{P}_{X|(0,\infty)}(B) + (1 - d)\, \delta_0(B) \tag{F.156}$$

Thus we have proven the expression of the probability measure. Now if $A \subset (0, \infty)$ is Borel, then $A \cap (0, \infty) = A$ and we have $\mathbb{P}_{X|(0,\infty)}(A) = \mathbb{P}_X(A)/d$.

$\square$

**Lemma F.5** (Radon–Nikodym Derivative Under Rectifications). *Let $X \sim \mathbb{P}_X$ be a real-valued random variable where $\mathbb{P}_X$ is absolutely continuous with respect to the Lebesgue measure $\lambda$, i.e. $\mathbb{P}_X \ll \lambda$. Consider $Z := \max(0, X) \sim \mathbb{P}_Z$ and let $\delta_0$ be the Dirac measure in Definition B.4. Then the Radon–Nikodym derivative of $\mathbb{P}_Z$ with respect to $\nu = \delta_0 + \lambda$ is given by*

$$\frac{d\mathbb{P}_Z}{d\nu}(x) = (1 - d) \cdot \mathbb{1}_{\{0\}}(x) + d \cdot \frac{d\mathbb{P}_{X|(0,\infty)}}{d\lambda}(x) \cdot \mathbb{1}_{(0,\infty)}(x) \tag{F.157}$$

*Proof.* Following the same arguments in Lemma B.6, we know that $\mathbb{P}_Z$ is absolutely continuous with respect to $\nu$, i.e. $\mathbb{P}_Z \ll \nu$. Again, following the same arguments in Lemma B.7, we observe that for any Borel $A \subset [0, \infty)$

$$\int_A \frac{d\mathbb{P}_Z}{d\nu} d\nu = \int_A \frac{d\mathbb{P}_Z}{d\nu} d\delta_0 + \int_A \frac{d\mathbb{P}_Z}{d\nu} d\lambda \tag{F.158}$$

Notice that

$$\int_A \frac{d\mathbb{P}_Z}{d\nu} d\delta_0 = \frac{d\mathbb{P}_Z}{d\nu}(0)\delta_0(A) = (1 - d) \cdot \delta_0(A) \tag{F.159}$$

and

$$\int_A \frac{d\mathbb{P}_Z}{d\nu} d\lambda = \int_A (1 - d) \cdot \mathbb{1}_{\{0\}}(x) d\lambda(x) + \int_A d \cdot \frac{d\mathbb{P}_{X|(0,\infty)}}{d\lambda}(x) \cdot \mathbb{1}_{(0,\infty)}(x) d\lambda(x) \tag{F.160}$$

$$= (1 - d) \cdot \int_A \mathbb{1}_{\{0\}}(x) d\lambda(x) + \int_{A \cap (0,\infty)} d \cdot \frac{d\mathbb{P}_{X|(0,\infty)}}{d\lambda}(x) \cdot d\lambda(x) \tag{F.161}$$

$$= 0 + d \cdot \int_{A \cap (0,\infty)} d\mathbb{P}_{X|(0,\infty)}(x) \tag{F.162}$$

$$= d \cdot \mathbb{P}_{X|(0,\infty)}(A) \tag{F.163}$$

Putting everything together, we have

$$\int_A \frac{d\mathbb{P}_Z}{d\nu} d\nu = (1-d) \cdot \delta_0(A) + d \cdot \mathbb{P}_{X|(0,\infty)}(A) = \mathbb{P}_Z(A) \tag{F.164}$$

Thus we have shown that the Radon–Nikodym derivative is correct. $\qquad\square$

Consider a real-valued random variable $X$ from some unknown distribution $\mathbb{P}_X$ that's absolutely continuous with respect to the Lebesgue measure. Let $Z = \max(X, 0)$ be the rectified random variable. Then by Lemma F.4, the probability measure of $Z$ can be written as

$$\mathbb{P}_Z = (1-d) \cdot \delta_0 + d \cdot \mathbb{P}_{X|(0,\infty)} \tag{F.165}$$

where $\delta_0$ is the Dirac measure and $1 - d := \mathbb{P}(Z = 0) = \mathbb{P}(X \leq 0)$ and

$$\mathbb{P}_{X|(0,\infty)}(A) = \frac{\mathbb{P}_X(A)}{\mathbb{P}_X((0,\infty))} \tag{F.166}$$

for any Borel $A \subset (0, \infty)$. This is a probabilistic model that is suitable for characterizing the neural network output feature marginal distributions after rectifications. Notice that Equation (F.165) is in the form presented in Definition F.3. Thus it's valid to compute the $d(Z)$-dimensional entropy for any distribution that follows the decomposition in Equation (F.165). We also observe that the Rectified Generalized Gaussian probability measure is just a special case of Equation (F.165).

By Definition F.3, the $d(Z)$-dimensional entropy is given by

$$\mathbb{H}_{d(Z)}(Z) = d(Z) \cdot \mathbb{H}_1(\mathbb{P}_{X|(0,\infty)}) + \mathbb{H}_0(\mathbb{1}_{(0,\infty)}(Z)) \tag{F.167}$$

In practice, we will have samples $\{z_i\}_{i=1}^B$ from the random variable $Z$. We can estimate the information dimension by

$$\hat{d}(Z) = \frac{1}{B} \sum_{i=1}^B \mathbb{1}_{(0,\infty)}(z_i) \tag{F.168}$$

Now we consider a subset $\{z_i\}_{i=1}^{B'}$ where each $z_i > 0$. The differential entropy over $\{z_i\}_{i=1}^{B'}$ can be computed using the $m$-spacing estimator (Vasicek, 1976; Learned-Miller et al., 2003)

$$\hat{\mathbb{H}}_1(\mathbb{P}_{X|(0,\infty)}) = \frac{1}{B'-m} \sum_{i=1}^{B'-m} \log\left(\frac{B'+1}{m}\left(z_{(i+m)} - z_{(i)}\right)\right) \tag{F.169}$$

where $m$ is a spacing hyperparameter, and $\{z_{(i)} \mid z_{(1)} \leq z_{(2)} \leq \cdots \leq z_{(B')}\}_{i=1}^{B'}$ are sorted samples of the original set $\{z_i\}_{i=1}^{B'}$. Putting these estimators together, the empirical $d(Z)$-dimensional entropy can be computed as

$$\hat{\mathbb{H}}_{d(Z)}(Z) = \hat{d}(Z) \cdot \hat{\mathbb{H}}_1(\mathbb{P}_{X|(0,\infty)}) + \hat{d}(Z) \log \frac{1}{\hat{d}(Z)} + (1 - \hat{d}(Z)) \log \frac{1}{1 - \hat{d}(Z)} \tag{F.170}$$

If we consider the multivariate case for the random vector $\mathbf{z} = \mathrm{ReLU}(\mathbf{x})$, where $\mathbf{x} \in \mathbb{R}^D$ follows some unknown distribution $\mathbb{P}_{\mathbf{x}}$ and $\mathrm{ReLU}(\cdot)$ applies coordinate-wise, then in general we cannot compute the $d(\mathbf{z})$-dimensional entropy of the joint distribution $\mathbb{P}_{\mathbf{z}}$ both due to lack of estimators and intractable complexity.

However, we can compute the upper bound of the joint entropy by computing the sums of the marginal entropies

$$\mathbb{H}_{d(\mathbf{z})}(\mathbf{z}) \leq \sum_{i=1}^D \mathbb{H}_{d(\mathbf{z}_i)}(\mathbf{z}_i) \tag{F.171}$$

where "$\leq$" reduces to the equality sign "$=$" if we have independence between all dimensions.

### F.4. Alternative Interpretation of the $d(\xi)$-dimensional Entropy

Let's denote the standard differential entropy as $\mathbb{H}_\lambda(X)$

$$\mathbb{H}_\lambda(X) = -\int \frac{d\mathbb{P}_X}{d\lambda} \log\left(\frac{d\mathbb{P}_X}{d\lambda}\right) d\lambda \tag{F.172}$$

where $\lambda$ is the Lebesgue measure, $X \sim \mathbb{P}_X$ is a real-valued random variable, and $\mathbb{P}_X \ll \lambda$. We know from Appendix F.3 that the probability measure of $Z := \text{ReLU}(X)$ is defined as

$$\mathbb{P}_Z = (1 - d) \cdot \delta_0 + d \cdot \mathbb{P}_{X|(0,\infty)} \tag{F.173}$$

Let's consider another definition of entropy with respect to the mixued measure $\nu := \delta_0 + \lambda$

$$\mathbb{H}_\nu(Z) = -\int \frac{d\mathbb{P}_Z}{d\nu} \log\left(\frac{d\mathbb{P}_Z}{d\nu}\right) d\nu \tag{F.174}$$

In Lemma F.6, we show that this coincides with the $d(Z)$-dimensional entropy in Definition F.3.

**Lemma F.6.** *The entropy* $\mathbb{H}_\nu(Z)$ *is equivalent to the* $d(Z)$-*dimensional entropy*

$$\mathbb{H}_\nu(Z) = H_{d(Z)}(Z) \tag{F.175}$$

*Proof.* We start by expanding the integral

$$\mathbb{H}_\nu(Z) = -\int \frac{d\mathbb{P}_Z}{d\nu} \log\left(\frac{d\mathbb{P}_Z}{d\nu}\right) d\nu \tag{F.176}$$

$$= -\int \frac{d\mathbb{P}_Z}{d\nu} \log\left(\frac{d\mathbb{P}_Z}{d\nu}\right) d\delta_0 - \int \frac{d\mathbb{P}_Z}{d\nu} \log\left(\frac{d\mathbb{P}_Z}{d\nu}\right) d\lambda \tag{F.177}$$

By the property of the Dirac measure, we have

$$-\int \frac{d\mathbb{P}_Z}{d\nu}(x) \log\left(\frac{d\mathbb{P}_Z}{d\nu}(x)\right) d\delta_0(x) = -\frac{d\mathbb{P}_Z}{d\nu}(0) \log\left(\frac{d\mathbb{P}_Z}{d\nu}(0)\right) = -(1 - d)\log(1 - d) \tag{F.178}$$

Lemma F.5 tells us that

$$\frac{d\mathbb{P}_Z}{d\nu}(x) = (1 - d) \cdot \mathbb{1}_{\{0\}}(x) + d \cdot \frac{d\mathbb{P}_{X|(0,\infty)}}{d\lambda}(x) \cdot \mathbb{1}_{(0,\infty)}(x) \tag{F.179}$$

Due to the term $\mathbb{1}_{\{0\}}(x)$, any Lebesgue measure evaluates to 0 since $\{0\}$ is a Lebesgue measure zero set. So effectively, we can write

$$-\int \frac{d\mathbb{P}_Z}{d\nu} \log\left(\frac{d\mathbb{P}_Z}{d\nu}\right) d\lambda = -\int d \cdot \frac{d\mathbb{P}_{X|(0,\infty)}}{d\lambda}(x) \cdot \mathbb{1}_{(0,\infty)}(x) \log(d \cdot \frac{d\mathbb{P}_{X|(0,\infty)}}{d\lambda}(x) \cdot \mathbb{1}_{(0,\infty)}(x)) d\lambda(x) \tag{F.180}$$

$$= -\int d \cdot \frac{d\mathbb{P}_{X|(0,\infty)}}{d\lambda}(x) \cdot \mathbb{1}_{(0,\infty)}(x) \log(\frac{d\mathbb{P}_{X|(0,\infty)}}{d\lambda}(x) \cdot \mathbb{1}_{(0,\infty)}(x)) d\lambda(x) \tag{F.181}$$

$$\quad - \int d \cdot \frac{d\mathbb{P}_{X|(0,\infty)}}{d\lambda}(x) \cdot \mathbb{1}_{(0,\infty)}(x) \log(d) d\lambda(x) \tag{F.182}$$

$$= d \cdot \mathbb{H}_1(\mathbb{P}_{X|(0,\infty)}) - d \log(d) \tag{F.183}$$

Combing the terms together, we have

$$\mathbb{H}_\nu(Z) = d \cdot \mathbb{H}_1(\mathbb{P}_{X|(0,\infty)}) - d\log(d) - (1 - d)\log(1 - d) \tag{F.184}$$

$$= d \cdot \mathbb{H}_1(\mathbb{P}_{X|(0,\infty)}) + d\log\left(\frac{1}{d}\right) + (1 - d)\log\left(\frac{1}{d - 1}\right) \tag{F.185}$$

By Definition F.3, the information dimension $d(Z) = d$. Notice how $\mathbb{H}_0(\delta_0) = 0$. So we have

$$\mathbb{H}_\nu(Z) = d(Z) \cdot \mathbb{H}_1(\mathbb{P}_{X|(0,\infty)}) + d(Z)\log\left(\frac{1}{d(Z)}\right) + (1 - d(Z))\log\left(\frac{1}{d(Z) - 1}\right) = H_{d(Z)}(Z). \tag{F.186}$$

$\square$

### F.5. Generalization of the Entropy Decomposition of Total Correlation

The standard definition of total correlation for the the random vector $\mathbf{x} = (\mathbf{x}_1, \ldots, \mathbf{x}_D) \sim \mathbb{P}_{\mathbf{x}}$ in $D$ dimensions is

$$\text{TC}(\mathbf{x}) = D_{\text{KL}}\left(\mathbb{P}_{\mathbf{x}} \middle\| \prod_{i=1}^{D} \mathbb{P}_{\mathbf{x}_i}\right) = \int \log\left(\frac{d\mathbb{P}_{\mathbf{x}}}{\prod_{i=1}^{D} \mathbb{P}_{\mathbf{x}_i}}\right) d\mathbb{P}_{\mathbf{x}} \tag{F.187}$$

which only involves the joint probability measure and the product of marginal probability measures. However, when it comes to the entropy decomposition of the total correlation, we know that

$$\text{TC}(\mathbf{x}) = \sum_{i=1}^{D} \mathbb{H}_\lambda(\mathbf{x}_i) - \mathbb{H}_{\lambda^{\otimes D}}(\mathbf{x}) \tag{F.188}$$

where $\lambda$ is the Lebesgue measure over $\mathbb{R}$ and $\lambda^{\otimes D}$ is the Lebesgue measure over $\mathbb{R}^d$. For our purposes, we adopt the decomposition

$$\text{TC}(\mathbf{x}) = \sum_{i=1}^{D} \mathbb{H}_\nu(\mathbf{x}_i) - \mathbb{H}_{\nu^{\otimes D}}(\mathbf{x}) \tag{F.189}$$

where $\nu := \delta_0 + \lambda$ and $\mathbb{H}_\nu$ is defined in Lemma F.6 and is equivalent to the $d$-dimensional entropy. So we have

$$\text{TC}(\mathbf{x}) = \sum_{i=1}^{D} \mathbb{H}_{d(\mathbf{x}_i)}(\mathbf{x}_i) - \mathbb{H}_{d(\mathbf{x})}(\mathbf{x}) \tag{F.190}$$

## G. Hilbert-Schmidt Independence Criterion

For two random variables $X, Y$ with empirical samples $\mathbf{x}, \mathbf{y} \in \mathbb{R}^{B \times 1}$, the empirical Hilbert-Schmidt Independence Criterion (HSIC) is given by

$$\text{HSIC}(X, Y) = \frac{1}{(B-1)^2} \text{Tr}(\mathbf{KHLH}) \tag{G.191}$$

where $\mathbf{K}_{ij} = k(\mathbf{x}_i, \mathbf{x}_j)$, $\mathbf{L}_{ij} = l(\mathbf{x}_i, \mathbf{x}_j)$ for some kernels $k, l$, $\mathbf{H} := \mathbf{I} - (1/B) \cdot \mathbf{11}^\top$ is the centering matrix, and $\text{Tr}(\cdot)$ is the trace operator (Gretton et al., 2005). We denote the normalized HSIC as

$$\text{nHSIC}(X, Y) = \frac{\text{HSIC}(X, Y)}{\sqrt{\text{HSIC}(X, X) \cdot \text{HSIC}(Y, Y)}} \tag{G.192}$$

Both HSIC and nHSIC capture nonlinear dependencies beyond second-order statistics and thus serve as a proxy for measuring statistical independence beyond inspecting the covariance matrix.

For a feature random vector $\mathbf{z} \in \mathbb{R}^d$, we can obtain the nHSIC matrix by computing all pair-wise $\text{nHSIC}(\mathbf{z}_i, \mathbf{z}_j)$. In our experiments, we report the average of the off-diagonals of the nHSIC matrix in Figure 5b. Following Mialon et al. (2022), we pick the Gaussian kernel where the bandwidth parameter $\sigma$ is the median of pairwise $\ell_2$ distances between samples. Due to the presence of rectifications, we set $\sigma$ to be the standard deviation of the positive activations.

## H. Baseline Designs

**CL and NCL**. We denote SimCLR (Chen et al., 2020) as CL. Non-Negative Contrastive Learning (NCL) (Wang et al., 2024) essentially applies the SimCLR loss over rectified features and thus is a sparse variant of contrastive learning.

**VICReg and NVICReg**. VICReg (Bardes et al., 2022) minimizes the $\ell_2$ distance between the features of semantically related views while regularizing the empirical feature covariance matrix towards scalar times identity $\gamma \cdot \mathbf{I}$. We design a sparse version of VICReg, which we call Non-Negative VICReg (NVICReg), that applies the same VICReg loss over rectified features.

**ReLU and RepReLU**. Let $z \in \mathbb{R}$. NCL (Wang et al., 2024) adopts a reparameterization of the standard rectified non-linearity as

$$\text{RepReLU}(z) := \text{ReLU}(z).\text{detach}() + \text{GeLU}(z) - \text{GeLU}(z).\text{detach}() \tag{H.193}$$

where $\text{detach}()$ blocks gradient flow. The $\text{RepReLU}(\cdot)$ is equivalent to $\text{ReLU}(\cdot)$ in the forward pass but allows gradient for negative entries. For NCL and NVICReg, we use NCL-ReLU and NVICReg-ReLU to denote usage of $\text{ReLU}(\cdot)$ and NCL-RepReLU and NVICReg-RepReLU when using $\text{RepReLU}(\cdot)$.

For our Rectified LpJEPA, we note that we just use $\mathrm{ReLU}(\cdot)$ to avoid extra hyperparameter tuning. We defer detailed investigations on the activation functions to future work.

**LpJEPA and LeJEPA**. Rectified LpJEPA regularizes rectified features towards Rectified Generalized Gaussian distributions. We also design LpJEPA, which regularize non-rectified features towards the Generalized Gaussian distributions using the same projections-based distribution matching loss. When $p = 2$, LpJEPA reduces to LeJEPA (Balestriero & LeCun, 2025), since $\mathcal{GN}_2(\mu, \sigma) = \mathcal{N}(\mu, \sigma^2)$. For $0 < p \leq 1$, LpJEPA is still penalizing the $\|\cdot\|_p^p$ norms of the features and thus serves as another set of sparse baselines even though all entries are non-zero.

# I. Connection to Non-Negative VCReg

The Cramér–Wold device motivates matching the feature distribution to the Rectified Generalized Gaussian target $\prod_{i=1}^d \mathcal{RGN}_p(\mu, \sigma)$, whose coordinates are i.i.d. in the population target. Prior work such as VICReg (Bardes et al., 2022) demonstrates that explicitly removing second-order dependencies via covariance regularization is already sufficient to prevent representational collapse in practice. This motivates us to investigate whether RDMReg likewise controls second-order dependencies more explicitly.

In Proposition I.1, we show a conditional covariance result: if the projected feature distribution matches the RGG target along the eigenvectors of the centered feature covariance matrix, then the centered covariance is isotropic. This statement clarifies the relationship to Non-Negative VCReg, but it is not a claim that finite random projections exactly recover VCReg.

**Proposition I.1** (Implicit Regularization of Second-Order Statistics). *Let $\mathbf{z} \in \mathbb{R}^d$ be a neural network feature random vector with covariance matrix $\mathrm{Cov}[\mathbf{z}] = \boldsymbol{\Sigma}$. We denote the eigendecomposition as $\boldsymbol{\Sigma} = \mathbf{U}\boldsymbol{\Lambda}\mathbf{U}^\top$ with the set of eigenvectors being $\{\mathbf{u}_i\}_{i=1}^d$. Let $\mathbf{y} \sim \prod_{i=1}^d \mathcal{RGN}_p(\mu, \sigma)$ be the Rectified Generalized Gaussian random vector and define $\gamma := \mathrm{Var}[\mathcal{RGN}_p(\mu, \sigma)] \in (0, \infty)$. If $\mathbf{u}_i^\top \mathbf{z} \stackrel{d}{=} \mathbf{u}_i^\top \mathbf{y}$ for all $i \in \{1, \ldots, d\}$, then $\boldsymbol{\Sigma} = \gamma \cdot \mathbf{I}_d$.*

*Proof.* Let $\boldsymbol{\Sigma} = \mathrm{Cov}[\mathbf{z}]$ and let $\boldsymbol{\Sigma} = \mathbf{U}\boldsymbol{\Lambda}\mathbf{U}^\top$ be its eigendecomposition, where $\mathbf{U} = [\mathbf{u}_1, \ldots, \mathbf{u}_d]$ is orthonormal and $\boldsymbol{\Lambda} = \mathrm{diag}(\lambda_1, \ldots, \lambda_d)$. Since $\mathbf{y} \sim \prod_{i=1}^d \mathcal{RGN}_p(\mu, \sigma)$ has i.i.d. coordinates with variance $\gamma := \mathrm{Var}[\mathcal{RGN}_p(\mu, \sigma)]$, its covariance satisfies $\mathrm{Cov}[\mathbf{y}] = \gamma \mathbf{I}_d$. Hence, for any vector $\mathbf{u}_i$ such that $\|\mathbf{u}_i\|_2 = 1$,

$$\mathrm{Var}(\mathbf{u}_i^\top \mathbf{y}) = \mathbf{u}_i^\top (\gamma \mathbf{I}_d) \mathbf{u}_i = \gamma \cdot \|\mathbf{u}_i\|_2^2 = \gamma. \tag{I.194}$$

By the assumption $\mathbf{u}_i^\top \mathbf{z} \stackrel{d}{=} \mathbf{u}_i^\top \mathbf{y}$ for all $i$, the variances of the one-dimensional projected marginals are equal, i.e.

$$\mathrm{Var}(\mathbf{u}_i^\top \mathbf{z}) = \mathrm{Var}(\mathbf{u}_i^\top \mathbf{y}) = \gamma \qquad \forall i. \tag{I.195}$$

On the other hand, for each eigenvector $\mathbf{u}_i$,

$$\mathrm{Var}(\mathbf{u}_i^\top \mathbf{z}) = \mathbf{u}_i^\top \boldsymbol{\Sigma} \mathbf{u}_i = \lambda_i \cdot \|\mathbf{u}_i\|_2^2 = \lambda_i, \tag{I.196}$$

where $\lambda_i$ is the $i$-th eigenvalue of $\boldsymbol{\Sigma}$. Therefore $\lambda_i = \gamma$ for all $i$, so $\boldsymbol{\Lambda} = \gamma \mathbf{I}_d$.

Substituting back into the eigendecomposition yields

$$\boldsymbol{\Sigma} = \mathbf{U}\boldsymbol{\Lambda}\mathbf{U}^\top = \mathbf{U}(\gamma \mathbf{I}_d)\mathbf{U}^\top = \gamma \mathbf{I}_d, \tag{I.197}$$

which is a scalar multiple of the identity. Hence all off-diagonal entries of $\boldsymbol{\Sigma}$ vanish and the covariance matrix is isotropic. $\square$

Thus, matching the target projected marginals along covariance eigenvectors is sufficient to control the centered covariance matrix of neural network features. In practice, we always have $B \ll D$. Thus truncated SVD has $\mathcal{O}(B^2 D)$ complexity using dense methods (Golub & Van Loan, 2013), or $\mathcal{O}(BDk)$ when computing only the top-$k$ eigenvectors via Lanczos or randomized methods (Parlett, 1998; Halko et al., 2011).

Since our feature matrix $\mathbf{Z} \in \mathbb{R}^{B \times D}$ is obtained after rectifications, this gives a non-negative analogue of covariance regularization when eigenvector projections are used: it encourages non-negative neural network features to have isotropic centered covariance. This second-order view is related to, but weaker than, exact recovery of the full VICReg objective. If one uses an uncentered second-moment penalty instead of centered covariance, the corresponding empirical objective can be written in a non-negative matrix-factorization-like form (Lee & Seung, 2000):

$$\|\gamma \cdot \mathbf{I}_d - \tilde{\mathbf{Z}}^\top \tilde{\mathbf{Z}}\|_F^2, \tag{I.198}$$

where $\tilde{\mathbf{Z}} := (1/\sqrt{B}) \cdot \mathbf{Z}$ remains non-negative for rectified features. We therefore distinguish this uncentered NMF interpretation from the centered covariance statement in Proposition I.1. Non-Negative Contrastive Learning (NCL) (Wang et al., 2024) shows that applying SimCLR loss over rectified features recovers a form of NMF over the rescaled variant of the Gram matrix. Based on the Gram-Covariance matrix duality between contrastive and non-contrastive learning (Garrido et al., 2022), we observe a similar duality in NMF and defer the detailed investigations to future work.

# J. Additional Experimental Results

In the following section, we include additional experimental results for evaluating our Rectified LpJEPA methods.

## J.1. Linear Probe over CIFAR-100

In Table 5, we report linear probe performances of Rectified LpJEPA and other dense and sparse baselines over CIFAR-100. Rectified LpJEPA achieves competitive sparsity-performance tradeoffs.

## J.2. Ablations on Projector Dimensions

In Table 12, we compare VICReg, LeJEPA, and Rectified LpJEPA with varying projector dimensions. We observe that Rectified LpJEPA consistently attains competitive or better performances.

## J.3. Rectified LpJEPA with ViT Backbones

We evaluate whether the strong performance of Rectified LpJEPA with a ResNet backbone shown in Table 1 generalizes across encoder architectures. As shown in Table 13, Rectified LpJEPA remains competitive when instantiated with a ViT encoder.

## J.4. ImageNet-1K Experiments

We further evaluate Rectified LpJEPA in a larger-scale ImageNet-1K pretraining setting. Due to computational constraints, we run a controlled 100-epoch comparison rather than an extensive 1000-epoch hyperparameter sweep. We compare VICReg against Rectified LpJEPA with $p = 1.0$, $\mu = 0$, and $\sigma = \sigma_{\mathrm{GN}}$, using either random projection vectors only or a mixture of random projections and eigenvectors of the feature covariance matrix.

As shown in Table 2, Rectified LpJEPA with random projections only underperforms VICReg on encoder accuracy after 100 epochs, while the eigenvector-augmented variant slightly improves over VICReg on encoder accuracy and substantially improves projector accuracy. This supports our observation in Appendix J.8 that, in short training regimes, random-projection RDMReg may spend optimization effort reducing higher-order dependencies without explicitly prioritizing second-order covariance structure. Incorporating eigenvector projections provides a direct second-order signal while retaining the distribution-matching objective over random projections.

*Table 2*. **ImageNet-1K Linear Probe Results after 100 Epochs.** We report encoder and projector top-1 accuracy (%). Rectified LpJEPA uses $\mathcal{RGN}_{1.0}(0, \sigma_{\mathrm{GN}})$ as the target distribution.

| Method | Encoder Acc1 $\uparrow$ | Projector Acc1 $\uparrow$ |
|---|---|---|
| VICReg | 60.52 | 46.23 |
| $\mathcal{RGN}_{1.0}(0, \sigma_{\mathrm{GN}})$ | 57.95 | 46.07 |
| $\mathcal{RGN}_{1.0}(0, \sigma_{\mathrm{GN}})$ + Eigenvector Proj. | **60.63** | **48.84** |

## J.5. Batch Size Dependency of RDMReg

RDMReg is a two-sample sliced distribution-matching objective, so its empirical estimate depends on the batch size used to approximate one-dimensional Wasserstein distances. We therefore evaluate the sensitivity of Rectified LpJEPA to the pretraining batch size on CIFAR-100. As shown in Table 3, performance improves rapidly at small batch sizes and begins to saturate around batch sizes 128–256. This behavior is consistent with the design goal of avoiding the large-batch negative-sample requirements of contrastive objectives while still obtaining stable distribution-matching estimates.

*Table 3*. **Batch Size Dependency of RDMReg on CIFAR-100.** We report mean $\pm$ standard deviation of encoder and projector top-1 accuracy (%).

| Batch Size | 24 | 32 | 48 | 64 | 128 | 256 | 512 |
|---|---|---|---|---|---|---|---|
| Enc Acc1 $\uparrow$ | 64.88±0.17 | 66.01±0.44 | 67.47±0.17 | 67.59±0.49 | 67.69±0.55 | 68.29±0.30 | 67.88±0.10 |
| Proj Acc1 $\uparrow$ | 56.19±0.07 | 58.77±0.59 | 61.11±0.22 | 62.24±0.10 | 63.54±0.10 | 64.41±0.22 | 64.16±0.12 |

## J.6. Runtime Comparison between SIGReg and RDMReg

We compare the wall-clock cost of SIGReg and RDMReg under the same CIFAR-100 pretraining setup. As shown in Table 4, RDMReg is only slightly slower than SIGReg in end-to-end training time and uses comparable GPU memory. This suggests that, at the batch sizes and projection counts used in our experiments, the sorting operation required by the sliced Wasserstein distance is not a dominant bottleneck relative to the shared projection and neural-network computation.

*Table 4.* **Runtime Comparison between SIGReg and RDMReg.** Both methods are trained for 100 epochs on CIFAR-100.

| Method | Training Time (min) ↓ | Avg. GPU Util. (%) ↑ | GPU Memory |
|--------|----------------------|----------------------|------------|
| SIGReg | 19.02 | 46.59 | 2.65 GB |
| RDMReg | 19.27 | 45.04 | 2.63 GB |

*Table 5.* **Linear Probe Results on CIFAR-100.** Acc1 (%) is higher-is-better (↑); sparsity is lower-is-better (↓). **Bold** denotes best and underline denotes second-best in each column (ties allowed).

| | | Encoder Acc1 ↑ | Projector Acc1 ↑ | L1 Sparsity ↓ | L0 Sparsity ↓ |
|--|--|----------------|------------------|---------------|---------------|
| **Rectified LpJEPA** | $\mathcal{RGN}_{2.0}(0, \sigma_{\text{GN}})$ | 66.29 | 62.15 | 0.3773 | 0.7357 |
| | $\mathcal{RGN}_{1.0}(0, \sigma_{\text{GN}})$ | 65.97 | 62.22 | 0.3019 | 0.6474 |
| | $\mathcal{RGN}_{0.75}(0, \sigma_{\text{GN}})$ | 65.78 | **62.80** | 0.2583 | 0.6099 |
| | $\mathcal{RGN}_{0.50}(0, \sigma_{\text{GN}})$ | 66.10 | 62.74 | 0.1996 | 0.5727 |
| | $\mathcal{RGN}_{1.0}(-2, \sigma_{\text{GN}})$ | 64.75 | 59.08 | 0.0236 | 0.0489 |
| **Sparse Baselines** | NCL-ReLU | 66.23 | 61.88 | 0.0228 | 0.0503 |
| | NVICReg-ReLU | 63.76 | 58.82 | 0.7415 | 0.8935 |
| | NCL-RepReLU | **66.32** | 61.40 | **0.0202** | **0.0426** |
| | NVICReg-RepReLU | 63.83 | 58.53 | 0.1551 | 0.2657 |
| **Dense Baselines** | SimCLR | 66.00 | 61.95 | 0.6364 | 1.0000 |
| | VICReg | 63.78 | 58.82 | 0.8660 | 1.0000 |
| | LeJEPA | 65.65 | 62.69 | 0.6379 | 1.0000 |

## J.7. Big-O Efficiency Analysis

Let $B$ denote batch size, $D$ feature dimension, $P$ number of random projections, and $K$ number of frequency/grid evaluations used by SIGReg. We compare only the distribution-matching components, ignoring shared operations such as drawing and normalizing random projection vectors.

RDMReg consists of sampling from the target RGG distribution, projecting both features and target samples, sorting projected samples along the batch dimension for each projection, and reducing the resulting one-dimensional Wasserstein distances. These steps have complexity

$$\mathcal{O}(BD) + \mathcal{O}(BDP) + \mathcal{O}(PB \log B) + \mathcal{O}(BP), \tag{J.199}$$

so the dominant cost is

$$\mathcal{O}(BDP + PB \log B). \tag{J.200}$$

SIGReg shares the projection cost and additionally evaluates a characteristic-function discrepancy over a fixed grid, giving

$$\mathcal{O}(BDP + BPK), \tag{J.201}$$

with dominant projection cost $\mathcal{O}(BDP)$ when $K$ is treated as a fixed constant.

Thus, RDMReg and SIGReg share the same leading projection cost $\mathcal{O}(BDP)$, while RDMReg incurs an additional $\mathcal{O}(PB \log B)$ sorting term. In practice, our batch-size study in Table 3 indicates that RDMReg does not require very large batches, and the end-to-end comparison in Table 4 suggests that this sorting term is modest in our implementation. A limitation of both implementations is that several projection and reduction operations are not fused kernels, so optimized kernels could further reduce memory movement and wall-clock overhead.

## J.8. Additional Results on Eigenvectors

We would like to know whether incoporating the eigenvectors of the empirical feature covariance matrices into the projection directions for RDMReg can lead to faster convergence and directly removes second-order dependencies. To this end, we pretrain Rectified LpJEPA with RDMReg and log the variance and covariance loss defined in VICReg (Bardes et al., 2022).

The variance loss computes the $\ell_2$ distance between the diagonal of the empirical feature covariance matrix and the theoretical variance of the Rectified Generalized Gaussian distribution as we derived in Proposition B.9. The covariance loss is simply the sum of the off-diagonal entries of the empirical feature covariance matrix scaled by $1/D$, where $D$ is the feature dimension. We emphasize that we don't incorporate the variance and covariance losses into optimizations but only use them as evaluation metrics.

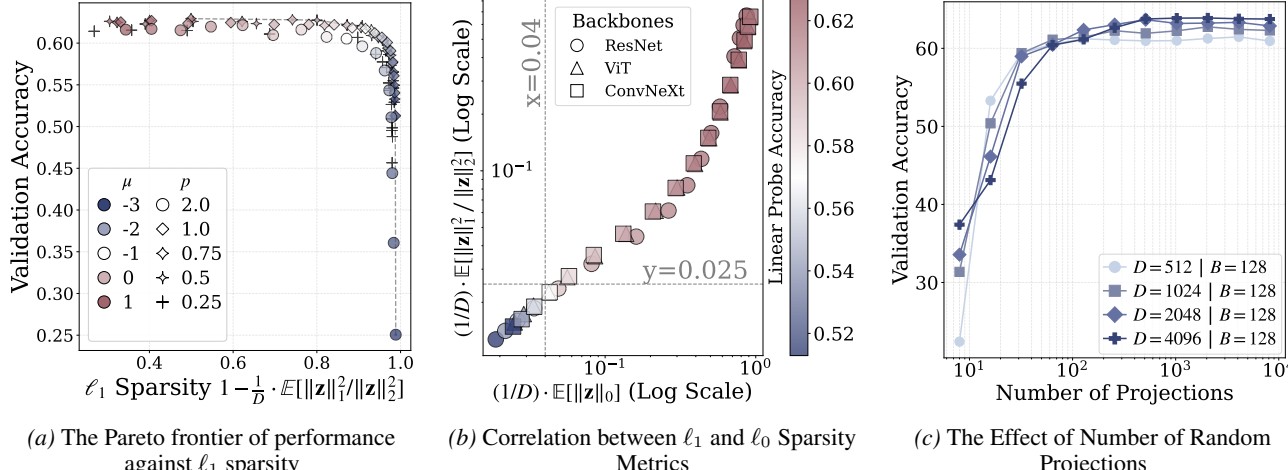

*(a)* The Pareto frontier of performance against $\ell_1$ sparsity

*(b)* Correlation between $\ell_1$ and $\ell_0$ Sparsity Metrics

*(c)* The Effect of Number of Random Projections

*Figure 14.* **Additional results on the sparsity-performance tradeoffs, the correlation between different sparsity metrics, and the effect of numbers of random projections on performance**. (a) We present another version of Figure 3c where the sparsity metric is switched from $\ell_0$ to $\ell_1$. Again, we observe the same Pareto frontier with a sharp drop in performance under extreme sparsity. (b) Across different backbones, Rectified LpJEPAs with the target distribution being the Rectified Laplace distributions learn sparser representations as we decrease the mean shift value $\mu$. Specifically, we observe that the $\ell_0$ and $\ell_1$ metrics are quite correlated. Thus both metrics are effective in measuring sparsity. (c) We test Rectified LpJEPA models with batch size $B = 128$ and varying feature dimension $D$ as long as numbers of projections. As we increase the dimensions $D$, we observe that the number of random projections required for good performance do not grow and are small relative to $D$. Hence Rectified LpJEPA maintains stable sampling efficiency as the feature dimension grows in these experiments.

In Figure 15, we show that incorporating eigenvectors indeed leads to faster convergence and better performance and also signficant reductions in the variance and covariance loss. This further validates our observations on the Non-Negative VCReg recovery (Proposition I.1) of the RDMReg loss as we observe significant reductions in second-order dependencies.

## J.9. Additional Results on Transfer Sparsity

In Figure 13, we plot the $\ell_0$ and $\ell_1$ sparsity metrics for baselines and Rectified LpJEPA across different downstream transfer tasks. We observe that Rectified LpJEPA exhibits larger variations in the sparsity values across datasets, indicating that sparsity can be used as a crude proxy for whether the task at hand is within the training distribution.

In Figure 20, we further probe whether the sparsity metric can be used as a signal for whether groups of inputs are correctly or incorrectly classified by the model. We observe that this is partially true when the inputs are from the pretraining dataset. The distribution of the $\ell_1$ sparsity metrics is distinct between correctly and incorrectly classified examples. The divergence is less prominent for downstream transfer tasks, and we defer further investigations to future work.

# K. Qualitative Analysis of Rectified LpJEPA

In the following section, we present qualitative analyses of Rectified LpJEPA models and baseline models pretrained over ImageNet-100. We use the target distribution $\mathcal{RGN}_p(\mu, \sigma_{GN})$ to denote Rectified LpJEPA with hyperparameters $\{\mu, \sigma_{GN}, p\}$. In Appendix K.1, we visualize nearest-neighbor retrieval of selected exemplar images in representation space. We present additional visual attribution maps in Appendix K.2.

## K.1. $k$-Nearest Neighbors Visualizations

For a selected exemplar image, we retrieve its top-$k$ nearest neighbors from the ImageNet-100 validation set using cosine similarity over frozen projector features. Retrieved neighbors are outlined in green if their labels match the exemplar's class and in red otherwise.

In Figure 16, we visualize the top-7 nearest neighbors of a pirate ship exemplar for Rectified LpJEPA with $p = 1$ under varying mean-shift values $\mu$, alongside dense and sparse baseline models. Despite substantial variation in feature sparsity induced by $\mu$, Rectified LpJEPA consistently retrieves semantically coherent neighbors: across all settings, the retrieved images belong exclusively to the pirate ship class. Combined with the competitive linear-probe performance reported in Table 1, these qualitative results suggest that Rectified LpJEPA preserves semantic structure even in highly sparse regimes.

We next consider a more challenging exemplar depicting a tabby cat in the foreground against a laptop background. As shown in Figure 17, dense baselines such as SimCLR (denoted CL for brevity) retrieve a mixture of cat and laptop images, indicating label-agnostic encodings

*Table 6.* **1-shot linear probe accuracy (%) using *encoder* features.** All results are in the 1-shot setting. Avg. denotes the mean across datasets.

| model | DTD | cifar10 | cifar100 | flowers | food | pets | avg. |
|---|---|---|---|---|---|---|---|
| Non-negative VICReg | 11.86 | 65.62 | 24.95 | 5.09 | 24.82 | 29.74 | 27.01 |
| Non-negative SimCLR | 11.17 | 67.67 | 24.23 | 5.59 | 24.44 | 19.71 | 25.47 |
| **Our methods** | | | | | | | |
| $\mathcal{RGN}_{1.0}(0, \sigma_{\text{GN}})$ | 9.89 | 68.31 | **26.27** | 4.21 | **25.98** | 17.66 | 25.39 |
| $\mathcal{RGN}_{1.0}(-1, \sigma_{\text{GN}})$ | 9.47 | 67.19 | 23.58 | 4.85 | 24.93 | 16.60 | 24.44 |
| $\mathcal{RGN}_{1.0}(-2, \sigma_{\text{GN}})$ | 12.66 | 66.36 | 23.91 | **10.18** | 24.88 | 25.18 | **27.20** |
| $\mathcal{RGN}_{1.0}(-3, \sigma_{\text{GN}})$ | 9.31 | 50.39 | 8.35 | 6.13 | 10.63 | 21.72 | 17.76 |
| $\mathcal{RGN}_{2.0}(0, \sigma_{\text{GN}})$ | **14.04** | **69.47** | 24.09 | 4.86 | 25.68 | 24.34 | 27.08 |
| $\mathcal{RGN}_{2.0}(-1, \sigma_{\text{GN}})$ | 12.82 | 65.97 | 24.12 | 4.91 | 24.44 | 20.20 | 25.41 |
| $\mathcal{RGN}_{2.0}(-2, \sigma_{\text{GN}})$ | 13.88 | 59.62 | 15.88 | 7.84 | 16.21 | 24.97 | 23.07 |
| $\mathcal{RGN}_{2.0}(-3, \sigma_{\text{GN}})$ | 11.06 | 55.71 | 11.42 | 8.18 | 12.92 | 12.13 | 18.57 |
| **Dense baselines** | | | | | | | |
| VICReg | 10.85 | 63.98 | 21.52 | 6.23 | **25.29** | 23.33 | 25.20 |
| SimCLR | 12.82 | 66.90 | 23.93 | **10.60** | 24.99 | 24.12 | **27.23** |
| LeJEPA | **15.85** | 68.07 | **24.57** | 5.79 | 23.72 | **24.34** | 27.06 |

that capture multiple objects present in the scene. In contrast, both sparse baselines and Rectified LpJEPA predominantly retrieve images of cats, except in the extreme sparsity setting of Rectified LpJEPA with $\mu = -3$. In this regime, the retrieved neighbors consist almost exclusively of laptop images. This behavior suggests that under extreme sparsity, either information about the cat is lost or the background laptop features dominate over the cat features.

To distinguish between these possibilities, we further probe Rectified LpJEPA with $\mu = -3$ by cropping the exemplar image to retain only the cat foreground. As shown in Figure 18, once the background is removed, Rectified LpJEPA with $\mu = -3$ consistently retrieves images of cats. This shows that even under extreme sparsity, Rectified LpJEPA still preserves information instead of being a lossy compression of the input, and we hypothesize that retrieval of solely laptop images in Figure 17 is due to competitions between features in the scene rather than information loss.

## K.2. Visual Attribution Maps

To further support the above observations, we visualize attribution maps for the tabby cat exemplar and its cropped variant using Grad-CAM-style heatmaps (Selvaraju et al., 2019). Specifically, we backpropagate a scalar score derived from the representation—the squared $\ell_2$ norm of the projector feature—to a late backbone layer, and compute a weighted combination of activations that is overlaid on the input image.

As shown in Figure 19, when the background laptop is removed, all models concentrate their attributions on the cat, consistent with the cat-only retrieval behavior observed in Figure 18. For the full image containing both the cat and the laptop, attributions are more spatially spread-out. Notably, Rectified LpJEPA with $\mu = -3$ places a large fraction of its attribution mass on the background, aligning with its tendency to retrieve laptop images in Figure 17.

Taken together, these results suggest that, for the examined examples, Rectified LpJEPA can perform task-agnostic encoding under extreme sparsity without simply discarding the relevant object information. This behavior aligns with our objective of learning sparse representations whose continuous components retain high entropy, encouraging the representation to preserve input information while remaining agnostic to downstream tasks.

## L. Implementation Details

### L.1. Pretraining data and setup

We conduct self-supervised pretraining on ImageNet-100, a 100-class subset derived from ImageNet-1K (da Costa et al., 2022), using a ResNet-50 encoder. Unless otherwise specified, all methods are trained with identical data, architecture, optimizer, and augmentation pipelines to ensure fair comparison.

### L.2. Architecture

The encoder is followed by a 3-layer MLP projector with hidden and output dimension 2048. For Rectified LpJEPA variants, denoted $\mathcal{RGN}_p(\mu, \sigma_{\text{GN}})$, we append a final $\text{ReLU}(\cdot)$ to the projector output to enforce explicit rectifications in the representation space. When

*Table 7.* **1-shot linear probe accuracy (%) using *projector* features.** All results are in the 1-shot setting. Avg. denotes the mean across datasets.

| model | DTD | cifar10 | cifar100 | flowers | food | pets | avg. |
|---|---|---|---|---|---|---|---|
| Non-negative VICReg | 8.62 | 33.47 | 7.51 | 3.25 | 8.78 | 17.28 | 13.15 |
| Non-negative SimCLR | 6.70 | 41.71 | 9.24 | 2.91 | 9.17 | 15.37 | 14.18 |
| **Our methods** | | | | | | | |
| $\mathcal{RGN}_{1.0}(0, \sigma_{\mathrm{GN}})$ | 6.81 | **48.96** | 11.46 | 2.03 | **12.09** | 16.38 | 16.29 |
| $\mathcal{RGN}_{1.0}(-1, \sigma_{\mathrm{GN}})$ | 7.82 | 47.14 | 11.20 | 2.75 | 11.43 | 14.94 | 15.88 |
| $\mathcal{RGN}_{1.0}(-2, \sigma_{\mathrm{GN}})$ | 10.37 | 38.36 | 9.13 | **5.56** | 9.53 | **20.93** | 15.65 |
| $\mathcal{RGN}_{1.0}(-3, \sigma_{\mathrm{GN}})$ | 5.69 | 32.25 | 5.16 | 2.70 | 5.76 | 19.02 | 11.76 |
| $\mathcal{RGN}_{2.0}(0, \sigma_{\mathrm{GN}})$ | **10.59** | 47.74 | 10.61 | 2.70 | 11.46 | 20.50 | **17.27** |
| $\mathcal{RGN}_{2.0}(-1, \sigma_{\mathrm{GN}})$ | 9.84 | 46.97 | **12.33** | 3.07 | 11.70 | 19.38 | 17.21 |
| $\mathcal{RGN}_{2.0}(-2, \sigma_{\mathrm{GN}})$ | 9.57 | 35.08 | 6.59 | 4.49 | 7.41 | 20.69 | 13.97 |
| $\mathcal{RGN}_{2.0}(-3, \sigma_{\mathrm{GN}})$ | 9.20 | 18.36 | 2.64 | 2.33 | 3.64 | 7.14 | 7.22 |
| **Dense baselines** | | | | | | | |
| VICReg | 8.09 | 39.35 | 6.74 | 3.04 | 9.34 | 19.49 | 14.34 |
| SimCLR | **12.39** | 48.05 | **11.52** | **7.24** | **14.16** | **21.34** | **19.12** |
| LeJEPA | 8.30 | **48.17** | 11.27 | 4.47 | 11.84 | 18.32 | 17.06 |

$p = 1$, our target distribution is Rectified Laplace. For $p = 2$, the RDMReg loss is matching to the Rectified Gaussian distribution.

## L.3. Data augmentation

Following the standard protocol in da Costa et al. (2022), we generate two stochastic views per image using: random resized crop (scale in $[0.2, 1.0]$, output resolution $224 \times 224$), random horizontal flip ($p = 0.5$), color jitter ($p = 0.8$; brightness 0.4, contrast 0.4, saturation 0.2, hue 0.1), random grayscale ($p = 0.2$), Gaussian blur ($p = 0.5$), and solarization ($p = 0.1$).

## L.4. Optimization

We pretrain for 1000 epochs using LARS optimizer (You et al., 2017) with a warmup+cosine learning rate schedule (10 warmup epochs). Unless otherwise specified, we use batch size 128, learning rate 0.0825 for the encoder, learning rate 0.0275 for the linear classifier, momentum 0.9, and weight decay $10^{-4}$. All ImageNet-100 experiments are run on a single node with a single NVIDIA L40S GPU.

## L.5. Distribution matching objective

For Rectified LpJEPA, we set the invariance weight to $\lambda_{\mathrm{sim}} = 25.0$ and the RDMReg loss weight to $\lambda_{\mathrm{dist}} = 125.0$. We perform distribution-matching using the sliced 2-Wasserstein distance (SWD) with 8192 random projections per iteration.

## L.6. Compute and runtime

A full 1000-epoch ImageNet-100 pretraining run takes approximately 2d 7h wall-clock time on a single NVIDIA L40S GPU. To speed-up training, we pre-load the entire ImageNet-100 dataset to CPU memory to avoid additional I/O costs. We find that this significantly improves GPU utilization and also minimizes the communication time. However, we're not able to do this for larger-scale datasets.

## L.7. Transfer evaluation protocol

We evaluate transfer performance with frozen-feature linear probing on six downstream datasets: DTD, CIFAR-10, CIFAR-100, Flowers-102, Food-101, and Oxford-IIIT Pets. For each pretrained checkpoint, we freeze the encoder (and projector when applicable) and train a single linear classifier on top of both encoder features and projector features.

We report three label regimes: 1% (1-shot), 10% (10-shot), and 100% (All-shot) of the labeled training data. The linear probe is trained for 100 epochs using Adam with learning rate $10^{-2}$, batch size 512, and no weight decay. Evaluation inputs are resized to 256 on the shorter side, then center-cropped to $224 \times 224$ and normalized with ImageNet statistics; we apply no data augmentation during probing.

## L.8. Reproducibility.

All ImageNet-100 pretraining results are reported from a single run (seed 5), trained with mixed-precision (16-mixed) on a single NVIDIA L40S GPU (one node, one GPU; no distributed training).

*Table 8.* **10-shot linear probe accuracy (%) using *encoder* features.** All results are in the 10-shot setting. Avg. denotes the mean across datasets.

| model | DTD | cifar10 | cifar100 | flowers | food | pets | avg. |
|---|---|---|---|---|---|---|---|
| Non-negative VICReg | 37.23 | 77.88 | 47.32 | 31.16 | 48.33 | 55.49 | 49.57 |
| Non-negative SimCLR | 40.11 | 79.11 | 50.35 | 23.96 | 50.60 | 50.37 | 49.08 |
| **Our methods** | | | | | | | |
| $\mathcal{RGN}_{1.0}(0, \sigma_{\mathrm{GN}})$ | **41.91** | 79.58 | 49.67 | 24.93 | 49.97 | 55.63 | 50.28 |
| $\mathcal{RGN}_{1.0}(-1, \sigma_{\mathrm{GN}})$ | 38.19 | 78.98 | 48.76 | **32.87** | 48.51 | 55.41 | 50.45 |
| $\mathcal{RGN}_{1.0}(-2, \sigma_{\mathrm{GN}})$ | 40.32 | 79.25 | 45.86 | 22.72 | 46.51 | **56.91** | 48.60 |
| $\mathcal{RGN}_{1.0}(-3, \sigma_{\mathrm{GN}})$ | 19.63 | 66.93 | 24.45 | 8.25 | 26.12 | 32.76 | 29.69 |
| $\mathcal{RGN}_{2.0}(0, \sigma_{\mathrm{GN}})$ | 40.90 | **80.15** | **50.69** | 29.76 | **50.34** | 53.07 | **50.82** |
| $\mathcal{RGN}_{2.0}(-1, \sigma_{\mathrm{GN}})$ | 39.63 | 79.00 | 47.51 | 26.85 | 47.39 | 55.60 | 49.33 |
| $\mathcal{RGN}_{2.0}(-2, \sigma_{\mathrm{GN}})$ | 30.80 | 73.88 | 34.61 | 12.03 | 35.56 | 44.92 | 38.63 |
| $\mathcal{RGN}_{2.0}(-3, \sigma_{\mathrm{GN}})$ | 15.27 | 68.30 | 24.84 | 7.64 | 29.79 | 26.63 | 28.74 |
| **Dense baselines** | | | | | | | |
| VICReg | 38.35 | 76.25 | 45.63 | 26.52 | 48.30 | 52.19 | 47.88 |
| SimCLR | **41.70** | 77.88 | 46.87 | **31.66** | 49.27 | 49.74 | 49.52 |
| LeJEPA | 38.67 | **79.06** | **49.02** | 30.18 | **49.34** | **53.88** | **50.03** |

## L.9. Continuous mapping theorem evaluation (post-hoc ReLU probes).

For the continuous mapping theorem ablation (Figure 12b), we evaluate pretrained checkpoints by extracting frozen encoder/projector features and optionally applying a post-hoc rectification $\mathrm{ReLU}(\cdot)$ at evaluation time. We report (i) sparsity statistics computed on the validation features before and after rectification, and (ii) linear probe accuracy when training on dense (pre-ReLU) features versus rectified (post-ReLU) features.

Linear probes are trained for 100 epochs using SGD (momentum 0.9) with a cosine learning-rate schedule, learning rate $10^{-2}$, batch size 512, and weight decay $10^{-6}$. We use the same deterministic evaluation preprocessing described in Appendix L.7.

## L.10. Vision Transformer (ViT) experiments

For the ViT results in Table 13, we use a ViT-Small backbone (`vit_small`). The encoder is followed by a 3-layer MLP projector with hidden and output dimension 2048 (i.e., a 2048–2048–2048 MLP), and we apply a final $\mathrm{ReLU}(\cdot)$ to the projector output. We pretrain on ImageNet-100 for 1000 epochs using AdamW (batch size 128, learning rate $5 \times 10^{-4}$, weight decay $10^{-4}$) under the same augmentation pipeline described above. Other hypers are the same as used for ResNet-50 Experiments. A full 1000-epoch ImageNet-100 pretraining run for ViT takes approximately 2d 6h wall-clock time. Unless otherwise specified, we use mixed-precision (16-mixed) training on a single NVIDIA L40S GPU.

*Table 9.* **10-shot linear probe accuracy (%) using *projector* features.** All results are in the 10-shot setting. Avg. denotes the mean across datasets.

| model | DTD | cifar10 | cifar100 | flowers | food | pets | avg. |
|---|---|---|---|---|---|---|---|
| Non-negative VICReg | 22.50 | 43.24 | 10.56 | 12.86 | 12.65 | 32.92 | 22.46 |
| Non-negative SimCLR | 23.09 | 48.38 | 17.73 | 8.72 | 14.97 | 32.87 | 24.29 |
| **Our methods** | | | | | | | |
| $\mathcal{RGN}_{1.0}(0, \sigma_{\mathrm{GN}})$ | 24.47 | **58.06** | 22.40 | 9.97 | 19.73 | **40.17** | 29.13 |
| $\mathcal{RGN}_{1.0}(-1, \sigma_{\mathrm{GN}})$ | 25.90 | 57.61 | 22.39 | 11.40 | 21.24 | 38.18 | 29.45 |
| $\mathcal{RGN}_{1.0}(-2, \sigma_{\mathrm{GN}})$ | 23.83 | 44.69 | 15.32 | 9.19 | 14.65 | 33.31 | 23.50 |
| $\mathcal{RGN}_{1.0}(-3, \sigma_{\mathrm{GN}})$ | 18.19 | 36.39 | 9.03 | 7.46 | 8.67 | 26.17 | 17.65 |
| $\mathcal{RGN}_{2.0}(0, \sigma_{\mathrm{GN}})$ | 22.82 | 57.93 | 21.26 | 12.07 | 19.03 | 36.90 | 28.33 |
| $\mathcal{RGN}_{2.0}(-1, \sigma_{\mathrm{GN}})$ | **27.39** | 57.32 | **24.07** | 13.06 | 21.54 | 40.01 | **30.57** |
| $\mathcal{RGN}_{2.0}(-2, \sigma_{\mathrm{GN}})$ | 22.23 | 40.83 | 12.58 | 8.93 | 12.01 | 29.38 | 20.99 |
| $\mathcal{RGN}_{2.0}(-3, \sigma_{\mathrm{GN}})$ | 9.89 | 23.81 | 5.03 | 6.02 | 6.15 | 10.08 | 10.16 |
| **Dense baselines** | | | | | | | |
| VICReg | 23.09 | 41.20 | 10.24 | 10.60 | 13.55 | 32.49 | 21.86 |
| SimCLR | **33.09** | **59.56** | **25.91** | **15.68** | **28.61** | **42.00** | **34.14** |
| LeJEPA | 24.95 | 55.76 | 19.55 | 10.72 | 17.17 | 38.98 | 27.85 |

*Table 10.* **All-shot linear probe accuracy (%) using *encoder* features.** All-shot uses the full training set. Avg. denotes the mean across datasets.

| model | DTD | cifar10 | cifar100 | flowers | food | pets | avg. |
|---|---|---|---|---|---|---|---|
| Non-negative VICReg | 63.56 | 82.68 | 60.40 | 82.55 | 60.39 | 78.63 | 71.37 |
| Non-negative SimCLR | 64.68 | 84.41 | 62.95 | 84.97 | 63.32 | 76.56 | 72.82 |
| **Our methods** | | | | | | | |
| $\mathcal{RGN}_{1.0}(0, \sigma_{\mathrm{GN}})$ | **65.05** | 84.50 | 62.73 | 83.75 | **62.65** | 78.00 | 72.78 |
| $\mathcal{RGN}_{1.0}(-1, \sigma_{\mathrm{GN}})$ | 64.68 | 83.97 | 60.75 | 81.77 | 60.25 | 77.68 | 71.52 |
| $\mathcal{RGN}_{1.0}(-2, \sigma_{\mathrm{GN}})$ | 63.67 | **85.31** | 58.11 | 81.62 | 58.39 | 77.38 | 70.75 |
| $\mathcal{RGN}_{1.0}(-3, \sigma_{\mathrm{GN}})$ | 49.04 | 79.11 | 43.94 | 44.72 | 46.22 | 61.30 | 54.06 |
| $\mathcal{RGN}_{2.0}(0, \sigma_{\mathrm{GN}})$ | 64.52 | 85.06 | **64.51** | 84.09 | 62.35 | 78.25 | **73.13** |
| $\mathcal{RGN}_{2.0}(-1, \sigma_{\mathrm{GN}})$ | 64.26 | 84.21 | 59.90 | 81.59 | 59.47 | 77.65 | 71.18 |
| $\mathcal{RGN}_{2.0}(-2, \sigma_{\mathrm{GN}})$ | 57.87 | 82.19 | 49.81 | 69.38 | 51.90 | 68.93 | 63.35 |
| $\mathcal{RGN}_{2.0}(-3, \sigma_{\mathrm{GN}})$ | 53.35 | 78.95 | 45.32 | 57.47 | 46.32 | 52.22 | 55.61 |
| **Dense baselines** | | | | | | | |
| VICReg | 62.77 | 81.47 | 59.23 | 80.96 | 60.44 | 77.11 | 70.33 |
| SimCLR | **64.73** | 82.49 | 60.10 | 80.47 | **61.18** | 71.44 | 70.07 |
| LeJEPA | 63.19 | **83.54** | **62.38** | **83.07** | 61.14 | **78.30** | **71.94** |

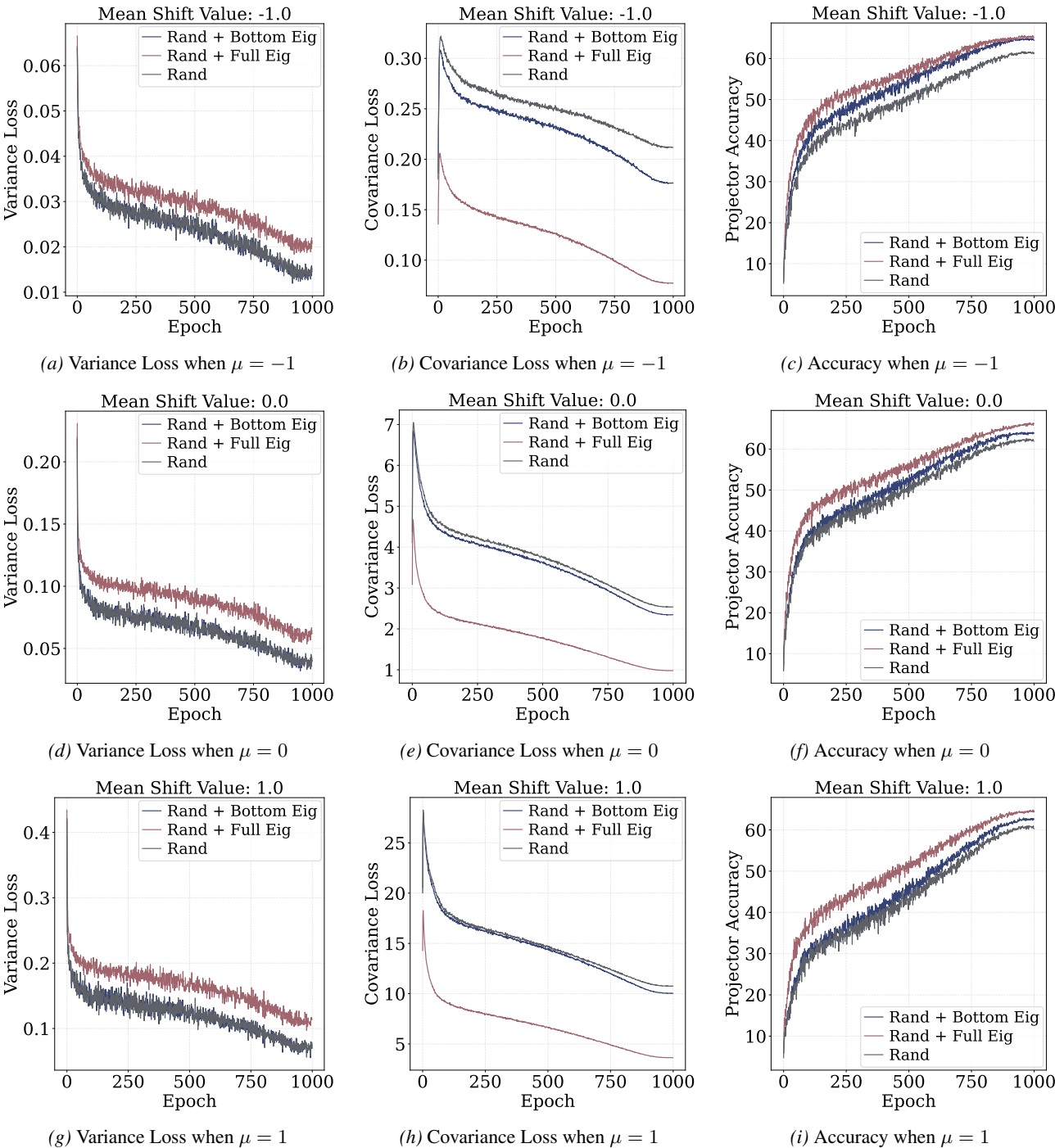

*Figure 15.* **Incorporating eigenvectors into random projections accelerates implicit VCReg loss minimization and speed-up convergence**. We pretrain Rectified LpJEPA over CIFAR-100 with target distributions $\mathcal{RGN}_1(\mu, \sigma_{\mathrm{GN}})$ where the mean shift value $\mu \in \{-1, 0, 1\}$. We consider three settings of selecting the projection vectors $\{c_i\}_{i=1}^N$ where we set $N = 8192$. We denote the setting as "Rand" if all $\mathbf{c}_i$ are uniformly sampled from the unit $\ell_2$ sphere. Since in our setting the batch size is always smaller than the feature dimension, i.e. $B < D$, we call the setting "Rand + Full Eig" if we select the top-$B$ eigenvectors and mix them with $N - B$ random projection vectors. We also consider "Rand + Bottom Eig" by sampling the bottom half $B/2$ eigenvectors and mixing with $N - B/2$ random projections. (a) (d) (g) We evaluate the variance loss between the three settings across varying $\mu$. Incorporating all eigenvectors puts overly strong constraints, whereas penalizing the bottom half eigenvectors gives us good performances. (b) (e) (h) the covariance losses are evaluated during training and plotted for different projection vector settings across $\mu$. Regularizing all top-$B$ eigenvector projections leads to significant implicit covariance loss minimization, where the covariance loss is the average of the off-diagonal of the empirical covariance matrix. (c) (f) (i) We report the projector accuracy against epoch for different projection settings. Using eigenvectors leads to both faster convergence and ultimately better performance.

*Table 11.* **All-shot linear probe accuracy (%) using *projector* features.** All-shot uses the full training set. Avg. denotes the mean across datasets.

| model | DTD | cifar10 | cifar100 | flowers | food | pets | avg. |
|---|---|---|---|---|---|---|---|
| Non-negative VICReg | 35.80 | 47.70 | 14.90 | 32.04 | 15.74 | 44.84 | 31.83 |
| Non-negative SimCLR | 36.86 | 50.24 | 21.92 | 25.57 | 17.70 | 47.02 | 33.22 |
| **Our methods** | | | | | | | |
| $\mathcal{RGN}_{1.0}(0, \sigma_{\mathrm{GN}})$ | 41.49 | 63.14 | 29.46 | 39.96 | 26.70 | 52.55 | 42.22 |
| $\mathcal{RGN}_{1.0}(-1, \sigma_{\mathrm{GN}})$ | 42.82 | 63.05 | 32.96 | 39.68 | 28.98 | 55.08 | 43.76 |
| $\mathcal{RGN}_{1.0}(-2, \sigma_{\mathrm{GN}})$ | 39.84 | 48.69 | 19.47 | 29.65 | 18.21 | 44.51 | 33.39 |
| $\mathcal{RGN}_{1.0}(-3, \sigma_{\mathrm{GN}})$ | 28.72 | 37.47 | 11.13 | 12.65 | 9.61 | 36.09 | 22.61 |
| $\mathcal{RGN}_{2.0}(0, \sigma_{\mathrm{GN}})$ | 41.81 | 62.82 | 29.31 | 37.32 | 24.80 | 53.18 | 41.54 |
| $\mathcal{RGN}_{2.0}(-1, \sigma_{\mathrm{GN}})$ | **45.00** | **63.85** | **34.13** | **42.72** | **30.30** | **56.09** | **45.35** |
| $\mathcal{RGN}_{2.0}(-2, \sigma_{\mathrm{GN}})$ | 34.52 | 43.39 | 15.19 | 23.34 | 14.44 | 40.39 | 28.55 |
| $\mathcal{RGN}_{2.0}(-3, \sigma_{\mathrm{GN}})$ | 21.38 | 24.92 | 6.58 | 7.94 | 7.20 | 16.63 | 14.11 |
| **Dense baselines** | | | | | | | |
| VICReg | 37.55 | 44.84 | 13.37 | 34.23 | 17.03 | 43.45 | 31.75 |
| SimCLR | **51.65** | **63.71** | **35.10** | **45.73** | **36.06** | **57.78** | **48.34** |
| LeJEPA | 40.05 | 56.81 | 23.42 | 43.31 | 20.70 | 51.19 | 39.25 |

*Table 12.* **Projector Dimension Ablation on ImageNet-100.** For each method and projector dimension, we report the best linear probe accuracy (%) over learning rates $\{0.3, 0.03\}$. Encoder and projector accuracies are measured using frozen features. Bold indicates the best value within each projector-dimension block.

| Method | Encoder Acc1 ↑ | Projector Acc1 ↑ |
|---|---|---|
| **Projector Dim. = 512** | | |
| VICReg | 63.72 | 57.80 |
| LeJEPA | 65.53 | 59.18 |
| $\mathcal{RGN}_{1.0}(0, \sigma_{\mathrm{GN}})$ | 67.56 | 61.34 |
| $\mathcal{RGN}_{2.0}(0, \sigma_{\mathrm{GN}})$ | **68.31** | **61.74** |
| **Projector Dim. = 2048** | | |
| VICReg | 68.73 | 61.81 |
| LeJEPA | 67.18 | 60.12 |
| $\mathcal{RGN}_{1.0}(0, \sigma_{\mathrm{GN}})$ | 69.33 | **64.90** |
| $\mathcal{RGN}_{2.0}(0, \sigma_{\mathrm{GN}})$ | **69.54** | 64.85 |

*Table 13.* $\mathcal{RGN}_{1.0}(\mu, \sigma_{\mathrm{GN}})$ **Mean-Shift Sweep (ViT) with Baselines.** We report encoder Acc1 (val_acc1), projector Acc1 (val_proj_acc1), and sparsity metrics. **Bold** indicates best in a column; underline indicates second best. For sparsity columns, lower is better (more sparse).

| Method | Enc Acc1 ↑ | Proj Acc1 ↑ | $\ell_1$ Sparsity ↓ | $\ell_0$ Sparsity ↓ |
|---|---|---|---|---|
| **Ours: $\mathcal{RGN}_{1.0}(\mu, \sigma_{\mathrm{GN}})$ (Mean Shift Value, MSV)** | | | | |
| $\mathcal{RGN}_{1.0}(1.0, \sigma_{\mathrm{GN}})$ | 74.34 | 65.60 | 0.6459 | 0.9359 |
| $\mathcal{RGN}_{1.0}(0.5, \sigma_{\mathrm{GN}})$ | 74.58 | 66.42 | 0.4768 | 0.8825 |
| $\mathcal{RGN}_{1.0}(0.0, \sigma_{\mathrm{GN}})$ | **75.44** | **67.16** | 0.2730 | 0.7721 |
| $\mathcal{RGN}_{1.0}(-0.5, \sigma_{\mathrm{GN}})$ | 74.80 | 66.86 | 0.1407 | 0.5526 |
| $\mathcal{RGN}_{1.0}(-1.0, \sigma_{\mathrm{GN}})$ | 74.18 | 65.14 | 0.0737 | 0.3067 |
| $\mathcal{RGN}_{1.0}(-1.5, \sigma_{\mathrm{GN}})$ | 74.88 | 63.70 | 0.0390 | 0.1227 |
| $\mathcal{RGN}_{1.0}(-2.0, \sigma_{\mathrm{GN}})$ | 73.54 | 60.70 | 0.0238 | 0.0523 |
| $\mathcal{RGN}_{1.0}(-2.5, \sigma_{\mathrm{GN}})$ | 72.06 | 57.96 | 0.0188 | 0.0357 |
| $\mathcal{RGN}_{1.0}(-3.0, \sigma_{\mathrm{GN}})$ | 71.64 | 57.46 | 0.0132 | 0.0220 |
| **Baselines (Dense)** | | | | |
| LeJEPA | 65.36 | 59.12 | 0.6369 | 1.0000 |
| VICReg | 72.06 | 63.56 | 0.7877 | 1.0000 |
| SimCLR | 74.18 | 66.86 | 0.6663 | 1.0000 |
| **Baselines (Sparse / NonNeg)** | | | | |
| NonNeg-VICReg | 71.64 | 65.42 | 0.5075 | 0.7066 |
| NonNeg-SimCLR | 74.48 | 63.76 | **0.0016** | **0.0023** |

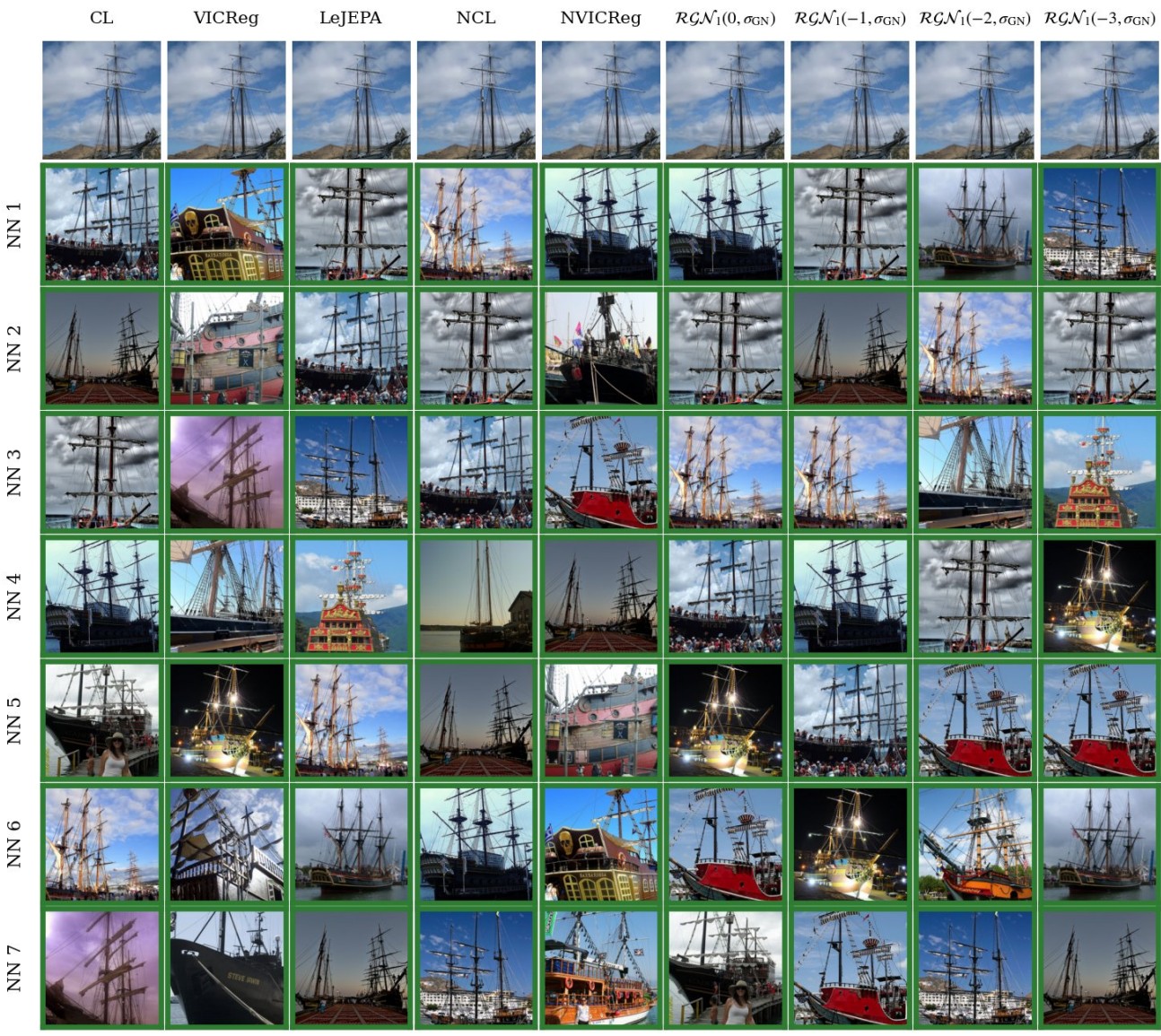

*Figure 16.* **Nearest neighbors in feature space (ImageNet synset; unambiguous class).** Top-$k$ cosine nearest neighbors in the *projector* space for a query labeled as pirate ship. Both dense and sparse methods retrieve pirate ships consistently, illustrating that even at high sparsity our models can preserve semantic consistency when the query is unambiguous.

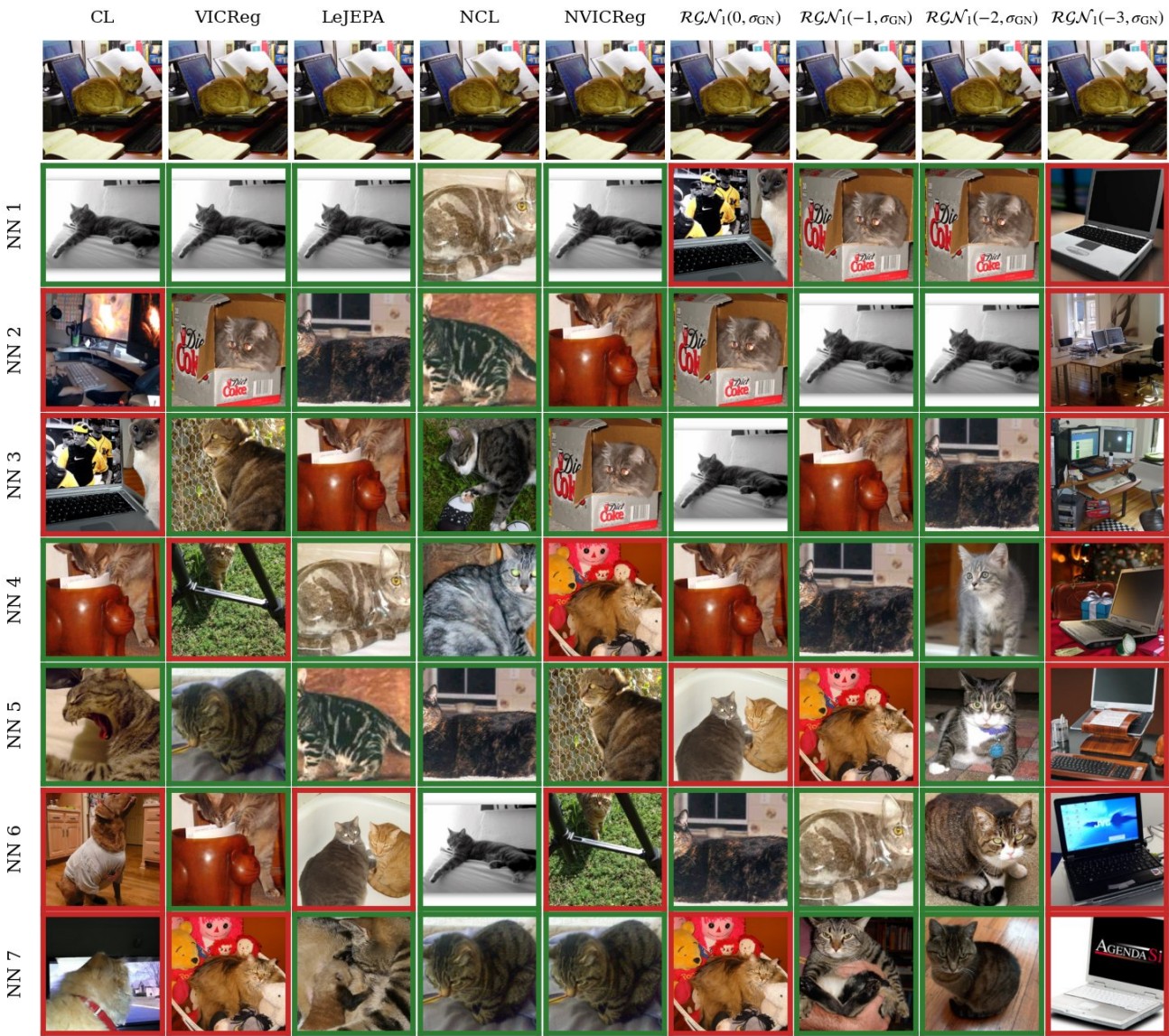

*Figure 17.* **Nearest neighbors in feature space (ImageNet synset; full scene).** Top-$k$ cosine nearest neighbors in the *projector* space for a query labeled as tabby cat (n02123045) that contains both the cat and a salient laptop/desk context. Dense methods (e.g., SimCLR) can return a mixture of cat and laptop/desk neighbors. In contrast, highly sparse $\mathcal{RGN}_{1.0}(\mu, \sigma_{\mathrm{GN}})$ variants tend to commit to a single factor: at MSV$= -2$ neighbors are predominantly tabby cats, while at MSV$= -3$ neighbors flip to predominantly laptop/desk images.

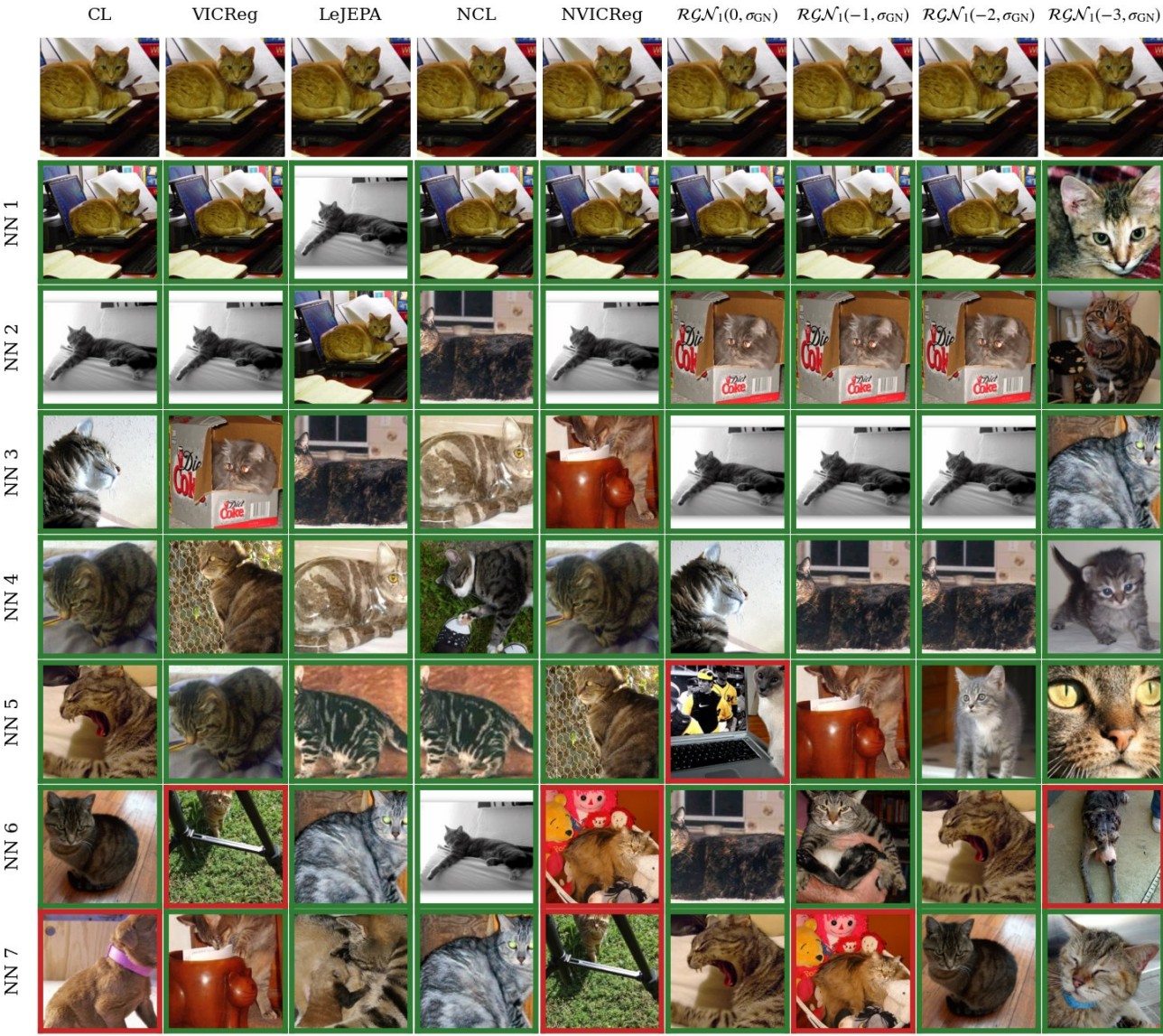

*Figure 18.* **Nearest neighbors in feature space (probe crop).** Top-$k$ cosine nearest neighbors in the *projector* space for a zoomed-in query that isolates the cat from Figure 17 by removing the competing laptop/desk cues. In this less ambiguous setting, neighbors remain tabby cats across both dense and highly sparse methods, including the most sparse $\mathcal{RGN}_{1.0}(\mu, \sigma_{\mathrm{GN}})$ variants.

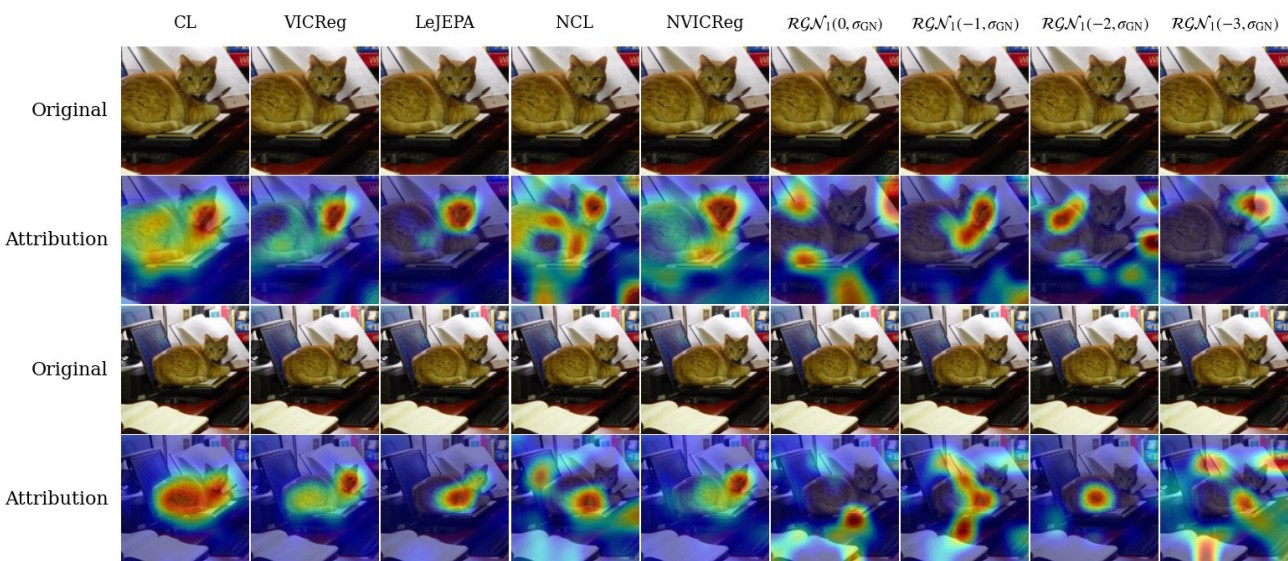

*Figure 19.* **Representation-focused attribution across methods.** Grad-CAM-style attribution maps computed on the *projector* representation for two views of the same scene (a tabby cat lying on a laptop). Rows compare dense baselines (SimCLR, VICReg, LeJEPA), sparse baselines (RepReLU variants), and our $\mathcal{RGN}_p(\mu, \sigma_{\mathrm{GN}})$ family (where $p = 1.0$ corresponds to Laplace and $\mu$ is the mean-shift value, MSV) at increasing mean-shift values, which induce increasing sparsity.

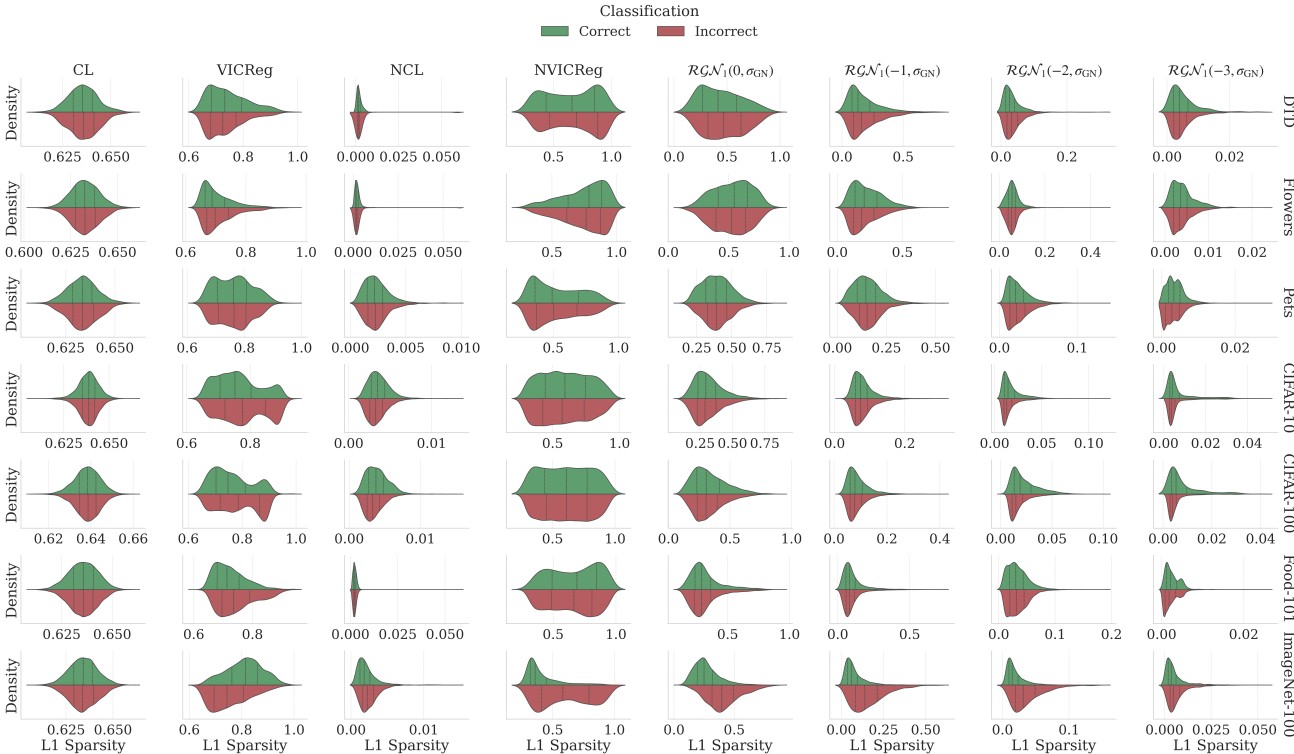

*Figure 20.* **Distribution of representation sparsity in the transfer setting.** Violin plots showing the distribution of output-feature $\ell_1$ sparsity for correctly (green) and incorrectly (red) classified samples across datasets and methods. All models are pretrained on ImageNet-100 and evaluated using a full-shot linear-probe transfer setup, where a linear classifier is trained on frozen representations using 100% of each downstream dataset. Sparsity is computed per sample from the frozen output features.

