# OpenReview forum: "Rectified LpJEPA: Joint-Embedding Predictive Architectures with Sparse and Maximum-Entropy Representations"
_ICML.cc/2026/Conference — ICML 2026 regular_

### Official Review · Reviewer_ek7H · 2026-02-27

**Soundness:** 3
**Presentation:** 3
**Significance:** 3
**Originality:** 3
**Overall Recommendation:** 4
**Confidence:** 3

**Summary:**

The paper investigates the phenomenon of representation collapse in Joint-Embedding Predictive Architectures (JEPA). The authors argue that current mitigations, such as LeJEPA, which regularize features toward isotropic Gaussian distributions, fail to capture sparsity, which is a key attribute of efficient biological and artificial representations. To address this, the authors introduce Rectified LpJEPA, which utilizes a novel distribution family called Rectified Generalized Gaussian (RGG). By employing Rectified Distribution Matching Regularization (RDMReg) via a sliced 2-Wasserstein distance, the model aligns representations to the RGG distribution, enabling controllable sparsity and non-negativity while maximizing entropy. The authors perform empirical evaluation of sparsity, feature independence and downstream performance (classification accuracy).

**Compliance With Llm Reviewing Policy:**

Affirmed.

**Final Justification:**

While the authors have addressed my primary questions, the submission still lacks empirical justification for the claimed benefits of induced sparsity. Furthermore, the core motivation remains anecdotal rather than proven. Consequently, I am maintaining my original score of 4.

**Key Questions For Authors:**

- Can the author please provide empirical and theoretical complexity analysis of the proposed framework?
- Since a core motivation for sparsity is often interpretability, have the authors investigated whether the sparse features in Rectified $L_p$JEPA correspond to more distinct visual concepts than those in LeJepa?

**Limitations:**

no, consider discussing runtime complexity of the loss function and the varying performance on classification tasks

**Strengths And Weaknesses:**

*Strengths*
- Clear motivation (mode collapse challenge and sparsity as a key property)
- The authors introduce the Rectified Generalized Gaussian (RGG) family and the RDMReg objective to enable controllable sparsity and non-negativity. While the framework recovers the Rectified Gaussian distribution at $p=2$, the extension and application of the rectified variant to the broader Generalized Gaussian family ($p \neq 2$) represents a novel contribution to representation learning.
- The paper empirically demonstrates that the method produces features with lower redundancy and higher statistical independence compared to LeJEPA

*Weaknesses*
- The empirical results on downstream tasks are currently limited to classification and not entirely convincing. While the method succeeds in inducing sparsity, the reported improvements in classification accuracy are inconsistent and vary.
- The introduction of sliced 2-Wasserstein distance (RDMReg objective) increases the computational complexity of the framework. Given the limited gains in downstream accuracy, the trade-off between this added complexity and the empirical benefit is not fully justified.

---

> ### Author Rebuttal · Authors · 2026-03-30
>
> Thank you for your constructive feedback! We address your concerns below.
>
> ## 1) Limited Experiments
>
> We have included an additional ImageNet-1K experiment. See our respone to Reviewer 6cvq titled `1) Limited Experimental Scale`.
>
> ## 2) Other Use Case of the RDMReg
>
> While we did not run interpretability analysis for sparse representation, we want to highlight another potential motivation / application.
>
> In our current paper, we motivate sparsity based on observed success of sparse coding in neuroscience and signal processing literatures.
>
> However, our ultimate target application is world modeling with sparse dynamics. According to [1], using dense isotropic Gaussian as the target distribution might necessarily increase the dimensionality of the feature representation even though the intrinsic dimensionality of the task is low. So using sparse representations might be useful to eliminate extra dimensions which can be spurious, linearly dependent, or redundant, and hence disrupt planning and control. We plan to test out this in our future work.
>
> We would like to emphasize that the primary goal of our paper is to test whether if it's even possible to have a sparse representation that's not collapsed by matching to rectified sparse distributions. It's not trivial that it will work, and we have also gone further to propose the Rectified Generalized Gaussian distribution by combining insights from information maximization, maximum-entropy, and Lp geometry.
>
> We hope that by proposing this new technique, we can inspire more thinking about "sparse representation" and in what scenario it would be useful. Ultimately, we also hope that this is one step towards ML systems that can be more efficient.
>
>
> ## 3) Efficiency Analysis
>
> Let $B$ be the batch size, $D$ be the feature dimension, and $P$ be the number of random projections. Since we're comparing RDMReg and SIGReg, we ignore the complexity involved in the same operations present in both methods like sampling the random projection vectors and normalizing them to the unit sphere.
>
> **RDMReg** consists of:
> - Sampling from target RGG: $O(BD)$
> - Projection matmul (features + target): $O(BDP)$
> - Sorting along batch for each projection: $O(P \cdot B \log B)$
> - Final reduction: $O(BP)$
>
> So the overall complexity is $O(BD + BDP + P\cdot B\log B + BP)$. The dominating term is just $O(BDP + P\cdot B\log B)$.
>
> **SIGReg** SIGReg consists of:
> - Projection matmul: $O(BDP)$
> - Characteristic function evaluation (fixed grid): $O(BPK)$
>
> So the overall complexity is $O(BDP + BPK)$, and the dominating term is just $O(BDP)$.
>
> **Key Differences**. Both of them share the same asymptotic projection cost $O(BDP)$, but RDMReg suffers from an extra $O(P\cdot B \log B)$ cost due to sorting.
>
> However, as we show in the following table where we vary batch sizes for RDMReg, the performance on CIFAR-100 quickly saturates when batch size reaches around $128$ and $256$.
>
> **Batch Size Dependency of RDMReg**
> | Metric / Batch Size | 24 | 32 | 48 | 64 | 128 | 256 | 512 |
> | --- | --- | --- | --- | --- | --- | --- | --- |
> | Acc1 | 64.88 ± 0.17 | 66.01 ± 0.44 | 67.47 ± 0.17 | 67.59 ± 0.49 | 67.69 ± 0.55 | ***68.29 ± 0.30*** | 67.88 ± 0.10 |
> | Projector Acc1 | 56.19 ± 0.07 | 58.77 ± 0.59 | 61.11 ± 0.22 | 62.24 ± 0.10 | 63.54 ± 0.10 | ***64.41 ± 0.22*** | 64.16 ± 0.12 |
>
> This is somewhat expected since we're developing this method precisely because we don't want contrastive methods which requires large number of negative samples, a.k.a. we don't want large batch size for good performance.
>
> So this extra sorting complexity $O(P\cdot B \log B)$ for RDMReg can be treated as simply a constant term since we won't need large batch size and large number of projections anyway.
>
> In practice, we observe that RDMReg is only slightly slower than SIGReg on the CIFAR-100 pretraining as shown in the following table.
>
> There could be other factors since we're measuring the overall wall time, but this is a signal that the sorting operation is not a major bottleneck.
>
> **Runtime Comparison between SIGReg and RDMReg**.
>
> | Method   | Epochs Trained | Training Time (min) ↓ | Avg. GPU Utilization (%) ↑ | GPU Memory Allocated |
> |----------|----------------|----------------------|-----------------------------|----------------------|
> | SIGReg  | 100            | 19.02                | 46.59                       | 2.65 GB              |
> | RDMReg  | 100            | 19.27                | 45.04                       | 2.63 GB              |
>
> One limitation that both methods has right now is that there are no fused kernels for some of operations in the SIGReg and RDMReg implementation. It's trivial to realize that a lot of the operations can be implemented in a fused kernel to reduce I/O communication bottleneck. We will mention this in the limitation section of the paper.
>
> **Reference**
>
> [1] LeWorldModel: Stable End-to-End Joint-Embedding Predictive Architecture from Pixels. https://arxiv.org/abs/2603.19312

---

> > ### Author Rebuttal · Reviewer_ek7H · 2026-04-02
> >
> > While the authors have addressed my primary concerns, I recommend revising the motivation text in the paper. Specifically, the text should clarify the ultimate goal of the sparsity and explicitly link this objective to the presented results. Alternatively, adding an analysis of the learned representations would help bridge the gap between the initial motivation and the findings.

---

> > > ### Author Response · Authors · 2026-04-07
> > >
> > > We are grateful to the reviewer for the constructive feedback! We will provide additional texts on the benefit of sparsity. We will incorporate our additional analysis in appendix K to the main text. Thanks to the Reviewers' comments, these will undoubtedly improve the manuscript.

---

### Official Review · Reviewer_Wp6m · 2026-03-05

**Soundness:** 3
**Presentation:** 3
**Significance:** 3
**Originality:** 3
**Overall Recommendation:** 4
**Confidence:** 4

**Summary:**

This paper proposes Rectified LpJEPA, a novel self-supervised learning architecture that introduces sparsity and non-negativity into Joint-Embedding Predictive Architectures (JEPAs). The authors construct a rigorous mathematical progression: starting from the Generalized Gaussian (GG) distribution, deriving the Truncated Generalized Gaussian (TGG) distribution (which achieves maximum differential entropy under a fixed expected $\ell_p$ norm constraint), and finally defining the Rectified Generalized Gaussian (RGG) distribution via a mixture with the Dirac measure. To achieve efficient distribution matching, the authors propose Rectified Distribution Matching Regularization (RDMReg), utilizing the Cramér-Wold theorem to decompose high-dimensional distribution matching into 1D Sliced Wasserstein distance optimizations across random projections. This framework theoretically generalizes existing Gaussian-regularized JEPA models, achieves precise analytical control over the $\ell_0$ sparsity of representations through rectification, and preserves critical image semantics even in highly sparse regimes via its maximum-entropy properties.

**Compliance With Llm Reviewing Policy:**

Affirmed.

**Final Justification:**

After reading the rebuttal and follow-up, I still land in roughly the same place overall. I think the paper is technically serious, and the theory is clearly the strongest part of the submission. The authors also did a good job clarifying the stability issue around the sliced Wasserstein matching, being more careful about what the independence results do and do not show, and toning down the robustness claims.

What still keeps me from raising my score is my main original concern: I still do not see a convincing empirical case for why sparsity is actually needed here, beyond showing that the method can produce sparse representations without collapsing. The added discussion about identifiability is interesting, but the paper still stops short of showing concrete benefits tied to sparsity itself, such as clearer interpretability or meaningfully reduced interference. So my view of the paper is somewhat better after rebuttal, but not enough to change my final recommendation.

**Key Questions For Authors:**

1. Because RGG lacks closure under linear projections and has no analytical solution for distribution matching, the model relies on non-parametric hypothesis testing. However, this sample-dependent computation inherently introduces instability and could potentially slow down convergence speed. The original text lacks an experimental analysis on how the number of projections ($N$) and batch size ($B$) affect training stability and convergence speed.

2. While the paper dedicates significant space to complex mathematical derivations for solving the distribution matching problem under non-negative truncation, the fundamental motivation "why we need these sparse representations in the first place" remains ungrounded.

Specifically, the ablation study in Experiment 5.2 attempts to justify the rectification operation, but it relies on a circular argument. Forcing a truncated, non-negative feature to match a standard Gaussian distribution (which inherently contains negative space) will naturally lead to a performance collapse due to a simple mathematical mismatch. This predictable failure does not convincingly demonstrate the conceptual necessity of your proposed innovation.

Instead of relying on marginal accuracy improvements to justify the framework, I strongly suggest that the authors consolidate the findings scattered across Section 5.5 (independence), Section 5.7 (transferability), and Appendix K (visualizations). Please integrate these insights into the main text to form a cohesive, empirical argument that demonstrates how sparsity uniquely contributes to feature decoupling, robustness against interference, and model interpretability.

3. The argument for feature independence is questionable. The low nHSIC results shown in Figure 4(b) are confusing. Because the RDMReg loss function explicitly forces the features to fit an i.i.d. target distribution with mutually uncorrelated dimensions during training, obtaining better independence metrics during testing feels like a natural spillover of the algorithm design, rather than evidence that the model learned higher-order semantic correlations. Considering that the NCL method significantly outperforms the proposed method on this metric, and that lower nHSIC does not always correspond to higher classification accuracy, how can the authors prove that this independence is "task-relevant" and not just fitting an artificially imposed statistical constraint?

4. Regarding robustness, the transfer experiments in Tables 3-8 do not show a clear advantage. On modern high-performance backbones (ViT or large ResNets), differences between regularization methods on general datasets (like ImageNet) are often masked by the massive network capacity. If Rectified LpJEPA can only achieve "competitive" accuracy on clean test sets without demonstrating leadership under severe occlusion, extreme lighting, or adversarial interference, the claim of "strong robustness" is weak. When input information is severely corrupted (e.g., dropping 70% of pixel patches), can the sparse model prove to be more robust than the dense model?

5. There is a strange observation regarding the visual attribution maps: why is the cross-view consistency of the RGN method so poor? A robust representation should exhibit stable attribution across two different views of the same scene. However, in Figure 18, under certain sparse configurations, View 1 focuses on the cat's head, while View 2 shifts entirely to the background. Why does this happen?

6. Appendix D reveals a concerning structural contradiction. To circumvent computational overhead, the authors ultimately selected $\sigma_{GN}$ as the default scheme. However, under high sparsity, this directly causes the feature variance to plummet to 0.0002. Such extreme variance shrinkage should theoretically lead to dimensional collapse. Does the model's ability to sustain training rely on the successful design of the RDMReg objective, or does it merely rely on the projection head's powerful implicit scaling capabilities? Conversely, if $\sigma_{RGN}=1$ is enforced to guarantee healthy feature variance, it severely compromises sparsity. The authors need to explain how the framework achieves true theoretical self-consistency between "maintaining healthy variance" and "extreme sparsity."

7. The compared baseline methods typically report full results on ImageNet-1K, whereas the experiments in this paper are limited to ImageNet-100.

**If the authors can successfully address my major concerns above during the rebuttal, I would be very happy to raise my score.**

**Limitations:**

Please see the questions above. Additionally, there are a few minor writing issues that need to be addressed:
- In Section 3, multiple subsections are mislabeled as "Appendix 3.1" and "Appendix 3.2". This is maybe a LaTeX referencing error and should be uniformly corrected to "Section".
- The set $S$ mentioned in Lines 160-161 has not appeared previously in the text, lacking a clear mathematical definition or background, which easily confuses readers.
- In Line 309, it should be $x$ and $x'$.
- In Section 4, the term "Non-Negative VCReg" is suddenly introduced without prior explanation of its specific inheritance relationship with the mainstream VICReg. It is recommended to add a brief explanation upon its first mention and cite the core literature of both to facilitate understanding for readers not specialized in this subfield.
- The placements of Figure 3 and Figure 4 in the main text (Page 3 and Page 5) appear much earlier than the experimental section (Section 5), which disrupts the reading flow.
- The ImageNet-100 dataset, which serves as a core experimental benchmark multiple times, lacks a standard literature citation.

**Strengths And Weaknesses:**

**Strengths**:
1. **Solid and comprehensive theoretical foundation**: The paper goes beyond intuitive architectural design. It rigorously demonstrates the maximum entropy properties of the RGG distribution under $\ell_p$ constraints from the perspectives of information theory and measure theory. Building a distribution system from the mathematical ground up provides robust theoretical backing for sparse representation learning.
2. **Translation into an engineerable framework**: The authors successfully translate complex mathematical derivations into a logically complete and practically feasible algorithmic framework. By employing the Cramér-Wold device to decompose high-dimensional matching into 1D projections, the method elegantly circumvents the computational bottlenecks typically associated with high-dimensional distribution matching.
3. **Achieving feature sparsity without sacrificing accuracy**: Experimental results demonstrate that the proposed method induces high feature sparsity while maintaining, or even slightly improving, classification accuracy (e.g., Table 1 shows the Encoder slightly outperforming the LeJEPA baseline). This "high performance and high sparsity" trade-off is highly attractive in the self-supervised learning domain.

Weaknesses:
1. **The proof of "necessity" remains superficial**: While a massive amount of mathematical derivation is dedicated to solving the distribution matching challenges caused by non-negative truncation, the explanation for why this sparsity is fundamentally necessary lacks empirical depth.
2. **Logical gaps in the experimental section**: The experiments do not cohesively build a compelling argument for the practical superiority of sparse representations over dense ones in real-world or adverse conditions.

---

> ### Author Rebuttal · Authors · 2026-03-30
>
> Thank you for your thoughtful questions and constructive feedback! We respond to your questions below.
>
> ## 1) Motivation for Sparsity
>
> See our response to Reviewer ek7H titled `2) Other Use Case of the RDMReg`.
>
> ## 2) Stability of Two-Sample Sliced Wasserstein Distance
>
> Thank you for poining this out. Here is a comparison between LeJEPA and Rectified LpJEPA for CIFAR-100 pretraining under the controlled batch size and number of projection comparison: https://imgur.com/a/KhjPmSU. We observe no instability.
>
> Intuitively, all sample-dependent computation occurs in 1D projections. By the Cramér–Wold theorem, matching high-dimensional distributions reduces to matching univariate marginals, which is statistically stable with moderate samples, unlike high-dimensional KL.
>
> See our response to Reviewer ek7H titled `3) Efficiency Analysis` for additional discussion of batch size, projections, and big-O complexity.
>
> ## 3) Misc
>
> **Rectification "Necessity"**. Fig. 3(a) studies two mismatched settings: (1) rectified features matched to non-rectified targets, and (2) non-rectified features matched to rectified targets. The former collapses; the latter performs well even if it's mismatched, and we don't have sparsit due to the lack of rectifications on the features. This shows mismatch alone does not determine performances, motivating ablation. We will revise wording to "necessity of matched rectification between features and targets."
>
> **Integration into main text**. Agreed. Due to space constraint, we emphasized RDMReg and target RGG. We will move more analysis into the main text and shift background material (e.g., Generalized Gaussian) to the appendix.
>
> **Independence and Task-Relevance.** We agree that lower nHSIC alone does not prove that the induced independence is "task-relevant." Our intention in Fig. 4(b) was more modest: since RDMReg matches the learned features to an i.i.d. rectified target distribution, nHSIC serves as a crude proxy that this reduction in higher-order dependence is reflected empirically, beyond the second-order decorrelation targeted by VICReg/NVICReg.
>
> Total correlation would be the ideal metric but is intractable to evaluate in high dimensions. We will clarify that nHSIC is only a proxy and avoid overinterpretation. Task relevance is instead supported indirectly: sparse, low-dependence features retain competitive linear-probe and transfer performance across sparsity levels.
>
> **Robustness**. We did not intend to claim strong robustness. Our evidence supports a narrower claim: Rectified LpJEPA achieves competitive transfer while enabling controllable sparsity, with systematic variation across datasets. We will soften the wording accordingly.
>
> **Visual attribution maps**. We suspect degradation arises from the final ReLU, which can block gradients in Grad-CAM style attribution technique [1]. Grad-CAM requires computing the Jacobian of the final layer with respect to an intermediate layer, and ReLU might explicitly kills the gradient.
>
> To address this, we use RepReLU on the features:
>
> ```python
> class RepReLU(nn.Module):
>     def forward(self, x: torch.Tensor) -> torch.Tensor:
>         gelu_x = F.gelu(x)
>         relu_x = F.relu(x)
>         return gelu_x - gelu_x.detach() + relu_x.detach()
> ```
> We denote this as $\mathcal{R}^{\mathrm{Rep}}\mathcal{GN}$.
>
> For the same cat image, we observe better visual attribution map performance when using RepReLU: https://imgur.com/a/kNb8IoO.
>
> We also have a boat example where both ReLU and RepReLU give us consistent visual attribution across different view of the same scene: https://imgur.com/a/iW7VGXE.
>
> We acknowledge that visual attribution map analysis in general can vary quite a lot case-by-case, and we will include a discussion this limitation in our paper.
>
> **Choice of $\sigma_{GN}$ vs $\sigma_{RGN}$**. With $\sigma_{GN}$, Rectified Gaussian can have low variance (Fig. 6), so we do not recommend using it. For $p\leq 1$ (e.g., Rectified Laplace), we observe strong performance with high sparsity despite small variance.
>
> While $\sigma_{RGN}$ enforces unit variance, it reduces sparsity. Empirically, $(p,\sigma)$ induces a sparsity–performance Pareto frontier (Fig. 10), and there is no free lunch if we suddenly use $\sigma_{RGN}$ as we mentioned in line 1203 as well. We therefore use $\sigma_{GN}$ by default for better sparsity control via $\mu$. We agree further sweeps over $p$ are needed and will expand Appendix D.
>
> **ImageNet-1k**. See our response to Reviewer 6cvq titled `1) Limited Experimental Scale`
>
> **Writing Issues**. Thanks! We will include these fix in the revision.
>
> **Reference**
>
> [1] Selvaraju, Ramprasaath R., et al. "Grad-CAM: visual explanations from deep networks via gradient-based localization." International journal of computer vision 128.2 (2020): 336-359.

---

> > ### Author Rebuttal · Reviewer_Wp6m · 2026-04-03
> >
> > Thanks to the authors for the clarifications. The added details on the independence metric and efficiency make sense, and scaling back the robustness claims definitely improves the paper.
> >
> > I am still struggling with my main concern, though. The rebuttal shows we can get sparse representations without model collapse, but it doesn't really prove why we need them in this framework. The link between sparsity and concrete benefits (like interpretability or less interference) just isn't there yet empirically. The paper is definitely in better shape now, but since this core piece is still missing, my score remains a 4.

---

> > > ### Author Response · Authors · 2026-04-07
> > >
> > > We thank the reviewer for the constructive feedback. We address your remaining concerns below.
> > >
> > > **Motivation for Sparsity**
> > >
> > > As we mention in our response to Reviewer ek7H, we have another clear use case of our RDMReg method and that's part of our on-going work. We would also like to highlight another theoretical motivation for introducing sparsity in self-supervised learning.
> > >
> > > Recent work such as Non-Negative Contrastive Learning (NCL) [1] shows that standard InfoNCE-style contrastive objectives can admit multiple equivalent solutions, leading to non-identifiable representations. By introducing non-negativity constraints—which induce sparsity and factorized structure—they establish identifiability guarantees and improved representation structure.
> > >
> > > It's quite well-known that methods like Gaussian distribution matching (LeJEPA) has the same optimal solution as contrastive learning [2, 3]. Hence, all of them suffer from this non-identifiability issue since isotropic Gaussian is rotationally invariant / density function depends only on the L2 norm. Sparse representations forces axis-aligned representations, preventing a continuous spectrum of vectors to represent the ground-truth latent. For additional motivation on feature identifiability, see [4]
> > >
> > > **Other Contribution Beyond SSL**
> > >
> > > Although ICML is primarily a machine learning venue and we also submitted to this Deep Learning->Self-Supervised Learning track, we would like to highlight another contribution of our paper that's not immediately related to SSL: the introduction of the Rectified Generalized Gaussian (RGG) distribution.
> > >
> > > Our paper is the first paper to propose the RGG distribution (section 3.4), provide sparsity and entropy characterization (section 3.5, Theorem 3.6), formalize it using measure theory (appendix B.3), compute its first and second moment formula (appendix B.4), and extensively analyzes the property of this distribution between the $\sigma$ parameter and sparsity (appendix D).
> > >
> > > All of these take significant time and we firmly believe that this particular probability distribution carries wider implication beyond our current object of interest (i.e. representation learning) and evaluation suites. To give an example, Lasso regression with $\ell_1$ penalty on the weight is equivalent to Maximum A Posteriori (MAP) estimation with a Laplace prior, whereas Ridge regression with $\ell_2$ regularization on the weight corresponds to MAP estimation with a Gaussian prior [5]. These are fundamental observations on linear regressions and their probabilistic formulations. We believe RGG might also be applicable to the linear regression setting here, but this is outside the scope of our paper and we defer this to future work.
> > >
> > > Thank you again for your insightful comments! These will undoubtedly improve the manuscript.
> > >
> > > **Reference**
> > >
> > > [1] Wang, Y., Zhang, Q., Guo, Y., & Wang, Y. (2024). Non-negative Contrastive Learning. In International Conference on Learning Representations (ICLR).
> > >
> > > [2] Betser, Roy, et al. "InfoNCE Induces Gaussian Distribution." The Fourteenth International Conference on Learning Representations, 2026
> > >
> > > [3] Wang, Tongzhou, and Phillip Isola. "Understanding contrastive representation learning through alignment and uniformity on the hypersphere." International conference on machine learning. PMLR, 2020.
> > >
> > > [4] Zhang, Qi, Yifei Wang, and Yisen Wang. "Identifiable contrastive learning with automatic feature importance discovery." Advances in Neural Information Processing Systems 36 (2023): 58461-58477.
> > >
> > > [5] Bishop, Christopher M., and Nasser M. Nasrabadi. Pattern recognition and machine learning. Vol. 4. No. 4. New York: springer, 2006.

---

### Official Review · Reviewer_6cvq · 2026-03-12

**Soundness:** 3
**Presentation:** 3
**Significance:** 3
**Originality:** 3
**Overall Recommendation:** 5
**Confidence:** 4

**Summary:**

The paper introduces Rectified LpJEPA, which equips the JEPA framework with Rectified Distribution Matching Regularization (RDMReg). RDMReg aligns the rectified representations $  \operatorname{ReLU}(z)  $ to a novel Rectified Generalized Gaussian (RGG) family $  \mathcal{RGN}_p(\mu,\sigma)  $. RGG is a mixture of a Dirac mass at zero and a truncated Generalized Gaussian; it explicitly controls the expected $  \ell_0  $-norm (Proposition 3.5) while preserving maximum-entropy properties under sparsity constraints up to Rényi-information-dimension rescaling (Theorem 3.6).

**Compliance With Llm Reviewing Policy:**

Affirmed.

**Key Questions For Authors:**

See  Weaknesses

**Limitations:**

See  Weaknesses

**Strengths And Weaknesses:**

Strengths：
1. Strong theoretical contribution

2. Elegant and practical method: RDMReg uses only random $  \ell_2  $-sphere projections + sorted sliced 2-Wasserstein (Eq. 16), is dimension-independent (N=10–20 suffices, Figure 13c), and has linear complexity. The three parameters $  \{\mu,\sigma,p\}  $ directly determine expected $  \ell_0  $-norm (Eq. 9), offering clear interpretability.

3. Clean Generalization and Unification: Rectified LpJEPA recovers LeJEPA as a special case and provides a principled inductive bias toward sparsity without sacrificing invariance or task-relevant information.

Weaknesses：

1. Limited Scale of Experiments: Main results are on ImageNet-100 and CIFAR-100. Full ImageNet-1K pretraining (with modern backbones and longer schedules) is expected for JEPA-style papers to establish broader impact.

2. Hyperparameter Guidance: While controllable sparsity is well-demonstrated, practical recommendations for choosing p, μ, and σ (especially σ_GN vs. σ_RGN) are underdeveloped. A systematic ablation across the full continuous p range would strengthen the paper.

3. Efficiency Analysis: RDMReg requires multiple random projections (N ≈ 128–256). The overhead is shown to be dimension-independent, but no wall-clock time or memory comparison with SIGReg/LeJEPA is reported.

---

> ### Author Rebuttal · Authors · 2026-03-30
>
> Thank you for your supportive feedback! We address your concerns as follows:
>
> ## 1) Limited Experimental Scale
>
> Due to limited compute budgets, we're unable to extensively run ImageNet-1k jobs where we need to do a careful grid search over learning rates, hyperparameters for the RDMReg loss term, varying values of $\mu$, varying values of $p$, and varying choices of $\sigma$, and other potentially relevant hyperparameters.
>
> That said, we did manage to run a minimal ImageNet-1k comparison between the VICReg baseline and our Rectified LpJEPA when $\mu=0.0$ and $p=1.0$ in the following table. Again, due to the compute budget, we are only able to pretrain for 100 epochs instead of 1000 epochs. However, it's a controlled experiment and we have some signals.
>
> | Method                    | Encoder Acc1 | Projector Acc1 |
> | ------------------------- | ------------ | -------------- |
> | VICReg                    | 60.52        | 46.23          |
> | $\mathcal{RGN}\_{1}(0,\sigma\_{GN})$    | 57.95        | 46.07          |
> | $\mathcal{RGN}\_{1}(0,\sigma\_{GN})$ with Eigenvector Projections | **60.63**    | **48.84**      |
>
> We observe that with 100 epochs, standard Rectified LpJEPA using only random projections underperforms VICReg. If we mix eigenvectors of the feature covariance matrices with random projection vectors, then we're able to outperform VICReg.
>
> This is an observation we have with this kind of Cramér–Wold based distribution-matching methods in general. Basically, Cramér–Wold guarantees convergence to the target Rectified Generalized Gaussian distribution asymptotically. In other words, with infinite samples, our feature distribution will have minimum total correlation, i.e. complete statistical independence between dimensions, which implies isotropic covariance.
>
> However, if we just use random projections, then RDMReg will try to minimize all high-order correlations and dependencies at the same time, without prioritizing minimizing second-order dependencies as it's what VCReg exclusively does. So by incorporating the eigenvectors, we can force RDMReg to minimize the second-order dependencies first, while also minimizing all higher-order dependencies at the same time.
>
> We have shown this too in Figure 14 of our paper. However, on both CIFAR-100 and ImageNet-100, we have observe that with 1000 epochs, Rectified LpJEPA gets enough time to also minimize second-order dependencies pretty well while also minimize higher-order dependencies. This is also what we mean by Non-Negative VCReg recovery in Section I of our paper.
>
> We also refer to our response to Reviewer XoVZ with subsection titles `3) VCReg Recovery and Cramér–Wold` for an in-depth theoretical discussion.
>
> ## 2) Hyperparameter Guidance
>
> Regarding optimal set of hyperparameters, we agree that we need to highlight a good recipe! We will include a custom section in our paper later which recommends $p=1.0$, $\sigma=\sigma_{\operatorname{GN}}$, and varying $\mu$.
>
> We have also dedicated Appendix D for an extensive discussion over which $\sigma$ we should use. In Figure 10, we have sweeped varying values of $p$ and presented a few scatter plots. However, the points are cluttered and we will consider changing that to a heatmap later for more clarity.
>
> ## 3) Efficiency Analysis
>
> See our response to Reviewer ek7H titled `3) Efficiency Analysis`.

---

> > ### Author Rebuttal · Reviewer_6cvq · 2026-04-03
> >
> > Performance on large datasets is very poor, and it also requires tuning a large number of hyperparameters; the core issues of concern have not been resolved.

---

> > > ### Author Response · Authors · 2026-04-07
> > >
> > > Thank you for your feedback! We address your remaining concerns below:
> > >
> > > **Performance on large datasets is very poor**
> > >
> > > We respectfully disagree with the claim that our method performs poorly on large-scale datasets. For our ImageNet-1K experiment, we're comparing Rectified LpJEPA against VICReg, which is a well-known SSL method.
> > >
> > > Our experiments particularly run ImageNet-1K pretraining with only **100 epochs** as opposed to the standard **1000 epochs** training due to compute budget constraints and the length of the rebuttal period. Under head-to-head comparison, our method is already outperforming VICReg.
> > >
> > > **requires tuning a large number of hyperparameters**
> > >
> > > As we mentioned, the only thing that requires tuning is the mean-shift parameter $\mu$, and the fact that we need to tune $\mu$ is the nature of learning sparse representation with controllable degree of sparsity. We recommend fixing $p=1$ and $\sigma=\sigma_{\operatorname{GN}}$ as this corresponds to the Rectified Laplace distribution with a standard choice of the scale parameter.
> > >
> > > We will add the hyperparameter recipe to the paper in our later iteration. We would like to emphasize that our method does not require EMA or stop-grad, is non-contrastive with no dimension-dependent requirements on batch size and number of projections, and we propose the Rectified Generalized Gaussian distribution as our target distribution precisely to provide a more generalized view of how we can learn sparse and yet diverse representations.
> > >
> > > Thank you again for the constructive feedback—we will incorporate these clarifications and additional results in the final version.
> > >
> > > **Reference**
> > >
> > > [1] Bardes, Adrien, Jean Ponce, and Yann LeCun. "Vicreg: Variance-invariance-covariance regularization for self-supervised learning." arXiv preprint arXiv:2105.04906 (2021).

---

### Official Review · Reviewer_XoVZ · 2026-03-14

**Soundness:** 2
**Presentation:** 2
**Significance:** 2
**Originality:** 3
**Overall Recommendation:** 4
**Confidence:** 2

**Summary:**

This paper introduces Rectified LpJEPA, a JEPA variant that replaces Gaussian distribution matching with a rectified generalized Gaussian target via a sliced two-sample regularizer called RDMReg. The goal is to learn sparse, non-negative, view-invariant representations while retaining the anti-collapse benefits of projection-based distribution matching used in prior JEPA-style methods. The paper also develops an information-theoretic framing of the target family, relating its parameters to expected sparsity and to maximum-entropy / information-dimension interpretations. Empirically, the method is evaluated on image classification benchmarks and on sparsity–performance trade-offs, and is positioned as a strict generalization of earlier Gaussian-based JEPA regularizers.

**Compliance With Llm Reviewing Policy:**

Affirmed.

**Final Justification:**

The authors addressed my concerns well.

**Key Questions For Authors:**

1. The moment formulas for the rectified generalized Gaussian appear inconsistent with standard special cases, for example the rectified Gaussian case at (p=2, \mu=0). Can the authors verify and, if needed, correct the second-moment / variance expressions, and clarify whether the experimental code used the corrected formulas? A satisfactory clarification here would materially improve my confidence in the soundness of the theoretical development.
2. Can the authors precisely state and prove the claimed “maximum entropy under sparsity constraints” result? As currently written, the paper seems to establish an entropy characterization of the rectified distribution rather than a full max-entropy theorem over sparse mixed distributions. A sharper theorem or a more modest and accurate claim would strengthen the paper substantially.
3. What is the exact relationship between RDMReg and VCReg-style covariance regularization? In particular, does the relevant result concern centered covariance or only uncentered second moments, and should the current statement be weakened to a conditional isotropy claim? Clarifying this would help me assess how much of the claimed connection is rigorous versus motivational.
4. In the maximum-entropy proposition, does the i.i.d. product form require a product support such as the positive orthant? If so, I believe the theorem statement should be restricted accordingly. A clear response would resolve an important technical ambiguity.
5. Can the authors more clearly distinguish between the idealized population-level projection-matching argument and the actual empirical objective with finitely many projections and minibatches? Even a discussion of this gap, or a weaker but precise statement about what is and is not theoretically justified, would improve the paper.

**Limitations:**

The paper appears to have limited direct negative societal risk, but I would like to see a more explicit discussion of methodological limitations. In particular, the authors should more clearly acknowledge the gap between the population-level theory and the finite empirical sliced objective, as well as the assumptions needed for the entropy-based interpretation.

**Strengths And Weaknesses:**

The paper tackles an interesting and relevant problem. Existing JEPA regularizers typically favor dense Gaussian-like representations, whereas sparse and non-negative representations are often desirable both conceptually and practically. The proposed rectified generalized Gaussian target is a meaningful idea rather than a cosmetic variant, and the link between the target parameters and expected sparsity is a nice aspect of the formulation. I also found the motivation compelling, and the method could be useful to researchers interested in combining JEPA-style invariance with explicit sparsity control. In that sense, the paper is moderately original and potentially significant.
My main reservations are about soundness of the theoretical claims. Several central claims appear stronger than what is actually proved. First, the paper’s second-moment / variance expression for the rectified generalized Gaussian seems inconsistent with standard special cases, which threatens downstream claims involving variance and anti-collapse behavior. Second, the repeated claim that the proposed rectified family preserves “maximum entropy under sparsity constraints” does not appear fully established by the current theory: the paper seems to provide an entropy characterization of the rectified mixed distribution, but not a full max-entropy theorem over sparse mixed laws with the stated constraints. Third, the claim that RDMReg recovers a non-negative VCReg-style regularizer is overstated relative to the proposition shown; the current argument looks more like a conditional isotropy statement and seems to rely on uncentered second moments rather than centered covariance. Fourth, the support assumptions in the max-entropy proposition are not fully consistent: the product/i.i.d. factorization appears justified only for product supports such as the positive orthant, not for a generic subset of (\mathbb{R}^d). Finally, the paper uses Cramér–Wold/population-level distribution matching as motivation, but does not provide a theorem connecting this to the actual finite-projection, finite-batch sliced loss used in training.
Presentation is generally decent: the paper has a coherent narrative and the high-level idea is understandable. That said, the theory section would benefit from tighter notation, earlier introduction of assumptions, and correction of a few mathematical inconsistencies and typos. My strongest reservations concern the theory rather than the experimental motivation. The paper’s significance is promising, especially if the empirical findings are strong, but the current unsupported or over-strong theory claims limit how confidently others can build on it. Overall, I view the method and problem formulation as moderately novel, but the proof techniques themselves are mostly standard and the current theoretical development needs tightening before the paper can be considered technically solid.

---

> ### Author Rebuttal · Authors · 2026-03-30
>
> Thank you for your thoughtful and constructive feedback. We address your concerns below.
>
> ## 1) Second Moment Expression
>
> You are correct—there is a typo in our proof. The correct formula for the second moment (line B.39 line B.40) is
>
> $$\mathbb{E}[X^2]=\frac{1}{2}\bigg[\mu^2\bigg(1+\operatorname{sgn}(\mu)\,P\bigg(\frac{1}{p},\frac{|\mu|^p}{p\sigma^p}\bigg)\bigg)+
> 2\mu p^{1/p}\sigma\frac{\Gamma(2/p, |\mu|^p/(p\sigma^p))}{\Gamma(1/p)}+p^{2/p}\sigma^2\textcolor{red}{\frac{\Gamma(3/p)}{\Gamma(1/p)}}\bigg(1+\operatorname{sgn}(\mu)\,P \bigg(\frac{3}{p},\frac{|\mu|^p}{p\sigma^p}\bigg)\bigg)\bigg]$$
>
> where the missing factor is $\textcolor{red}{\frac{\Gamma(3/p)}{\Gamma(1/p)}}$ highlighted in red color. This term was dropped during algebraic manipulation (lines B.59–B.63). We will fix this in the revision.
>
> With this correction, the expression is consistent with known results. According to [1], the first and second moment of the Rectified Gaussian random variable $X\sim \operatorname{ReLU}(\mathcal{N}(0,\sigma^2))$ is
>
> $$E[X]=\frac{\sigma}{\sqrt{2\pi}},\quad E[X^2]=\frac{1}{2}\sigma^2$$
>
> With the missing piece $\textcolor{red}{\frac{\Gamma(3/p)}{\Gamma(1/p)}}$, we have
>
> $$E[X^2]=\frac{1}{2} (2^{2/2}\sigma^2* \textcolor{red}{\frac{1}{2}}*1)=\frac{1}{2}\sigma^2$$
>
> which agrees with the result from [1].
>
> We would also like to point out that in our supplementary material (the `​​rectified_gengaus_mean_var_unified` function in `./solo/losses/iter_dist.py`), we have been using the correct expression for the second moment. So everything is correct in our empirical experiments.
>
> ## 2) Entropy Characterization
>
> We agree this is a subtle point. Since RGG is not absolutely continuous with respect to Lebesgue, differential entropy is not defined. We therefore use Rényi information dimension and $d$-dimensional entropy.
>
> In Theorem 3.6, we show that the $d$-dimensional entropy of RGG is an affine transformation of the differential entropy of the truncated generalized Gaussian (TGG), which is maximum differential entropy under $\ell_p$ constraints (Prop. 3.3). Thus, rectification preserves this structure up to rescaling.
>
> We do not claim RGG is maximum $d$-dimensional entropy distribution. In fact, there are no existing mathematical theories on this to the best of our knowledge. Developing a useful entropy theory for such mixed-support distributions remains an open research problem. Existing work (e.g., direct-sum entropy over lower-dimensional supports. see [2]) does not fully address our setting. We view this as future theoretical work beyond this paper’s scope.
>
> ## 3) VCReg Recovery and Cramér–Wold
>
> Cramér–Wold is inherently asymptotic: convergence in distribution requires infinitely many projections unless strong assumptions (e.g., elliptical symmetry. see [3]) are made. For general neural network features, no known non-asymptotic guarantees exist.
>
> Practically, finite projections suffice to match key statistics (variance, sparsity, etc.), but exact distributional convergence can't be verified in high dimensions due to the curse of dimensionality.
>
> Regarding VCReg: we can only claim exact recovery in the limit of infinite projections. However, if projection directions align with covariance eigenvectors, $D$ projections suffice to enforce isotropic covariance (Prop. I.1, using centered covariance). We will correct the inconsistency where uncentered covariance was used in Eq. I.198.
>
> More precisely, RDMReg includes (non-negative) VCReg. VCReg removes second-order dependencies, while RDMReg targets full independence (zero total correlation) by matching a factorial distribution. Thus, VCReg is a second-order approximation of RDMReg.
>
> See section B.14 in [4] for a similar discussion.
>
> ## 4) Support is Correct
>
> Proposition 3.3 is based on Lemma E.1, which proves the maximum-entropy result (this is for truncated generalized gaussian) when the support $S$ is any subset of $\mathbb{R}^{d}$ with positive Lebesgue measure. Independence is actually a result of maximum-entropy under $\ell_p$ norm constraints.
>
> We will update line 210-211 because this makes it seem like $S$ is the product support.
>
> ## Conclusion
>
> We hope that we have clarified some of the concerns you have, and your feedback has been really constructive! We appreciate if you would consider raising your score, if you think we have addressed your concerns. Let us know if you have any more questions!
>
> **Reference**
>
> [1] Beauchamp, M. (2018). On numerical computation for the distribution of the convolution of N independent rectified Gaussian variables. Journal de la société française de statistique , 159 (1), 88-111.
>
> [2] Martins, André FT. "Reconciling the Discrete-Continuous Divide: Towards a Mathematical Theory of Sparse Communication."
>
> [3] Fraiman, Ricardo, Leonardo Moreno, and Thomas Ransford. "A Cramér–Wold theorem for elliptical distributions." Journal of Multivariate Analysis 196 (2023): 105176.
>
> [4] LeJEPA. https://arxiv.org/pdf/2511.08544

---

> > ### Author Rebuttal · Reviewer_XoVZ · 2026-04-07
> >
> > Thank you for the careful rebuttal. It clarified several of the points I had raised, and it has improved my overall view of the paper.
> > On the moment formula, I find your response satisfactory. You identified a specific missing factor, explained where the derivation went wrong, and clarified that the implementation already used the corrected expression. For rebuttal purposes, I consider this concern addressed.
> > On the entropy issue, I think you have largely addressed my concern, though mainly by narrowing the claim rather than by strengthening the theorem. My original concern was that the paper seemed to establish an entropy characterization of the rectified mixed law, rather than a full maximum-entropy theorem over all sparse mixed distributions. In the rebuttal, you now state what I think is the more appropriate and defensible version: the result is a $d(\xi)$-dimensional entropy characterization derived from the truncated generalized Gaussian, and not a claim that RGG is the maximum $d$-dimensional entropy distribution over the full mixed-support class. This is a weaker claim, but also a much more credible one, and I find that clarification acceptable.
> > On the VCReg / Cramér–Wold point, the rebuttal was also helpful. You now state clearly that exact recovery is justified only in the infinite-projection limit, that the relevant finite-dimensional rigorous statement concerns covariance isotropy under suitably chosen projections, and that the covariance should be centered. This resolves most of my concern. I would still suggest avoiding the strongest wording, such as saying that RDMReg includes VCReg,  unless this is qualified very carefully, since what I think is rigorous content appears asymptotic or conditional rather than a literal equivalence at the level of the finite training objective. Still, overall, I think this point is mostly addressed.
> > My one remaining substantive concern is the support/factorization issue. Here I do not think the rebuttal fully closes the gap.
> > However, independence does not follow from this form alone. Independence requires the support indicator to factorize as well; otherwise, the support itself couples the coordinates. A simple non-product support, such as a triangular region in $d=2$, gives a density of the form above that is not i.i.d. For that reason, I think the generic-support maximum-entropy statement and the product-form / i.i.d. interpretation need to be separated more carefully. If the intended revision is that Proposition 3.3 gives the maximizer on a generic $S$, while the product-form / i.i.d. interpretation is invoked only for product supports such as $(0,\infty)^d$, then that would resolve my remaining concern. As the rebuttal is currently written, however, I do not think this point is fully answered.
> > Overall, the rebuttal does materially improve the paper’s technical standing in my view. The moment issue is resolved, and the clarifications regarding the entropy claim and the gap between the population-level argument and the finite objective are both reasonable and useful. My concerns are partially resolved.
> > I am keeping my original score.

---

> > > ### Author Response · Authors · 2026-04-07
> > >
> > > Thank you for your constructive feedback!
> > >
> > > **Concerns on Support**
> > >
> > > Regarding: "If the intended revision is that Proposition 3.3 gives the maximizer on a generic $S$, while the product-form / i.i.d. interpretation is invoked only for product supports such as $(0,\infty)^d$, then that would resolve my remaining concern".
> > >
> > > Yes we meant that particularly for Proposition 3.3, we can use the general support. We will revise wording anywhere else to use the product support (i.e. the general support is for (truncated) generalized gaussian; for any of our discussion in rectified generalized gaussian, we will restrict to the product support).
> > >
> > > In fact, for the purpose of the discussion of our RGG, we will do the following: replace all general support with product support, and only mention that for Proposition 3.3 in particular, this proposition applies to both general support and the product support.
> > >
> > > We agree that for any other RGG discussion, we should use the product support, and it would just be more convenient to restrict our Proposition 3.3 to product support too and only mention somewhere else that this proposition applies to general support as well.
> > >
> > > We hope this clarifies your remaining concern.
> > >
> > > **Other Issue**
> > >
> > > It seems like we have resolved your concern on "moment", "entropy", "asymptotics", etc.
> > >
> > > **Score Request**
> > >
> > > We really appreciate if you would consider raising your score, if you think our additional discussions here address your concern! Let us know if you have any more questions!
> > >
> > > We would also like to thank you again for your constructive feedback. Your detailed suggestions have undoubtly improve our manuscript and we're deeply grateful for your insightful read into our paper!

---

### Decision · Program_Chairs · 2026-04-30

**Decision:**

Accept (regular)

**Comment:**

The contribution of the paper is a principled extension of JEPA regularization from dense Gaussian to rectified generalized Gaussian. Overall, a central concept considered by this paper is that sparsity can be incorporated into JEPA objectives in a theoretically motivated and practically effective way, rather than heuristic. The reviewers found the formulation elegant and the core idea meaningful, with clear appeal for self-supervised representation learning. The paper also presents solid empirical evidence. While there were concerns about the strength and precision of several theoretical claims, the author response addressed most of these issues satisfactorily. There are still few concerns the reviewers and the authors did not get the conclusion, but it impacts the overall contribution of paper. Please include them in the revision.